# High-resolution topography of the Antarctic Peninsula combining TanDEM-X DEM and REMA mosaic

Yuting Dong[1,2], Ji Zhao[3], Dana Floricioiu[2], Lukas Krieger[2], Thomas Fritz[2], Michael Eineder[2]

[1]School of Geography and Information Engineering, China University of Geosciences, Wuhan, China
[2]Remote Sensing Technology Institute (IMF), German Aerospace Center (DLR), Oberpfaffenhofen, Germany
[3]School of Computer Science, China University of Geosciences, Wuhan, China

Correspondence: Ji Zhao (zhaoji@cug.edu.cn)

**Abstract.** The Antarctic Peninsula (AP) is one of the widely studied polar regions because of its sensitivity to climate change and potential contribution of its glaciers to global sea level rise. Precise DEMs at high spatial resolution are very
demanded for investigating the complex glacier system of the AP at fine scale. However, the two most recent circum-Antarctic DEMs, the 12-m TanDEM-X DEM (TDM DEM) from bistatic InSAR data acquired between 2013 and 2014 and the Reference Elevation Model of Antarctica mosaic (REMA mosaic) at 8-m spatial resolution derived from optical data acquired between 2011 and 2017 have specific individual limitations in this area. The TDM DEM has the advantage of good data consistency and few data voids (approx. 0.85%), but there exist residual systematic elevation errors such as phase
unwrapping errors in the non-edited DEM version. The REMA mosaic has high absolute vertical accuracy but on AP it suffers from extended areas with data voids (approx. 8%). To generate a consistent, gapless and high-resolution topography product of the AP, we fill the data voids in TDM DEM with newly processed TDM raw DEM data acquired in austral winters of 2013 and 2014 and detect and correct the residual systematic elevation errors (i.e. elevation biases) in the TDM DEM with the support of the accurately calibrated REMA mosaic. Instead of a pixelwise replacement with REMA mosaic
elevations, these are providing reference values to correct the TDM elevation biases over entire regions detected through a path propagation algorithm. The procedure is applied iteratively to gradually correct the errors in the TDM DEM from large to small scale. The proposed method maintains the characteristics of an InSAR generated DEM and is minimally influenced by temporal or penetration differences between TDM DEM and REMA mosaic. The performance of the correction is evaluated with laser altimetry data from Operation Ice Bridge and ICESat-2 missions. The overall Root Mean Square Error
(RMSE) of the corrected TDM DEM has been reduced from more than 30 m to about 10 m which together with the improved absolute elevation accuracy indicate comparable values to REMA mosaic. The generated high-resolution DEM depicts the up-to-date topography of AP in detail and can be widely used for interferometric applications as well for as glaciological studies on individual glaciers or at regional scale.

## 1 Introduction

Antarctic Peninsula (AP) glaciers (north of $70\,°\,S$) have the potential to raise the global sea level by $69\pm5$ mm (Huss and Farinotti, 2014). In the recent decades they have undergone extensive changes as a consequence of regional climate warming and oceanographic change (Cook et al., 2005;Cook et al., 2014;Cook et al., 2016;Seehaus et al., 2018;Rott et al., 2018;Rignot et al., 2019;Dryak and Enderlin, 2020). As a complex mountainous coastal glacier system the mass balance of the individual glaciers is affected by climate and oceanographic forcings and also by the subglacial and surrounding topography (Cook et al., 2012). Digital Elevation Models (DEMs) are fundamental topographic data needed for investigating glacial features and to monitor glacier dynamics at individual glaciers or at regional scales. DEMs enable the delineation of drainage basins (Cook et al., 2014;Huber et al., 2017;Krieger et al., 2020a) and quantifying glacier mass balance with the geodetic method (Abdel Jaber et al., 2019;Krieger et al., 2020b;Rott et al., 2018;Helm et al., 2014). DEMs also support the mass budget method (Rignot et al., 2011b;Shepherd et al., 2018;Sutterley et al., 2014), calculating ice velocity (Rignot et al., 2011a;Mouginot et al., 2012), and provide constraints for geodynamic and ice flow modelling (Cornford et al., 2015;Ritz et al., 2015).

The previously released DEMs of AP are mostly covering the whole Antarctic continent. They have been derived from satellite radar altimetry (Helm et al., 2014;Li et al., 2017;Slater et al., 2018), laser altimetry (DiMarzio et al., 2007), a combination of both radar and laser altimetry (Bamber et al., 2009;Griggs and Bamber, 2009), optical photogrammetry (ASTER GDEM Validation Team, 2009, 2011;Abrams et al., 2020;Howat et al., 2019), the combination of several sources of remote sensing and cartographic data (Liu et al., 2001;Fretwell et al., 2013) and single pass Synthetic Aperture Radar (SAR) interferometry of the TanDEM-X mission (German Aerospace Center DLR, 2018). In addition, regional DEMs of the marginal areas of the ice sheet have been generated from stereoscopic data (Korona et al., 2009;Fieber et al., 2018). An overview table with the parameters of the AP DEMs can be found in Table S1 of the supplementary. They reveal large elevation uncertainty, coarse resolution, voids or incomplete data coverage over Antarctica, and particularly AP because of the complex mountainous terrain and cloudy weather. To generate a more accurate surface topography data of AP, Cook et al. (2012) have created a DEM posted at 100 m by improving the ASTER GDEM datasets and smoothing the erroneous surface, but the 100-m grid size is still too coarse to analyse the glaciers' features and dynamics at fine scale. Similarly, the recently released circum-Antarctic DEM called TanDEM-X PolarDEM (Wessel et al., 2021) has some improvements (edits and filled voids) of the TanDEM-X global DEM but with 90 m posting is insufficient for the small scale features present at AP. There are numerous small outlet glaciers at the AP especially on the west coast, e.g. more than 400 glaciers have basin areas less than 5 km$^2$. The high-resolution reference DEMs can facilitate some interferometric processing steps like the removal of the reference topographic phase for estimating ice velocity using InSAR technique (Mouginot et al., 2012) or single pass InSAR DEM generation (Rott et al., 2018). Besides, the high-resolution topographic data can also be used for terrain features calculation e.g. slope, aspect, hypsometry etc. in a more detailed way (Cook et al., 2014). To meet the demand for high-resolution topography information, two DEM products have been recently released. One is the 12-m

TanDEM-X global DEM (TDM DEM) based on InSAR data acquired over Antarctica between 2013 and 2014. The second is the 8-m Reference Elevation Model of Antarctica mosaic (REMA mosaic) derived from optical data acquired between 2011 and 2017 (Howat et al., 2019). The TDM DEM is characterized by good data consistency and few data voids (approx. 0.85%), but there are residual systematic elevation errors caused by phase unwrapping (PU) in the non-edited version. The REMA mosaic has the advantage of high absolute vertical accuracy and absence of regional outliers, but has a larger amount of data voids (approx. 8%) and limited temporal consistency due to the relatively wide time span of images used to generate the DEM.

To obtain a consistent, gapless and precise DEM product at high spatial resolution of AP, these two up-to-date DEMs with comparable spatial resolutions have been combined. The main goal is to eliminate the PU errors in the TMD DEM which prevail over other error sources. Since REMA mosaic has high absolute vertical accuracy, the height difference map between these two DEM datasets can emphasize the residual PU errors as regional discrepant values with distinct boundaries to unaffected regions. To maintain the consistency of a DEM dataset in terms of acquisition time and data source we propose to correct the residual PU errors in TDM DEM based on this elevation difference map. A novel multi-scale elevation biases detection and correction algorithm relying on a reference elevation surface is applied. The present method differs from existing DEM fusion techniques which usually combine the elevation information from different DEMs equally or by certain weights (Papasaika et al., 2009;Jiang et al., 2014;Gruber et al., 2016;Dong et al., 2018). Instead, adjacent pixels with similar elevation deviations from the real surface elevation can be automatically detected and merged into a common region, and then corrected with an average vertical offset compensation value specific to each detected region. Since remaining elevation biases in the TDM DEM exist at different scales, the height offset correction is performed to gradually eliminate these errors from large to small scale. The elevation accuracy of the resulting DEM was validated with laser altimetry data to illustrate the effectiveness of the proposed algorithm.

## 2 Experimental area and data

### 2.1 Experimental area

The Antarctic Peninsula (AP) between 63 °S and 70 °S (Figure 1), belonging to Graham Land, is a long coastal area along the Weddell Sea on the east side and the Bellingshausen Sea on the west side. Based on the newest glacier inventory of AP of Cook et al. (2014) and Huber et al. (2017), there are 860 marine-terminating glaciers out of 1590 glacier basins. It has complex mountainous terrain with elevations rising steeply from sea level at the coast towards snow-covered flat plateaus located above 1500 m. The highest peaks are close to 3000 m a.s.l. The outlet glaciers and cirques lie at lower altitude and flow into ice shelves or terminate as grounded or floating tidewater glaciers. Their accumulation areas connect with the plateaus directly or through the escarpments with steep slopes.

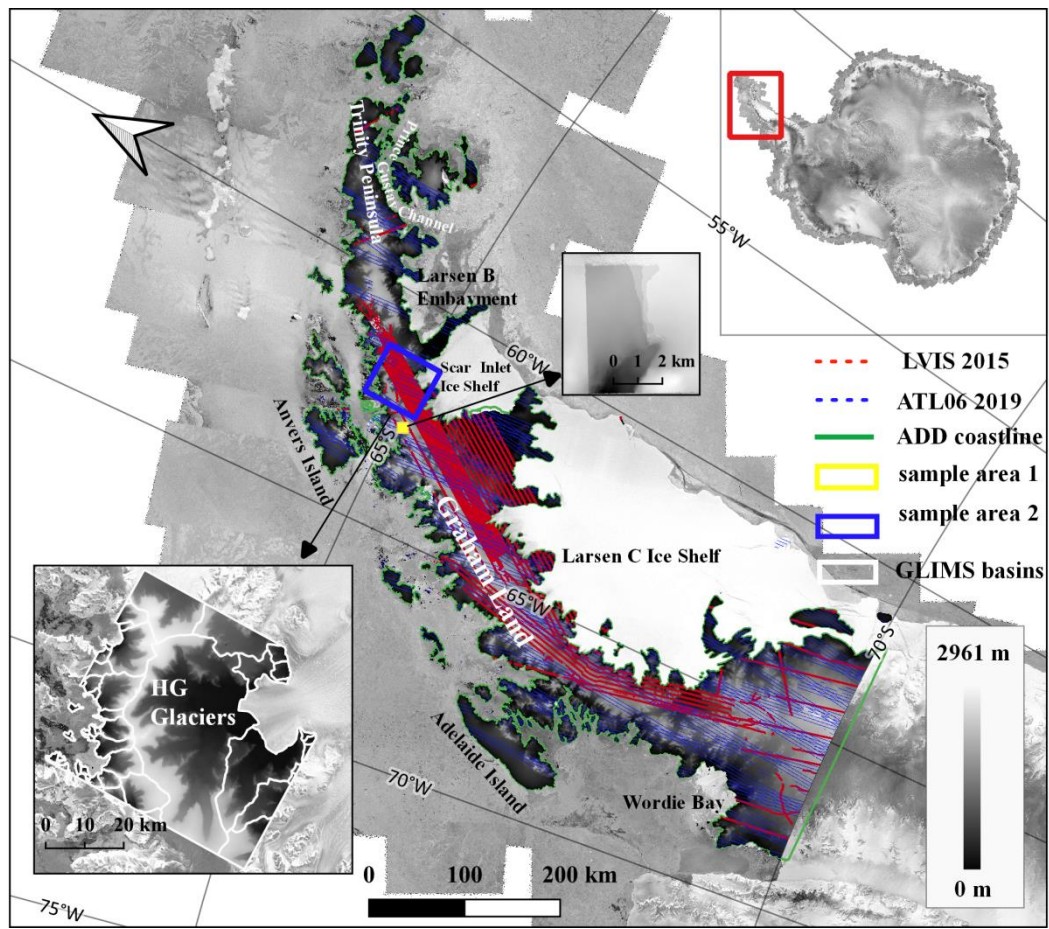

**Figure 1.** Experimental area and TanDEM-X DEM coverage over Antarctic Peninsula. Red and blue dotted lines: the footprints of the LVIS 2015 and ATL06 2019 laser points, respectively. Green outline: the coastline mask from Antarctic Digital Database (ADD). Blue and yellow boxes: sample areas of the experimental results corresponding to the two zoomed-in windows. White outline: glacier basins from GLIMS around Hektoria and Green (HG) Glaciers. Background: RADARSAT-1 Antarctic Mapping Project (RAMP) imagery mosaic (Jezek, 1999;Jezek et al., 2013) from Quantarctica (Matsuoka et al., 2021).

## 2.2 Experimental data

### 2.2.1 TanDEM-X DEM (TDM DEM)

The German TanDEM-X (TerraSAR-X add-on for Digital Elevation Measurements) mission is a bistatic SAR interferometer built by two almost identical satellites (TerraSAR-X and TanDEM-X) flying in close formation (Krieger et al., 2007;Krieger et al., 2013). The advantage of the single-pass SAR interferometer is to acquire highly coherent cross-track interferograms, which are not affected by temporal decorrelation and atmospheric phase delay. Besides, the TDM DEM is unaffected by the cloud cover or varying solar illumination conditions, which is the main reason for the completeness of TDM DEM. The

primary objective of the TanDEM-X mission was the generation of a worldwide, consistent, timely, and high-precision DEM as the basis for a wide range of scientific research. The resulting main product, the TDM DEM has a nominal pixel spacing in latitude direction of 0.4 arcsecond corresponding to approximately 12 m at the equator. The obtained overall absolute vertical accuracy at 90% confidence level is just 3.49 m and in areas covered with ice/snow like Greenland and Antarctica the obtained absolute vertical accuracy is about 6.37 m (Rizzoli et al., 2017a). The TDM DEM is also available with a pixel spacing of 1 arcsecond and 3 arcseconds (Wessel, 2018), but in our study over the AP, we focus on the nominal product at about 12-m cell size. The elevation values represent the ellipsoidal elevations relative to the WGS84 ellipsoid in the geographic coordinate system.

The bistatic InSAR data used for generating TDM DEM over Antarctica were acquired during two dedicated campaigns lasting from April to November of 2013 and 2014. The concentration of acquisition time over Antarctica reduces the temporal changes of the terrain surface and thus guarantees the consistency of the TDM DEM product. The TanDEM-X mission has acquired multi-coverage of Antarctica from different orbital directions and height ambiguities in order to compensate for geometric distortions (Gruber et al., 2016) and improve phase unwrapping with the dual-baseline phase unwrapping algorithm (Lachaise et al., 2018). However, due to the complicated mountainous terrain condition of AP, there still exist elevation biases caused by phase unwrapping errors and geometric distortions in the non-edited TDM DEM, which contaminates the vertical accuracy of TDM DEM. Besides, the elevation offset and horizontal shift because of calibration errors will also propagate into the final DEM product due to the mosaicking process.

### 2.2.2 The Reference Elevation Model of Antarctica (REMA) mosaic

The REMA DEM was generated from stereophotogrammetry with high-resolution optical, commercial satellite imagery and covers nearly 95% of the entire Antarctica. Unlike other common stereo-capable imagers such as the Advanced Spaceborne Thermal Emission and Reflection Radiometer (ASTER), the optical imagery used for generating REMA is of high spatial and radiometric resolution, which ensures accurate measurements over low-contrast ice sheet surface (Howat et al., 2019).

The REMA mosaic at 8 m resolution used in this paper was provided in $100 \times 100$ km$^2$ tiles and mosaicked from the individual time-stamped DEM strips which were quality-controlled and vertically registered (Howat et al., 2019). The absolute vertical accuracy of the REMA strips and mosaic products is less than 1 m based on validation with data acquired by three NASA Operation IceBridge (OIB) airborne lidar instruments: the Airborne Topographic Mapper (ATM), the Land, Vegetation and Ice Sensor (LVIS), and the ICECAP laser altimeter system (Howat et al., 2019). Considering the data acquisition efficiency and the effects of cloud cover and varying illumination, the limitations of the REMA mosaic are that the time span of stereo image acquisition to generate the REMA mosaic lasted for 7 years from 2011 to 2017 and the data voids in the final DEM mosaic at AP are visible in Figure S1 in the Supplementary material and are estimated to amount to approximately 8% of the AP's landmass area based on our statistics.

The REMA mosaic is referenced to WGS 84 ellipsoid and in polar stereographic projection with a central meridian of $0\degree$ and standard latitude of -71 °S. For the present paper, we converted the REMA mosaic Release 1.1 to the geographic coordinate system with the same grid size as the TDM DEM.

### 2.2.3 Laser altimetry data

In order to evaluate the vertical accuracy of the TDM DEM before and after automatic correction, we use the airborne laser altimetry data over Antarctica acquired by NASA OIB. In Figure 1 we selected the LVIS Level 2 geocoded elevation product acquired in September and October of 2015 for its dense coverage in the central part of AP (Blair and Hofton, 2019). The absolute vertical accuracy of LVIS is about 0.1 m and the footprint size is about 20–25 m (Hofton et al., 2008).

To obtain a complete evaluation of the whole experimental area, we also use the evenly distributed Level 3A geocoded
land ice height data set ATL06 acquired in austral winter of 2019 by the Advanced Topographic Laser Altimeter System (ATLAS) instrument of the ICESat-2 satellite mission (Smith et al., 2019). The ATL06 footprint is about 17 m in diameter and the surface elevation measurement accuracy is better than 0.1 m (Brunt et al., 2019). The coverage of ATL06 data is shown in Figure 1. For simplicity of the presentation, the two laser altimetry datasets used as validation data are abbreviated as LVIS 2015 and ATL06 2019.

### 2.2.4 Coastline mask

In order to improve the calculation efficiency, we use the coastline mask from Antarctic Digital Database (ADD) (https://add.data.bas.ac.uk/, last access: 13 February 2020) which is marked by the green outline in Figure 1. The current version 7.1 was last updated in August 2019. We have visually checked the agreement between the ADD coastline product and hillshade map of TDM DEM at AP and found most of the glacier fronts are contained within or agree with the ADD
coastline.

### 3 Methodology

We propose a novel method to detect and correct the residual systematic elevation errors (referred as elevation biases) in the 12 m TDM DEM facilitated by the REMA 8 m mosaic tiles. The detailed methodologies are organized in four modules (Figure 2). Firstly, we analysed the characteristics of the residual multi-scale elevation errors in the TDM DEM with the
REMA mosaic as ground reference. Secondly, we developed a path propagation algorithm to automatically detect the erroneous regions with elevation biases based on the scales of elevation errors and spatial adjacency. Thirdly, instead of replacing the erroneous elevation values with the corresponding REMA mosaic, we selected stable points from the buffer zone of the erroneous region in TDM DEM to fit a reference elevation surface and then calculate the compensation offset to the fitted elevation surface. Fourth, the above detection and correction procedure is iteratively performed to correct multi-
scale elevation errors. Details of each module are given in the following sections.

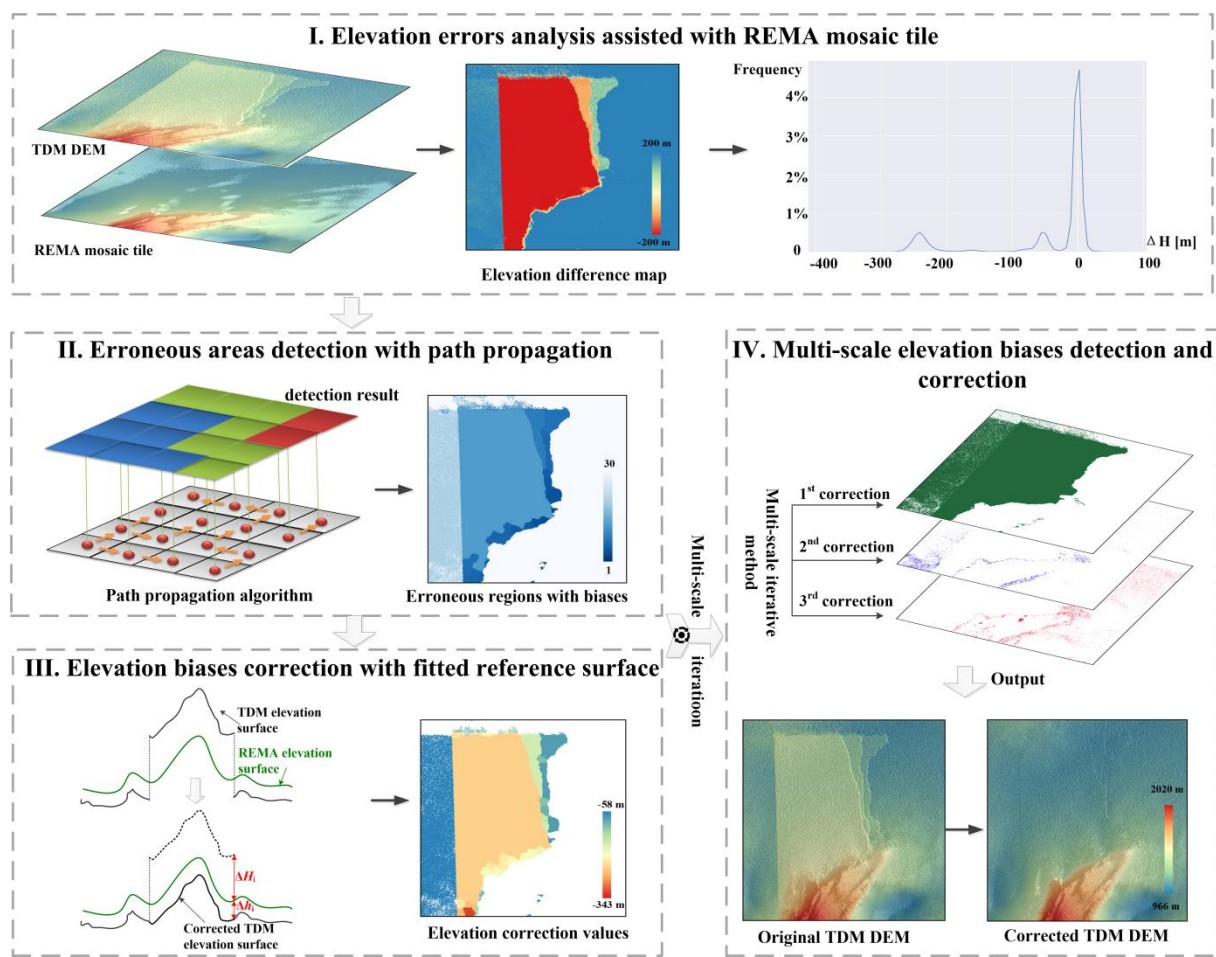

**Figure 2.** The framework of TDM DEM correction with REMA mosaic tiles organized in modules I to IV.

### 3.1 TDM DEM elevation errors analysis assisted with REMA mosaic

The remaining elevation errors in the TDM DEM include the random errors introduced from the phase noise and the systematic errors caused by the baseline calibration errors, geometric distortions such as layover and shadow, and phase unwrapping (PU) errors. Details on each of these elevation errors are given below.

(1) Random elevation errors

The random or theoretical error of the TDM DEM is linearly related to the interferometric phase error which in turn depends on the coherence and baseline geometry and slightly increases from near to far range (Rizzoli et al., 2012;Rizzoli et al., 2017a). A height error map (HEM) is accompanying each TDM DEM tile representing a combined estimate of the corresponding random elevation errors $\sigma_{ran}$ from the interferometric coherence and acquisition geometry (Wessel, 2016).

The TDM DEM is formed by a weighted average of DEMs acquired from multiple coverages to reduce the aforementioned

random errors. The relative vertical accuracy which accounts for random errors only is reported as 2.72 m and 2.41 m at the 90% confidence level for flat and steep terrain of Antarctica and Greenland (Rizzoli et al., 2017a). However, this relative vertical accuracy specification of the TDM DEM is a global statistic and local performance could be degraded, due to the presence of confined local outliers (Rizzoli et al., 2017a).

(2) Elevation errors from baseline calibration

The TDM DEM has gone through a sophisticated calibration process to improve the baseline accuracy, including instrument and baseline calibration (González et al., 2012). The correction of residual offsets and tilts in azimuth and range is performed by means of a least-squares block adjustment with ICESat laser altimetry data (Gruber et al., 2012). The final baseline accuracy is in the order of 1 mm for a ground extension of about 30 km $\times$ 50 km, which corresponds to a vertical

offset on the order of 1 m (Rizzoli et al., 2017a). A vertical offset is always accompanied by a tilt and a shift in range direction for DEM scenes. When combining the DEM scenes together into the final mosaic, the vertical offsets and horizontal shifts are likely to cause elevation biases or block-shaped elevation anomaly when there are residual phase unwrapping errors.

(3) Elevation errors from geometric distortions

At the high-relief terrain, the DEM quality is reduced due to the geometric distortions such as the layover or shadow. The erroneous regions affected by the geometric distortions can be data voids or outliers. This kind of elevation errors are usually compensated with the fusion of ascending and descending DEM acquisitions. For the TanDEM-X mission, the land mass was mapped at least twice with complementary imaging geometries and the acquired DEMs were screened and the non-discrepant data were grouped and then weighted averaged to generate the final TDM DEM, which can effectively fill in most

data gaps caused by layover or shadow (Gruber et al., 2016). The remaining elevation errors due to geometric distortions are small in spatial extent and sparsely distributed over the steep slopes oriented towards the radar or away from it.

(4) Elevation errors from phase unwrapping errors

The phase unwrapping (PU) is a crucial step in interferometric applications hence also in the surface elevation retrieval. It is very difficult to achieve an error-free PU at AP because the complex mountainous terrain is prone to cause dense fringes

and phase jumps. The phase unwrapping errors possibly exist in single TDM DEM acquisitions (TDM raw DEMs). Elevation differences between single TDM DEM acquisitions (TDM raw DEMs) accounting for PU errors are in the order of an integer multiple of the height of ambiguity (HoA). The HoA is the height that corresponds to one phase cycle after phase-to-height-conversion and is typically in the range of 30 to 80 m for most of the twin satellites baseline configurations during the nominal TanDEM-X acquisitions. Gruber et al. (2016) estimated the minimum elevation difference $dp_{PUthres}$ between

TDM raw DEMs introduced by phase unwrapping errors as $dp_{PUthres} = 0.75*min(|H_OA|)-4$ [m] considering the random elevation errors and the possible residual calibration inaccuracies (within 4 m). In our study, we detect the residual systematic elevation errors in TDM global DEM through calculating elevation difference map $\Delta H$ with REMA mosaic. The minimum elevation difference $\Delta H_{PUthres}$ due to phase unwrapping errors in TDM DEM is empirically adjusted to Eq. (1). The first item in the right part of Eq. (1) is reduced to $0.6*min(|H_OA|)$ [m] compared to Gruber et al. (2016)'s estimation because

the AP is a mountainous area with snow and ice cover which causes higher random elevation noise for both TDM DEM and REMA mosaic. $\Delta H_{PUthres}$ is then reduced by 1 m considering the calibration error of TDM global DEM is at about 1 m (Rizzoli et al., 2017a). Since the minimum HoA of the TanDEM-X mission is about 30 m, the minimum elevation difference to detect an inconsistency introduced by a PU error is approximately 17 m based on Eq. (1).

$$\Delta H_{PUthres} = 0.6 \cdot \min\left(\left|H_O A\right|\right) - 1 \tag{1}$$

Based on the above analysis, it can be seen that the remaining elevation errors in TDM DEM causing large inconsistencies are mainly introduced by the systematic elevation errors especially the PU errors. Therefore, we propose to detect and correct the remaining systematic elevation errors in the TDM DEM with the REMA mosaic as reference DEM. Figure 3a shows a sample area with PU error in TDM DEM, which is also visible as an elevation jump in the TDM DEM elevation profile crossing the boundary of the inconsistent region (Figure 3d) as well as a large discrepancy in the elevation difference map between TDM DEM and REMA mosaic (blue region in Figure 3c). In the elevation difference histogram (Figure 3e), the remaining elevation biases can be identified as side lobe adjoining the main lobe near zero. This abnormal elevation jump distinguishes the PU errors from the temporal change in elevation or penetration depth which are transitional changes with a certain trend. In other words, the elevation errors in TDM DEM caused by PU errors are characterized by local elevation discrepancies with abrupt elevation jumps at the boundary where they occur. Elevation errors caused by the geometric distortions such as layover or shadow also exist on rugged terrain. They have more variations in smaller spatial size compared to the phase unwrapping errors. In the following sections (Section 3.2 to Section 3.4), we are using the characteristics of the remaining elevation biases to detect and correct these large discrepancies present in the TDM DEM.

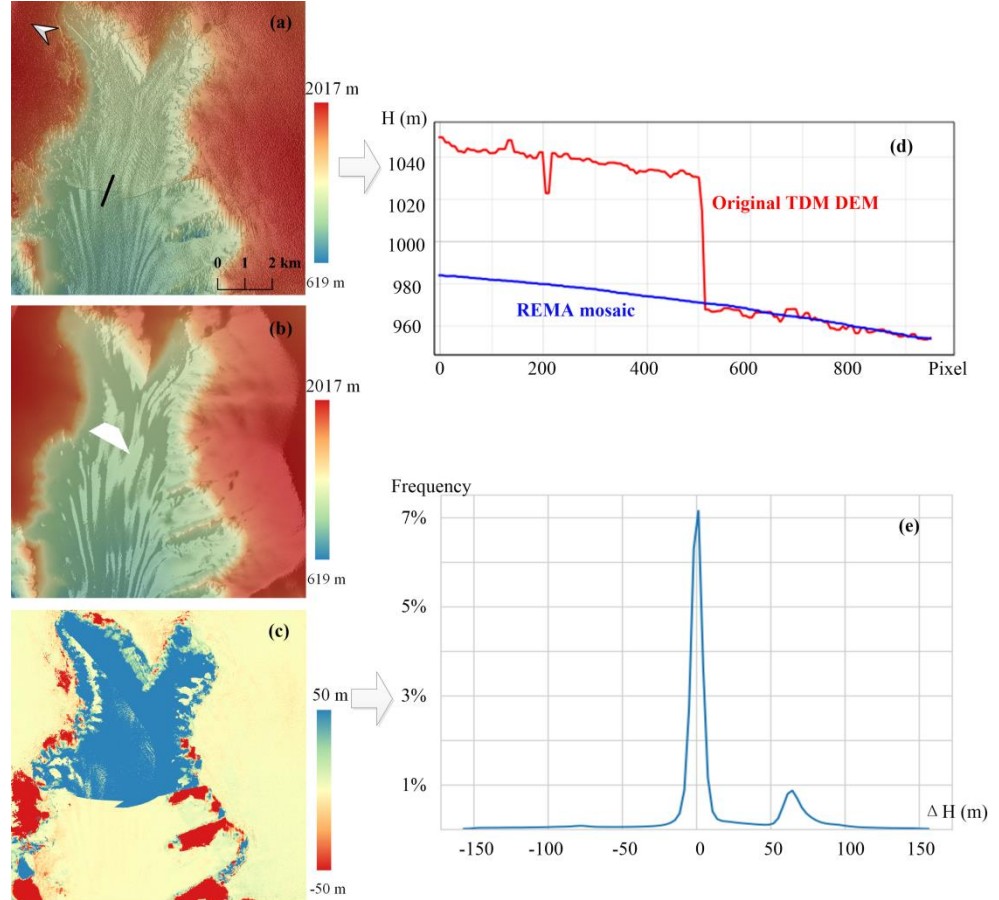

**Figure 3.** Sample area with residual phase unwrapping elevation errors. (a) TDM DEM. (b) REMA mosaic with data void in white. (c) Elevation difference map between TDM DEM and REMA mosaic. (d) Elevation profile corresponding to the black line in (a). (e) Histogram of the elevation difference map.

## 3.2 Erroneous areas detection with path propagation algorithm

The automatic detection of the areas with elevation errors from the elevation difference map between the TDM DEM and REMA mosaic is performed with a novel path propagation algorithm where neighboring pixels with similar local elevation
offset value are detected and merged into one region. Compared with the commonly used connected-component labeling method implemented on the famous image processing library (such as scikit-image library) for binary images, the path propagation is generally a region extraction algorithm, and takes the elevation difference as the input. The algorithm gradually merges and labels each adjacent target point with similar local elevation offsets along the searched path to form a correction area. An example of erroneous areas detection with path propagation algorithm is illustrated in Figure 4. The
elevation difference value in meters for each pixel used as input is shown in Figure 4a. The pixels can be divided into background and target pixels based on their corresponding elevation difference values. The background pixels (in grey in

Figure 4b and 4c) have elevation differences below a threshold value and will not be corrected in the following process. The remaining pixels are regarded as target pixels to be processed (light orange pixels in Figure 4b). The main task is to merge spatially adjacent target pixels with similar local elevation offsets into common regions. Then each of these regions can be corrected individually by the compensation value of the corresponding region. With the path propagation method, the target pixels will search their 4- or 8-neighborhood direction for homogeneous pixels. For simplicity of explanation in Figure 4b only the 4-neighborhood search is shown. The similarity criterion between the adjacent target pixels $i$, $j$ is the absolute difference of their elevations ($H$):

$$\left| H_i - H_{j \in N_i} \right| \leq T_{\Delta H} \tag{2}$$

Where $T_{\Delta H}$ is the given threshold and $N_i$ represents the neighborhood of pixel $i$. If the similarity criteria in Eq. (2) is fulfilled, the corresponding neighboring target pixels will be merged into the same region.

The new-added target pixels will continue searching their neighboring target pixels. To correctly compensate for the mean elevation offset of the erroneous region, it is important to detect the regions with homogeneous offset accurately. The existence of background pixels improves the calculation efficiency and most importantly cuts off the propagation path of target pixels. Furthermore, it is very important to properly inhibit the propagation path of target pixels not only with the background pixels but also based on the dissimilarity between the neighboring target pixels. In our example we set $T_{\Delta H}$=7 m for the neighboring pixels and pixels along the propagation path (marked with green arrows in Figure 4b) can be merged into one region. The propagation path stops at pixels with absolute elevation difference larger than $T_{\Delta H}$ as well as at the background pixels. Finally, the target pixels are merged into two separate regions according to the similarity of the elevation offsets (Figure 4c).

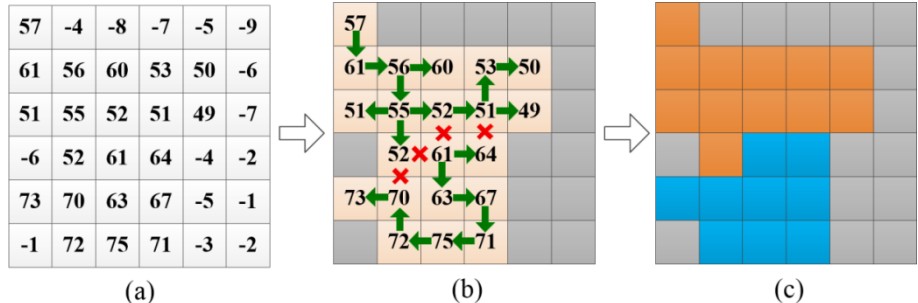

**Figure 4.** Erroneous areas detection with path propagation algorithm. (a) TDM DEM – REMA elevation difference values in meter, (b) Elevation jump detection with path propagation. Green arrows and red crosses respectively represent the adjacent pixels which can and can not be merged along the propagation path. Grey: background pixels. (c) Resulting automatically merged regions in orange and blue with mean elevation difference of 53.9 m and 68.4 m, respectively.

### 3.3 Elevation biases correction based on fitted reference surface

After merging the targeting pixels with similar local elevation offsets into regions, the elevation errors correction of TDM

DEM based on the REMA mosaic taken as reference is performed for each of these regions. Taking the differences due to the SAR signal penetration depth into snow and firn and to possible temporal elevation changes between the TDM DEM and REMA mosaic into consideration, we do not simply correct the TDM DEM to the reference elevation surface of REMA
mosaic directly. Instead, we create a buffer zone around each corrected region. Stable points whose elevation difference with REMA mosaic is less than a given threshold value are extracted from the buffer zone. The average elevation surface fitted from these selected stable points is used as a reference elevation surface for elevation offset correction. As shown in Figure 5 the correction elevation value $\Delta \hat{H}_{corr,i}$ for each region $i$ can be calculated as the sum of the average elevation difference between REMA mosaic and TDM DEM, $\Delta H_i$ and the mean elevation difference of selected stable points inside the buffer
zone, $\Delta h_i$:

$$\Delta \hat{H}_{corr,i} = \Delta H_i + \Delta h_i \qquad (3)$$

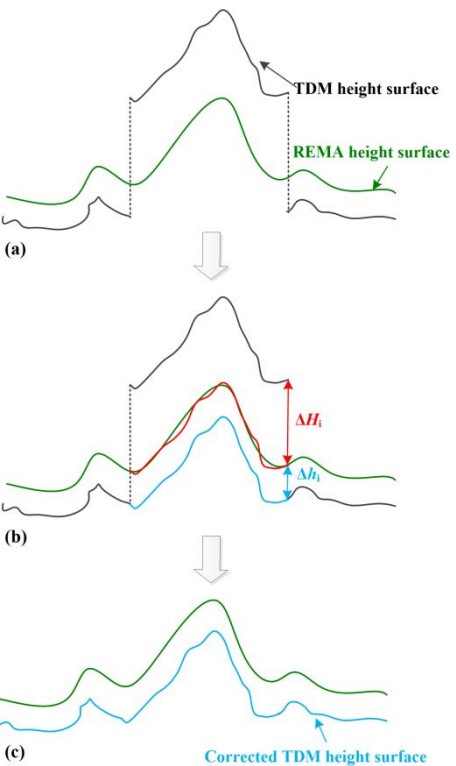

(a)

(b)

(c)

**Figure 5.** Schematic representation of the local elevation offset correction procedure. (a) Erroneous elevation jump of TDM DEM and the REMA surface elevation, (b) Correction of the jump with fitted elevation surface as in Eq. (3). Red: corrected elevation surface with mean offset, Blue: corrected elevation surface with additional fitted elevation surface. (c) Finally corrected TDM DEM.

## 3.4  Multi-scale corrections of elevation errors in TDM DEM

Since the residual elevation errors in TDM DEM may have a wide range of values, the histogram of the TDM DEM

elevation errors (quantified as differences to REMA mosaic elevations) usually has several side lobes adjoining the main zero centered lobe as illustrated in the first module of Figure 2 and Figure 3e. Actually, additional smaller side lobes within the non-zero peaks are likely. Consequently, the segmentation results of the erroneous regions from the path propagation algorithm may also contain pixels with elevation offsets at different scale. Thus, the inhomogeneous region can not be accurately corrected just with a single mean offset value.

To compensate the remaining elevation errors in the TDM DEM more accurately, we propose to adopt a multi-scale correction method to gradually correct the elevation errors from large to medium and small-scale. As described in Section 3.2 the path propagation algorithm is only performed among pixels with elevation difference larger than a certain threshold, all other pixels are labeled as background pixels. For each correction, the background pixels which do not need to be corrected are set based on the threshold that determines different scales of elevation biases. In order to achieve accurate segmentation results of the elevation inconsistency regions, the path propagation should be effectively cut off at the boundaries between different erroneous regions. The large-scale elevation errors have a clear boundary in the elevation difference map and can be easier detected and corrected. Therefore, the multi-scale correction method starts with the large-scale elevation errors by setting an empirically determined threshold on the TDM DEM to REMA mosaic elevation difference. In this step all the pixels with elevation difference less than the threshold are marked as background and no correction is applied. After this first iteration of large-scale elevation errors correction, the number of background and stable points needed for the following medium-scale or small-scale correction steps increases and the propagation path for target pixels merging can be well restricted and cut off. Hence, the homogeneity degree of the merged regions in terms of elevation errors increases accordingly. Similarly, the medium- or small-scale elevation errors are successively corrected to obtain a high-precision corrected DEM. In our experiments, we empirically set the elevation thresholds for the multi-scale correction as the large-scale errors (> 45 m), medium-scale elevation errors (> 20 m) and the small-scale elevation errors (> 5 m). Examples of how the multi-scale correction is applied are shown in Section 5.2.

**4 Experiments**

In order to test the effectiveness of the proposed algorithm at different spatial scales, we applied our methodology on a series of sample areas. Their spatial extent increased from local, about $11 \times 11$ km$^2$ large area, to glacier scale (yellow and blue rectangles in Figure 1, respectively) and finally cover the entire Antarctic Peninsula. The voids visible in the 8-m REMA mosaic tiles were filled with the oversampled 100-m REMA mosaic tiles leading to a gapless reference elevation map. To generate the elevation difference map, the REMA mosaic was resampled into the same grid size as the TDM DEM.

For quantitative accuracy evaluation of different DEM datasets, the laser altimetry data sets LVIS 2015 and ATL06 2019 described in Section 2.2.3 were used as ground reference. Differences between the laser altimetry points and the corresponding DEM elevation values obtained by bilinear interpolation were calculated. To show the vertical accuracy of different elevation intervals, we partitioned the laser points based on the elevation ranges. To compensate the temporal

differences between the laser points and the DEM datasets, we incorporated the large-scale elevation change maps from Smith et al. (2020). The annual surface elevation change was converted to elevation change by multiplying by the acquisition timespan between the DEMs and laser altimetry points. At the footprints of the ATL06 2019 laser points used for evaluation, the statistics of the temporal elevation corrections are -0.7 m (mean), -2.4 m (10% quantile) and 0.7 m (90% quantile) for REMA DEM, and -1.1 m (mean), -4.8 m (10% quantile) and 0.8 m (90% quantile) for TDM DEM. At the footprints of the LVIS 2015 laser points used for evaluation, the corresponding statistics are -0.4 m (mean), -1.2 m (10% quantile) and 0.6 m (90% quantile) for REMA DEM, and -0.6 m (mean), -3.2 m (10% quantile) and 0.3 m (90% quantile) for TDM DEM.

The temporal elevation change is compensated from the elevation difference between the DEMs and laser points before calculating the error statistics. To evaluate the elevation accuracy of the DEM datasets objectively and robustly from the outliers, we selected 4 typical error statistics: the median error (Median), Root Mean Square Error (RMSE), the 90% quantile of the absolute value of the elevation errors (which is also called the 90th percentile linear error, LE90) and the Mean Average Error (MAE). To better verify the effectiveness of the proposed correction algorithm, the error statistics were calculated independently for the corrected regions (before and after correction) and the ones left unchanged.

## 4.1 Experimental results at local area

When comparing the original TDM DEM and the corresponding REMA mosaic (Figure 6a and Figure 6b, respectively) elevation surface offsets with boundaries caused by phase unwrapping and DEM calibration errors are visible in the TDM DEM as well as in the elevation difference map (Figure 6c). We applied the proposed multi-scale correction algorithm to calculate the correction values (Figure 6d) based on the elevation difference map. Finally, the corrected TDM DEM (Figure 6e) results in a smooth elevation surface after successfully removing the elevation offsets. The elevation difference map between the corrected TDM DEM and REMA mosaic (Figure 6f) shows a more consistent trend around zero with elevation difference range reducing from ±200 m to ±50 m.

The DEM elevation errors at each LVIS and ATL06 point shown in Figure 6a were calculated and the error statistics are shown in Table 1. Considering the elevation range of the laser altimetry points located at this sample area, we merely calculate the statistics from 1500 to 2000 m. For REMA mosaic, the RMSE is less than 2 m and the MAE is no more than 3 m for both LVIS 2015 and ATL06 datasets at both corrected and uncorrected regions, indicating that the REMA mosaic is high-precision and qualified as ground reference for TDM DEM elevation errors correction. For the corrected TDM DEM at the corrected region, all the error statistics reduce considerably compared to the original TDM DEM. Specifically, the RMSE has decreased from larger than 90 m to less than 5 m and the MAE has decreased from larger than 110 m to less than 5 m at the corrected regions of TDM DEM for both LVIS 2015 and ATL06 datasets.

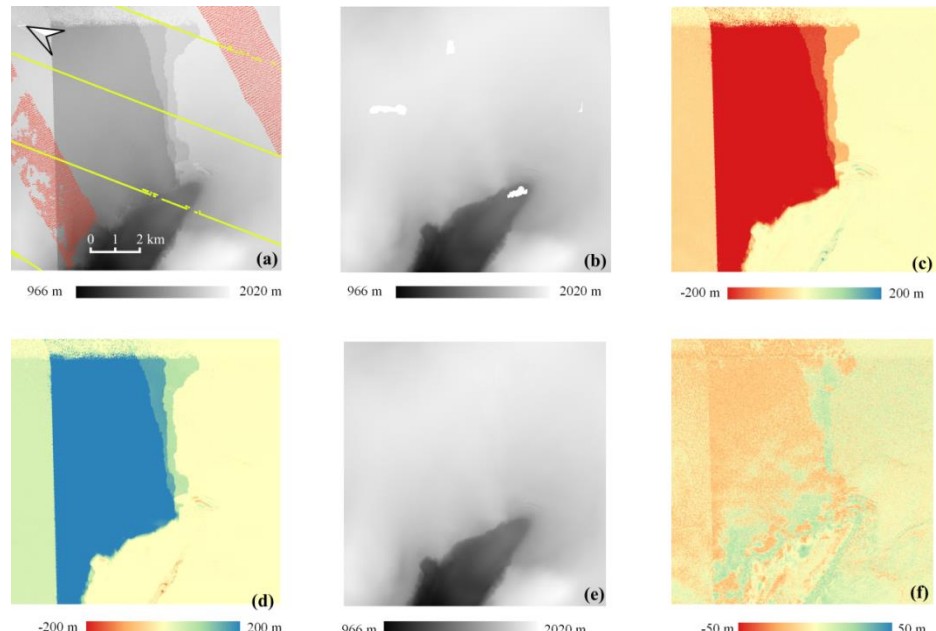

**Figure 6.** Experimental results on a local $11 \times 11$ km$^2$ sample area. (a) original TDM DEM. Red and yellow: the footprints of LVIS 2015 and ATL06 2019. (b) original REMA mosaic elevations with unfilled voids. (c) elevation difference: TDM DEM minus REMA mosaic. (d) correction map as obtained with Eq. (3). (e) corrected TDM DEM. (f) residual elevation difference: corrected TDM DEM minus REMA mosaic.

**Table 1**. Statistics of DEM elevation differences between the laser points and the REMA mosaic, original and corrected TDM DEMs over the local sample area in Figure 6. The elevation range is 1500 – 2000 m. All elevation units are in meters. Elevation difference: DEM elevation minus laser elevation. We separately considered the statistics for points affected by elevation biases where correction was applied and those falling into unchanged areas.

| | Corrected region | | | | | Unchanged region | | | | | Entire region | | | | |
|---|---|---|---|---|---|---|---|---|---|---|---|---|---|---|---|
| **LVIS 2015** | Count | Median | RMSE | LE90 | MAE | Count | Median | RMSE | LE90 | MAE | Count | Median | RMSE | LE90 | MAE |
| REMA mosaic | 24698 | 2.58 | 1.50 | 3.54 | 2.58 | 19111 | 1.10 | 0.20 | 1.35 | 1.10 | 43809 | 1.44 | 1.34 | 3.30 | 1.93 |
| Original TDM DEM | 24698 | -52.74 | 91.54 | 238.30 | 117.96 | 19111 | 2.08 | 3.30 | 6.50 | 3.14 | 43809 | -44.86 | 90.71 | 232.76 | 67.87 |
| Corrected TDM DEM | 24698 | 0.68 | 4.98 | 8.09 | 4.07 | | | | | | 43809 | 1.42 | 4.57 | 7.78 | 3.85 |
| **ATL06 2019** | Count | Median | RMSE | LE90 | MAE | Count | Median | RMSE | LE90 | MAE | Count | Median | RMSE | LE90 | MAE |
| REMA mosaic | 1118 | 1.29 | 1.12 | 2.01 | 1.35 | 493 | 0.73 | 0.93 | 1.38 | 0.83 | 1611 | 1.05 | 1.10 | 1.93 | 1.19 |
| Original TDM DEM | 1118 | -73.62 | 94.05 | 244.79 | 130.78 | 493 | 1.44 | 4.64 | 6.16 | 3.33 | 1611 | -49.51 | 99.04 | 241.97 | 91.78 |
| Corrected TDM DEM | 1118 | -2.62 | 4.36 | 7.22 | 4.04 | | | | | | 1611 | -1.20 | 4.46 | 7.17 | 3.82 |

## 4.2 Experimental results on Hektoria and Green Glaciers

For testing of our method at glacier scale, we selected an area of about 55 km $\times$ 60 km covering the Hektoria and Green (HG)
glaciers, two adjacent outlets on the Eastern AP. The elevation difference map between original TDM DEM and the void
filled REMA mosaic (Figure 7c) clearly shows regions with elevation errors of tens of meters in the TDM DEM. After
applying the same methodology demonstrated for the local experimental area (Section 4.1) the erroneous regions are
considerably reduced as revealed by the elevation difference map between the corrected TDM DEM and REMA mosaic
(Figure 7f).

The laser altimetry point measurements (coverage shown in Figure 7a) were used to validate our correction over HG area.
We divided the elevation range of the scene (18 m to 2150 m) into 5 intervals for which we calculated the corresponding
statistics of the elevation differences between the TDM and laser elevations (Table 2). There are about $3.5\times10^6$ laser
altimetry points of the LVIS 2015 dataset for validation while there are only about $1.9\times10^4$ points of ATL06 2019 dataset.
The variety in number of points can partly explain for the differences between error statistics. For example there are only 3
points of ATL06 2019 at elevation interval (>= 2000 m) that the corresponding statistics are less trustworthy than those of
the LVIS 2015 dataset with 13469 points. The steep escarpment, dropping abruptly about 500 m in elevation from 1500 to
1000 m a.s.l., generates layover and shadow in the SAR image and occlusion in the optical image and contributes to the high
elevation errors of DEMs in this interval (Rott et al., 2018). From a glaciological standpoint, the most dynamic areas are the
outlet glaciers mainly located below 1000 m a.s.l whereas the flat firn plateaus stretch above 1500 m a.s.l and have relatively
stable surface elevation (Rott et al., 2018). In Table 2, the error statistics are comparable for elevation range ($\leq$ 1000 m and $\geq$
1500 m) for both REMA mosaic and corrected TDM DEM indicating that the influence of the temporal surface elevation
change is compensated in error statistics due to the temporal change compensation. For the corrected region in Table 2, all
the error statistics have been reduced significantly. The MAE and RMSE of the original TDM DEM larger than 50 m for the
LVIS 2015 and larger than 40 m for the ATL06 2019 datasets have been reduced to about 10 m and 8 m, respectively, for
both validation datasets. The original TDM DEM at the unaltered region, the corrected TDM DEM and the REMA mosaic
have comparable elevation accuracies based on the error statistics. The effectiveness of the proposed multi-scale elevation
bias correction method is validated both qualitatively and quantitatively at the individual glacier scale.

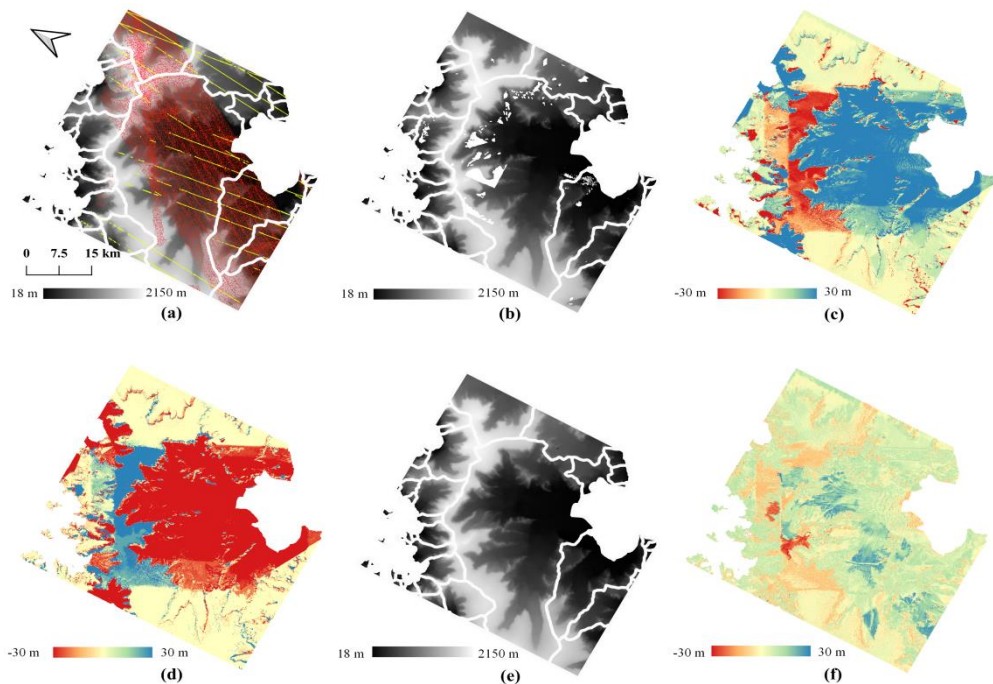

**Figure 7.** Experimental results of Hektoria and Green (HG) glaciers. (a) original TDM DEM. Red and yellow: the footprint of LVIS 2015 and ATL06 2019, respectively. (b) original REMA mosaic with voids. (c) elevation difference: TDM DEM minus REMA mosaic. (d) correction map for TDM DEM. (e) corrected TDM DEM. (f) residual elevation difference: corrected TDM DEM minus REMA mosaic.

Table 2. Statistics of DEM elevation differences between the laser altimetry points and REMA mosaic, original and corrected TDM DEMs over the Hektoria and
Green Glaciers area in Figure 7. All elevation units are in meters. Elevation difference: DEM elevation minus laser elevation. We separately considered the
statistics for points affected by elevation biases where correction was applied and those falling into unchanged areas.

| | | Corrected region | | | | | Unchanged region | | | | | Entire region | | | | |
|---|---|---|---|---|---|---|---|---|---|---|---|---|---|---|---|---|
| LVIS 2015 | Elevation Range | Count | Median | RMSE | LE90 | MAE | Count | Median | RMSE | LE90 | MAE | Count | Median | RMSE | LE90 | MAE |
| REMA mosaic | 15-2200 | 2309330 | -4.51 | 10.73 | 15.32 | 7.41 | 1184865 | 0.68 | 8.46 | 6.37 | 3.60 | 3494195 | -3.33 | 10.22 | 13.31 | 6.12 |
| | 15-500 | 1328896 | -5.18 | 7.24 | 13.86 | 6.84 | 176736 | 1.57 | 6.78 | 6.52 | 2.86 | 1505632 | -4.26 | 7.57 | 13.49 | 6.37 |
| | 500-1000 | 582698 | -3.68 | 11.57 | 17.25 | 7.86 | 372494 | 1.22 | 10.75 | 5.75 | 3.21 | 955192 | -0.83 | 11.57 | 14.93 | 6.05 |
| | 1000-1500 | 154632 | -2.40 | 23.60 | 24.85 | 11.67 | 123485 | 0.69 | 9.86 | 10.15 | 4.59 | 278117 | -0.41 | 18.83 | 18.94 | 8.52 |
| | 1500-2000 | 229635 | -4.02 | 11.46 | 15.84 | 6.83 | 432323 | -3.41 | 6.49 | 6.28 | 3.80 | 661958 | -3.78 | 8.55 | 7.03 | 4.85 |
| | 2000-2200 | 13469 | -4.40 | 3.47 | 8.70 | 5.39 | 79827 | -4.54 | 1.08 | 5.60 | 4.44 | 93296 | -4.53 | 1.69 | 5.72 | 4.57 |
| Original TDM DEM | 15-2200 | 2309330 | 46.70 | 56.58 | 69.75 | 51.73 | 1184865 | 1.55 | 7.91 | 9.78 | 4.78 | 3494195 | 22.43 | 49.27 | 61.59 | 35.81 |
| | 15-500 | 1328896 | 50.54 | 25.72 | 61.33 | 49.60 | 176736 | 6.01 | 6.64 | 12.80 | 6.50 | 1505632 | 49.09 | 27.96 | 60.56 | 44.54 |
| | 500-1000 | 582698 | 40.85 | 64.95 | 113.57 | 55.69 | 372494 | 3.29 | 6.65 | 9.29 | 4.79 | 955192 | 16.99 | 55.90 | 77.22 | 35.84 |
| | 1000-1500 | 154632 | 11.93 | 93.24 | 135.23 | 57.82 | 123485 | 1.77 | 11.05 | 14.97 | 6.49 | 278117 | 4.57 | 70.50 | 94.55 | 35.03 |
| | 1500-2000 | 229635 | -46.37 | 59.57 | 69.38 | 50.86 | 432323 | -1.11 | 7.84 | 8.26 | 4.00 | 661958 | -2.93 | 40.98 | 51.53 | 20.25 |
| | 2000-2200 | 13469 | -42.24 | 20.67 | 54.50 | 35.48 | 79827 | 0.44 | 3.54 | 5.25 | 2.58 | 93296 | -0.04 | 14.67 | 19.53 | 7.33 |
| Corrected TDM DEM | 15-2200 | 2309330 | 1.12 | 10.86 | 13.27 | 6.90 | | | | | | 3494195 | 1.32 | 9.65 | 12.27 | 6.15 |
| | 15-500 | 1328896 | 1.99 | 7.75 | 11.76 | 5.68 | | | | | | 1505632 | 2.40 | 7.69 | 11.88 | 5.76 |
| | 500-1000 | 582698 | 0.00 | 11.65 | 15.03 | 7.36 | | | | | | 955192 | 1.87 | 10.10 | 12.66 | 6.32 |
| | 1000-1500 | 154632 | 0.56 | 20.67 | 24.28 | 11.40 | | | | | | 278117 | 1.22 | 16.60 | 18.44 | 9.04 |
| | 1500-2000 | 229635 | -6.40 | 13.21 | 13.98 | 9.70 | | | | | | 661958 | -1.68 | 9.03 | 11.51 | 5.94 |
| | 2000-2200 | 13469 | -8.68 | 6.81 | 12.59 | 8.72 | | | | | | 93296 | 0.01 | 5.01 | 8.82 | 3.63 |
| ATL06 2019 | Elevation Range | Count | Median | RMSE | LE90 | MAE | Count | Median | RMSE | LE90 | MAE | Count | Median | RMSE | LE90 | MAE |
| REMA mosaic | 15-2200 | 8518 | -6.70 | 9.86 | 21.11 | 9.31 | 10372 | -4.05 | 6.26 | 11.13 | 5.73 | 18890 | -5.15 | 8.28 | 13.60 | 7.34 |
| | 15-500 | 5427 | -4.59 | 9.26 | 20.79 | 8.40 | 1430 | -5.14 | 6.55 | 8.20 | 5.29 | 6857 | -4.83 | 8.80 | 19.41 | 7.75 |
| | 500-1000 | 2156 | -10.63 | 10.47 | 24.45 | 11.63 | 4452 | -3.75 | 6.57 | 11.13 | 6.05 | 6608 | -5.52 | 8.83 | 13.23 | 7.87 |
| | 1000-1500 | 186 | 0.28 | 12.60 | 17.53 | 8.64 | 685 | -5.31 | 7.02 | 11.89 | 6.80 | 871 | -4.72 | 8.59 | 12.89 | 7.19 |
| | 1500-2000 | 746 | -9.16 | 8.66 | 16.54 | 9.27 | 3483 | -1.17 | 5.40 | 11.47 | 4.90 | 4229 | -2.29 | 6.30 | 11.57 | 5.67 |
| | 2000-2200 | 3 | -30.48 | 0.49 | 30.53 | 30.17 | 322 | -10.11 | 0.66 | 10.97 | 10.09 | 325 | -10.11 | 2.03 | 11.02 | 10.28 |

| | | | | | | | | | | | | | | | | |
|---|---|---|---|---|---|---|---|---|---|---|---|---|---|---|---|---|
| Original TDM DEM | 15-2200 | 8518 | 37.58 | 44.67 | 60.53 | 42.02 | 10372 | -2.73 | 6.07 | 9.71 | 5.21 | 18890 | 2.61 | 35.11 | 53.75 | 21.81 |
| | 15-500 | 5427 | 43.27 | 22.75 | 60.32 | 42.89 | 1430 | -4.04 | 4.83 | 8.32 | 4.84 | 6857 | 36.75 | 27.66 | 59.06 | 34.95 |
| | 500-1000 | 2156 | 29.85 | 46.31 | 54.16 | 33.98 | 4452 | -0.16 | 6.20 | 8.73 | 5.27 | 6608 | 3.19 | 31.09 | 39.96 | 14.64 |
| | 1000-1500 | 186 | 24.04 | 151.32 | 272.46 | 110.07 | 685 | -3.32 | 6.78 | 10.66 | 6.14 | 871 | -2.07 | 73.90 | 111.46 | 28.33 |
| | 1500-2000 | 746 | -41.34 | 27.91 | 66.62 | 41.92 | 3483 | -3.02 | 5.49 | 12.03 | 5.05 | 4229 | -4.35 | 18.90 | 35.78 | 11.56 |
| | 2000-2200 | 3 | -65.74 | 0.94 | 65.92 | 65.19 | 322 | -5.68 | 2.43 | 8.92 | 5.68 | 325 | -5.72 | 6.18 | 9.16 | 6.23 |
| Corrected TDM DEM | 15-2200 | 8518 | -3.68 | 10.84 | 17.61 | 8.74 | | | | | | 18890 | -3.06 | 8.59 | 14.40 | 6.82 |
| | 15-500 | 5427 | -0.59 | 10.12 | 17.18 | 7.91 | | | | | | 6857 | -1.88 | 9.29 | 16.23 | 7.27 |
| | 500-1000 | 2156 | -7.39 | 9.60 | 17.32 | 9.11 | | | | | | 6608 | -3.33 | 8.32 | 11.44 | 6.57 |
| | 1000-1500 | 186 | -1.38 | 12.39 | 14.30 | 8.72 | | | | | | 871 | -2.83 | 8.35 | 11.43 | 6.72 |
| | 1500-2000 | 746 | -13.00 | 10.57 | 20.32 | 13.54 | | | | | | 4229 | -4.11 | 7.41 | 15.00 | 6.60 |
| | 2000-2200 | 3 | -41.72 | 0.90 | 42.14 | 41.37 | | | | | | 325 | -5.59 | 4.34 | 8.97 | 5.94 |

## 4.3 Experimental results on Antarctic Peninsula

The multi-scale elevation errors correction was also applied to entire Antarctic Peninsula north of 70°S (Figure 8 left) covering about 95 000 km$^2$. Because the corrections are not visible over such a large area, we show the results on the detailed views of three sample areas marked as A, B and C (Figure 8a and Figure 8b). Within each area, the corrected elevations become smooth and continuous with elevation jumps successfully eliminated. The corrected TDM DEM was evaluated with the LVIS 2015 and ATL06 2019 datasets covering entire AP according to the footprints shown in Figure 1. The statistics of the DEM elevation errors at the laser points are presented in Table 3. There are about $3.1 \times 10^7$ and $0.8 \times 10^6$ laser points of the LVIS 2015 and ATL06 2019 datasets respectively, which are enough validation points to verify the elevation accuracy of the DEM datasets. In terms of the whole experimental area, the RMSEs of the plateaus above 1500 m are the smallest because of the flat topography while the RMSEs of the elevation intervals between 500 m and 1500 m are larger due to the existence of the escarpments. The elevation errors of the corrected region of the TDM DEM decrease clearly after the PU error correction. The RMSEs have reduced from about 100 m to 20 m and MAEs have decreased from about 60 m to 10 m for the LVIS 2015 and ATL06 2019 datasets. At the unaltered region, the RMSEs and MAEs are all less than 5 m for the TDM DEM and REMA mosaic for both LVIS 2015 and ATL06 2019 datasets. The error statistics in Table 3 indicate that the corrected TDM DEM has comparable elevation accuracy with the REMA mosaic.

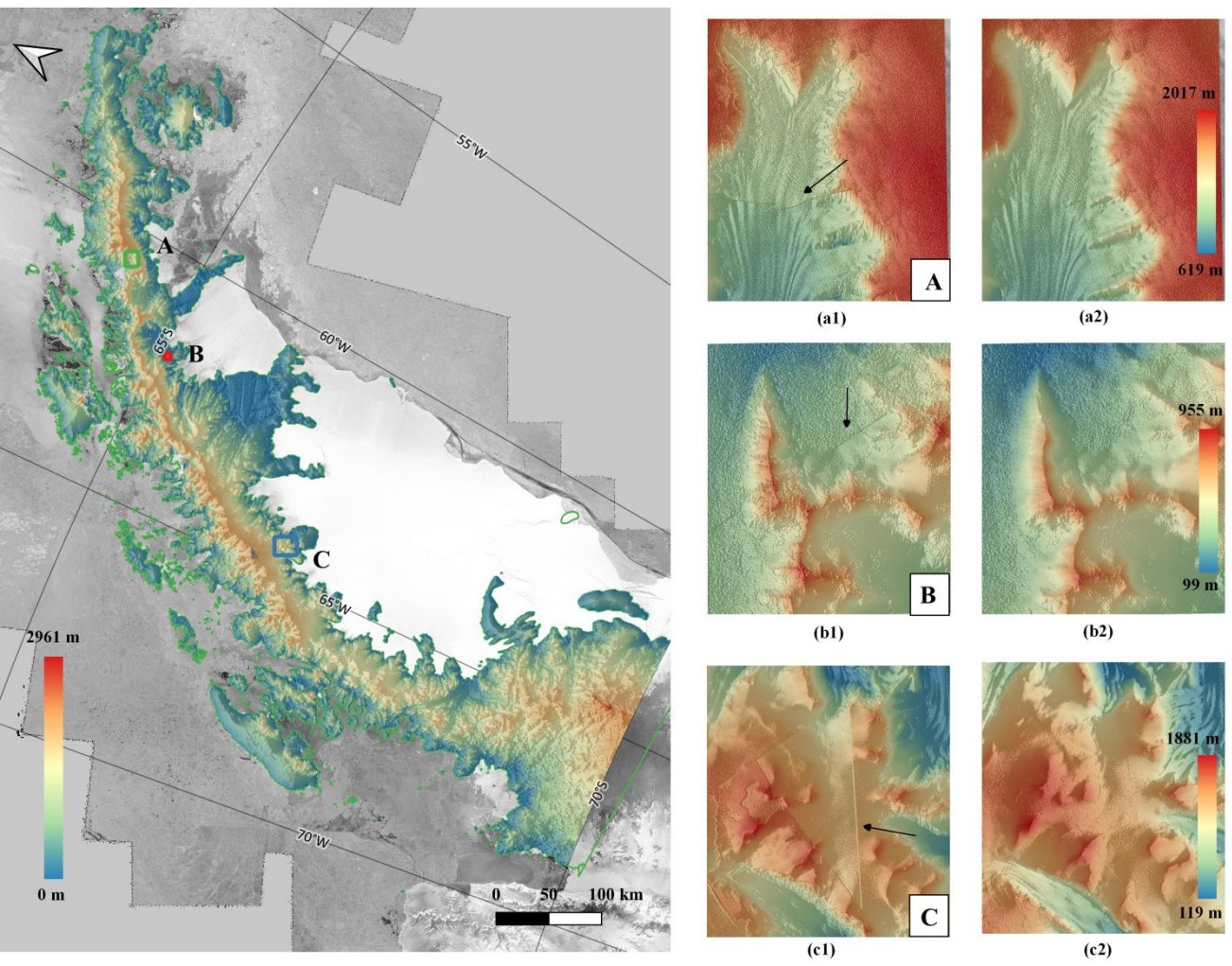

**Figure 8.** Left: Corrected TDM DEM of the Antarctic Peninsula and the location of the sample areas. Right: comparison of original TDM DEM (a) with the corrected TDM DEM (b) at the sample areas A, B and C. Black arrows point to the boundaries of the erroneous areas which have to be eliminated.

**Table 3.** Statistics of DEM elevation differences between the laser points and the REMA mosaic, original and corrected TDM DEMs over Antarctic Peninsula in Figure 8. All elevation units are in meters. Elevation difference: DEM elevation minus laser elevation. We separately considered the statistics for points affected by elevation biases where correction was applied and those falling into unchanged areas.

| | | Corrected region | | | | | Unchanged region | | | | | Entire region | | | | |
|---|---|---|---|---|---|---|---|---|---|---|---|---|---|---|---|---|
| **LVIS 2015** | Elevation Range | Count | Median | RMSE | LE90 | MAE | Count | Median | RMSE | LE90 | MAE | Count | Median | RMSE | LE90 | MAE |
| | 15-3000 | 4509465 | -2.31 | 24.66 | 22.42 | 11.53 | 26236208 | 0.93 | 3.56 | 5.11 | 2.22 | 30745673 | 0.83 | 10.06 | 6.83 | 3.59 |
| | 15-500 | 1739861 | -4.28 | 17.54 | 16.07 | 9.03 | 5581644 | 0.84 | 3.32 | 3.97 | 1.92 | 7321505 | 0.52 | 9.35 | 8.37 | 3.61 |
| REMA mosaic | 500-1000 | 1414749 | -2.25 | 29.15 | 30.60 | 14.23 | 5991863 | 0.79 | 4.26 | 5.08 | 2.36 | 7406612 | 0.68 | 13.35 | 9.08 | 4.63 |
| | 1000-1500 | 742678 | 1.11 | 32.59 | 34.10 | 15.51 | 7164443 | 1.11 | 3.62 | 5.78 | 2.41 | 7907121 | 1.11 | 10.57 | 6.83 | 3.64 |
| | 1500-2000 | 571826 | 1.17 | 17.33 | 15.07 | 7.58 | 6856833 | 0.93 | 2.96 | 5.05 | 2.11 | 7428659 | 0.93 | 5.59 | 5.41 | 2.53 |
| | 2000-3000 | 40351 | 0.62 | 20.11 | 8.82 | 7.49 | 641425 | 1.50 | 3.24 | 5.13 | 2.70 | 681776 | 1.48 | 5.91 | 5.20 | 2.99 |
| | 15-3000 | 4509465 | 13.28 | 90.66 | 117.73 | 53.45 | 26236208 | 2.13 | 4.29 | 6.17 | 3.20 | 30745673 | 2.31 | 34.95 | 19.01 | 10.57 |
| | 15-500 | 1739861 | 47.05 | 43.46 | 62.36 | 45.23 | 5581644 | 2.80 | 3.94 | 7.32 | 3.70 | 7321505 | 3.36 | 25.51 | 50.55 | 13.57 |
| Original TDM DEM | 500-1000 | 1414749 | 11.48 | 86.42 | 125.51 | 49.27 | 5991863 | 2.53 | 5.06 | 7.65 | 3.72 | 7406612 | 2.81 | 38.15 | 23.77 | 12.42 |
| | 1000-1500 | 742678 | -0.45 | 113.18 | 169.44 | 62.78 | 7164443 | 1.63 | 3.97 | 5.11 | 2.74 | 7907121 | 1.61 | 36.29 | 7.95 | 8.38 |
| | 1500-2000 | 571826 | -46.74 | 104.26 | 149.65 | 73.33 | 6856833 | 1.39 | 3.91 | 5.69 | 2.83 | 7428659 | 1.17 | 34.32 | 7.57 | 8.26 |
| | 2000-3000 | 40351 | -21.72 | 193.13 | 251.73 | 100.69 | 641425 | 1.92 | 3.93 | 6.47 | 3.25 | 681776 | 1.72 | 51.94 | 7.32 | 9.02 |
| | 15-3000 | 4509465 | 1.48 | 24.58 | 21.74 | 11.48 | | | | | | 30745673 | 2.09 | 10.16 | 8.38 | 4.46 |
| | 15-500 | 1739861 | 2.29 | 17.78 | 14.96 | 8.55 | | | | | | 7321505 | 2.76 | 9.36 | 9.62 | 4.85 |
| Corrected TDM DEM | 500-1000 | 1414749 | 1.33 | 29.09 | 30.19 | 14.03 | | | | | | 7406612 | 2.42 | 13.46 | 10.66 | 5.70 |
| | 1000-1500 | 742678 | 1.33 | 32.46 | 33.29 | 15.33 | | | | | | 7907121 | 1.62 | 10.58 | 6.67 | 3.96 |
| | 1500-2000 | 571826 | -1.11 | 17.51 | 14.66 | 9.21 | | | | | | 7428659 | 1.29 | 6.02 | 6.77 | 3.42 |
| | 2000-3000 | 40351 | -1.10 | 20.13 | 12.82 | 9.47 | | | | | | 681776 | 1.74 | 6.37 | 7.29 | 3.73 |
| **ATL06 2019** | Elevation Range | Count | Median | RMSE | LE90 | MAE | Count | Median | RMSE | LE90 | MAE | Count | Median | RMSE | LE90 | MAE |
| | 15-3000 | 40893 | 0.03 | 23.54 | 17.48 | 9.47 | 786528 | 0.11 | 3.92 | 4.75 | 2.12 | 827421 | 0.11 | 6.48 | 5.27 | 2.48 |
| | 15-500 | 11018 | -2.48 | 24.95 | 22.73 | 12.30 | 200171 | 0.26 | 4.87 | 5.96 | 2.57 | 211189 | 0.23 | 7.42 | 6.76 | 3.08 |
| REMA mosaic | 500-1000 | 9607 | -2.95 | 32.77 | 24.15 | 13.88 | 183561 | -0.25 | 3.42 | 5.41 | 2.31 | 193168 | -0.28 | 8.05 | 6.35 | 2.89 |
| | 1000-1500 | 7057 | 0.75 | 25.24 | 15.36 | 9.19 | 158079 | 0.29 | 4.52 | 4.36 | 1.88 | 165136 | 0.30 | 6.84 | 4.70 | 2.19 |
| | 1500-2000 | 11460 | 1.24 | 8.68 | 9.28 | 4.16 | 210538 | -0.14 | 2.77 | 3.90 | 1.79 | 221998 | -0.09 | 3.34 | 4.13 | 1.91 |
| | 2000-3000 | 1751 | 1.18 | 6.31 | 7.30 | 3.35 | 34179 | 0.84 | 2.13 | 3.46 | 1.55 | 35930 | 0.84 | 2.50 | 3.64 | 1.63 |

| | Range | N | | | | | N | | | | | N | | | | |
|---|---|---|---|---|---|---|---|---|---|---|---|---|---|---|---|---|
| Original TDM DEM | 15-3000 | 40893 | -2.28 | 129.42 | 198.82 | 69.84 | 786528 | 0.36 | 4.31 | 5.89 | 2.81 | 827421 | 0.35 | 30.47 | 7.08 | 6.12 |
| | 15-500 | 11018 | 22.81 | 43.10 | 60.54 | 32.78 | 200171 | 1.41 | 5.24 | 7.00 | 3.16 | 211189 | 1.50 | 11.89 | 8.59 | 4.71 |
| | 500-1000 | 9607 | 2.60 | 84.04 | 97.18 | 40.32 | 183561 | 0.38 | 4.42 | 6.67 | 3.09 | 193168 | 0.41 | 19.31 | 7.62 | 4.94 |
| | 1000-1500 | 7057 | -8.96 | 134.47 | 244.50 | 88.04 | 158079 | -0.03 | 3.53 | 5.18 | 2.43 | 165136 | -0.07 | 30.62 | 5.95 | 6.09 |
| | 1500-2000 | 11460 | -54.02 | 136.03 | 238.52 | 101.98 | 210538 | -0.32 | 3.61 | 5.11 | 2.54 | 221998 | -0.46 | 38.04 | 6.30 | 7.67 |
| | 2000-3000 | 1751 | -46.50 | 278.26 | 549.20 | 181.24 | 34179 | -0.90 | 3.15 | 4.95 | 2.65 | 35930 | -1.03 | 71.03 | 5.68 | 11.35 |
| Corrected TDM DEM | 15-3000 | 40893 | -1.09 | 22.95 | 16.73 | 9.74 | | | | | | 827421 | 0.33 | 6.61 | 6.57 | 3.22 |
| | 15-500 | 11018 | 0.73 | 23.63 | 20.30 | 11.46 | | | | | | 211189 | 1.40 | 7.43 | 7.80 | 3.62 |
| | 500-1000 | 9607 | -1.72 | 31.78 | 21.02 | 13.18 | | | | | | 193168 | 0.32 | 8.31 | 7.29 | 3.63 |
| | 1000-1500 | 7057 | -1.43 | 24.91 | 14.24 | 9.27 | | | | | | 165136 | -0.07 | 6.20 | 5.69 | 2.79 |
| | 1500-2000 | 11460 | -1.82 | 9.85 | 11.76 | 6.10 | | | | | | 221998 | -0.36 | 4.16 | 5.90 | 2.85 |
| | 2000-3000 | 1751 | -0.47 | 8.76 | 10.10 | 5.75 | | | | | | 35930 | -0.87 | 3.82 | 5.64 | 2.93 |

# 5 Discussion

## 5.1 The effectiveness of the proposed method for the different elevation error patterns

The results presented in Section 4 demonstrate the effective elimination of the residual elevation errors in the TDM DEM product through validation with the high-precision laser altimetry data. Examples of residual elevation errors in the original TDM DEM product were visualized in the elevation difference maps (Figure 6c and Figure 7c) while the effects of the applied corrections were shown in Figure 6f and Figure 7f. In this section, the elevation errors and their removal in the TDM DEM are analysed along several profiles located as in Figure 9a and Figure 10a and belonging to the experimental areas used in Sections 4.1 and 4.2. From the profiles in Figure 9 and Figure 10b to Figure 10e the erroneous elevations can be roughly divided into two patterns. Their influences on the effectiveness of the proposed method are evaluated qualitatively below.

In Figure 9, the profile can be divided into sub-segments with similar offsets ranging from tens to hundreds of meters. Also, in Figure 10d and Figure 10e, spatially-connected points along the profiles L3 and L4 deviate from the correct elevation values with a certain offset. These jumps are much larger than the elevation difference between TDM DEM and REMA mosaic and cannot be caused by the X-band microwave penetration depth (ranging from several meters to 10 m for high penetration conditions) (Rizzoli et al., 2017b) or temporal surface elevation changes (-3 m/a at HG between 2013 and 2016) (Rott et al., 2018). Besides, the clear boundary in the DEM hillshade map (Figure 9a) and the elevation jumps in the elevation profiles (Figure 9 and Figure 10b–Figure 10e) further confirm the existence of the residual elevation errors and exclude the influence of signal penetration and temporal elevation surface changes. This kind of local elevation offsets are typical elevation errors introduced from PU due to erroneous determination of phase ambiguity. The path propagation method described in Section 3.2 can automatically detect the local regions affected by elevation offsets and segment them into sub-regions with similar offset. The correction method proposed in Section 3.3 takes the offsets between TDM DEM and REMA mosaic at stable areas around the erroneous region into consideration, thus avoiding over-correcting the TDM DEM to the REMA mosaic. As a result, the corrected elevation profiles are continuous and smooth (black lines in Figure 9 and Figure 10b–Figure 10e) and the spatial details are well preserved even after eliminating the offset e.g. as along profile L4 (Figure 10e).

Unlike the local continuous region with similar elevation offsets, profile L1 (Figure 10b) shows the elevation inconsistency pattern when the erroneous region neither has a unified elevation offset like L3 (Figure 10d) and L4 (Figure 10e) nor can be segmented into sub-regions with similar offset and clear boundary as in Figure 9. The regional elevation offsets in Figure 10b are still related to phase unwrapping errors. However, the scene-based weighted average processing when generating the final TDM global DEM mosaic make it difficult to distinguish the original PU errors in the raw DEMs. In addition, the residual calibration errors may introduce near a vertical offset and a horizontal tilt (and shift), thus contributing to the elevation inconsistency in the mosaicked TDM DEM. Under these circumstances, the elevation correction depends on the detected erroneous regions through the path propagation algorithm and is more influenced by the REMA

DEM. Another particular case is shown for profile L2 (Figure 10c) where elevation anomalies at small horizontal spatial scale occur. L2 can be seen as a combination of different patterns where the proposed correction method can also work effectively by removing elevation offsets and noise.

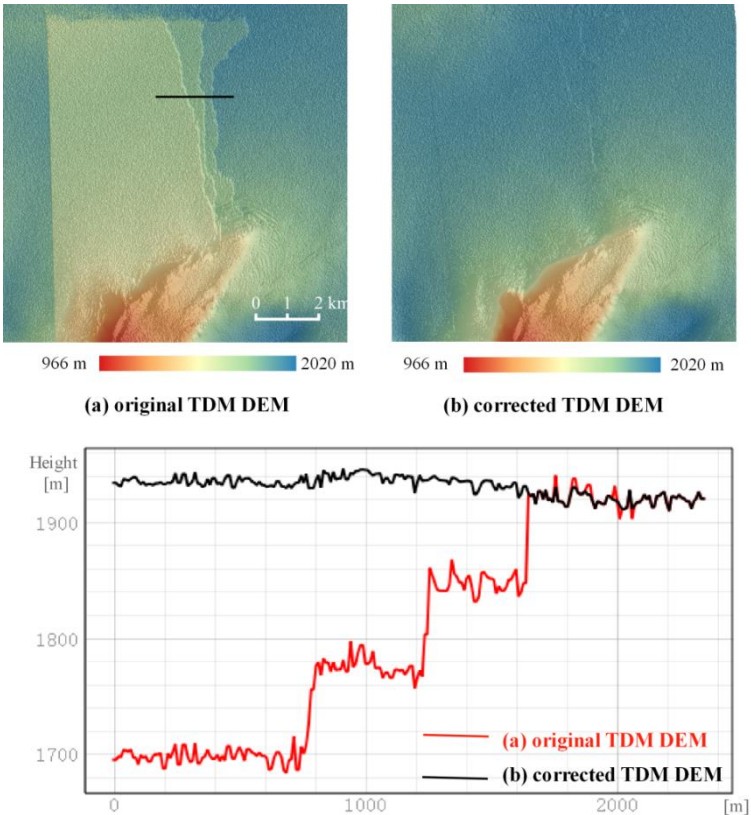

**Figure 9.** Elevation profiles along the black line extracted from (a) original and (b) corrected TDM DEM of local sample area.

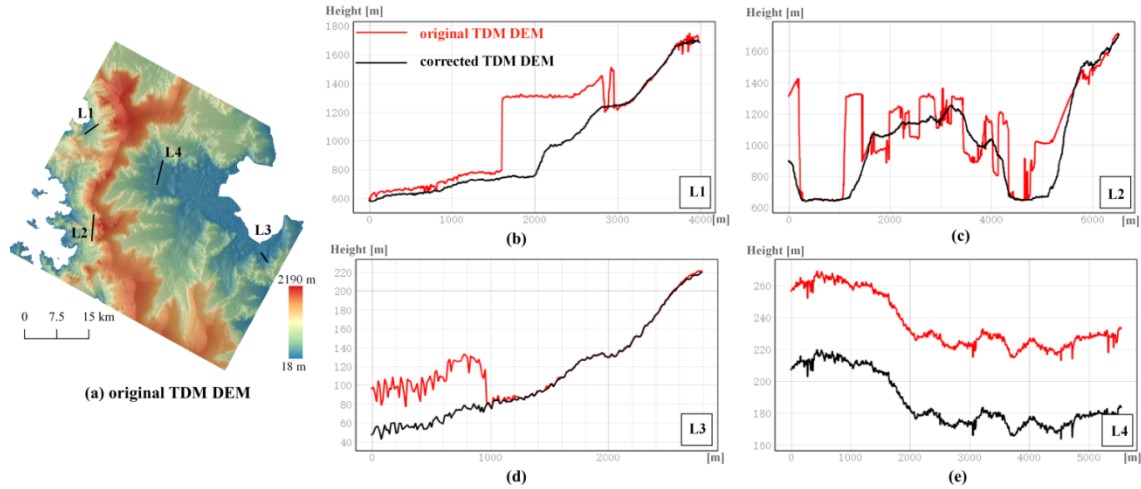

**Figure 10.** Four elevation profiles extracted along lines L1-L4 from original and corrected TDM DEM at HG area.

### 5.2 The importance of multi-scale elevation error correction strategy

Taking into consideration the various vertical scales of the residual elevation errors we have proposed the multi-scale elevation error correction method in Section 3.4. Here we discuss the necessity of the multi-scale elevation errors correction strategy as well as the validation methods.

    In our experiments, we applied corrections of elevation errors at three scales: large (> 45 m), medium (> 20 m) and small-scale (> 5 m). These three thresholds were empirically determined from the intermediate correction results. In the following
we explain in detail the process of multi-scale correction on the local and HG experimental areas. In the elevation difference maps of typical regions with PU errors such as in Figure 11a and Figure 12a, the elevation discrepancies have absolute values larger than 50 m. Then at the first correction, we set the threshold as 45 m to separate the background pixels. After this first iteration the remaining elevation discrepancies reduce to absolute errors larger than 20 m as visible in Figure 11b and Figure 12b. After applying the second correction (with 20 m threshold), residual elevation offsets are still visible in the
elevation difference maps (Figure 11c and Figure 12c). Therefore we set 5 m as the threshold for the small-scale correction. Besides using the thresholds for erroneous regions detection, the correction also depends on the fitted reference elevation surface defined by the stable points extracted from the buffer zone. If there are not enough stable points, the correction will not be applied. To differentiate these small residual DEM elevation errors from elevation changes due to natural processes like the penetration depth or temporal surface change in the final small-scale correction we take also the area of the merged
regions into account. Considering the penetration depth and the temporal surface change affect a relatively large area and are changing with transitional trend, only small regions (e.g. less than 100 pixels) with elevation jumps will be corrected during the third iteration. The magnitude of the elevation differences could be reduced gradually after each correction step as obvious from the decreasing elevation range in the difference maps with final improved results visible in Figure 11d and Figure 12d.

The corresponding elevation difference statistics after each correction when compared to LVIS 2015 and ATL06 2019 data are given in Table S2 and Table S3 in the supplementary for the local and HG area, respectively. MAE and RMSE are decreasing significantly after each iteration. The mean of the absolute correction elevation and the coverage percentage for the detected elevation discrepancies (Table 4) diminish obviously after each iteration of correction.

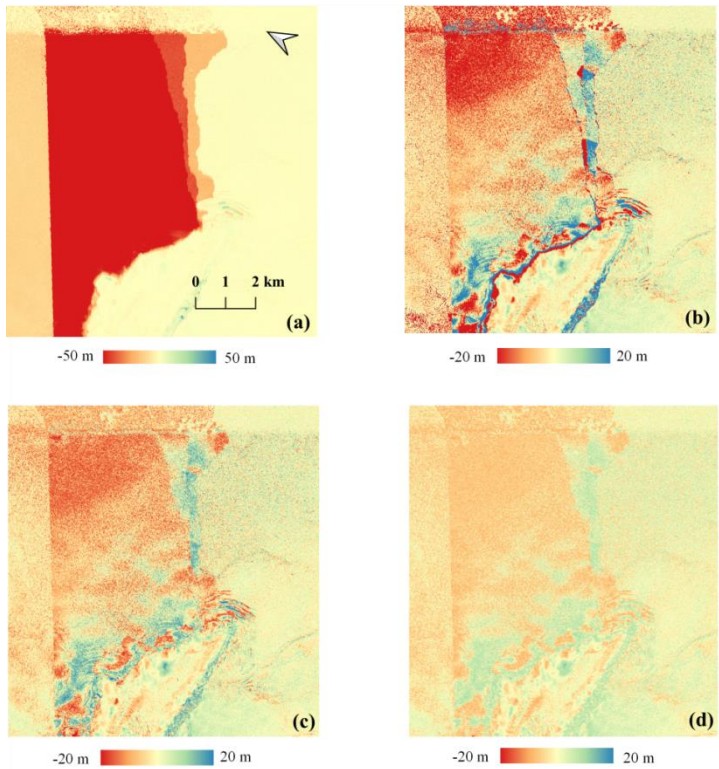


**Figure 11.** Elevation difference maps of the local area between (a) original TDM DEM, (b) TDM DEM after 1st, (c) TDM DEM after 2nd, (d) TDM DEM after 3rd multi-scale correction and REMA mosaic. Elevation difference: TDM DEM minus REMA mosaic.

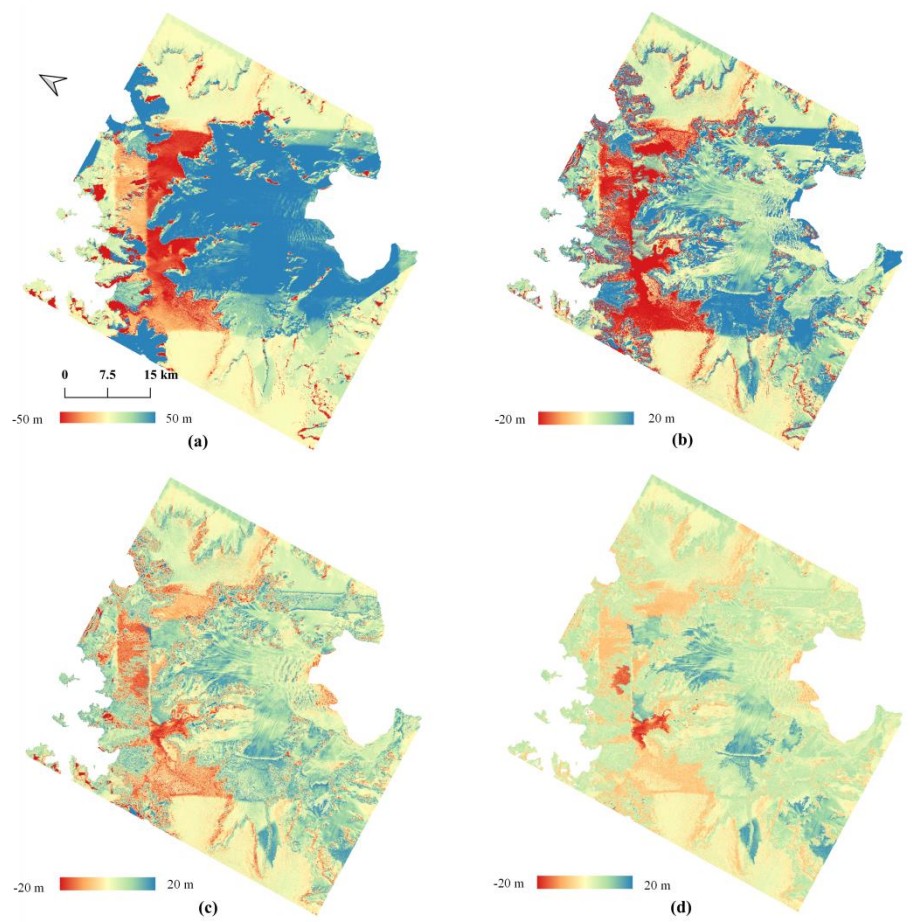

**Figure 12.** Elevation difference maps of HG area between (a) original TDM DEM, (b) TDM DEM after 1st correction, (c) TDM DEM after 2nd correction, (d) TDM DEM after 3rd correction and the REMA mosaic. Elevation difference: TDM DEM minus REMA mosaic.

**Table 4.** The mean absolute correction value and coverage percentage for the detected elevation discrepancies.

| | Experimental local area | | | Experimental HG area | | |
|---|---|---|---|---|---|---|
| | No.of pixels | Mean_ABS [m] | Percentage | No.of pixels | Mean_ABS [m] | Percentage |
| After 1st correction | | 162.2 | 48.27% | | 87.94 | 35.20% |
| After 2nd correction | 1,031,020 | 33.3 | 10.15% | 20,563,089 | 32.94 | 28.27% |
| After 3nd correction | | 12.55 | 18.55% | | 8.82 | 23.68% |

## 5.3 The influence of reference DEMs' spatial resolution and data voids

The proposed method relies on absolute vertical accuracy and spatial resolution of the reference DEM. REMA mosaic has

high absolute vertical accuracy and is generated from optical photogrammetry without PU errors (Howat et al., 2019), which is favorable for correcting elevation biases in TDM DEM. Ideally the reference DEM should have comparable spatial resolution with the DEM to be corrected like the 12-m TDM DEM and 8-m REMA mosaic. However, there are about 8% gaps of the landmass in the 8-m REMA mosaic based on our statistics. In our experiments, the 8-m REMA mosaic has been

resampled to the same spatial resolution of the TDM DEM of 12 m before the generation of the elevation map. The data voids of 8-m REMA mosaic were filled by the 100-m REMA mosaic whose voids have been filled by the 100-m edited ASTER GDEM (Howat et al., 2019). Therefore, the analysis of the influence of data voids on the proposed correction algorithm is actually to analyse the influence of the different spatial resolution between the original TDM DEM and the reference DEM.

To evaluate the influence of the different spatial resolution of the reference DEMs on the correction algorithm, we performed a contrast experiment here that we use the 8-m and 100-m REMA mosaic as reference DEMs for correcting the original TDM DEM at two sample areas as in Figure 11 and Figure 12. The corrected DEMs were evaluated by the altimetry datasets LVIS 2015 and ATL06 2019 as in Figure 13. At HG area the MAE is overall higher for the corrected TDM DEM with 100-m REMA mosaic compared to the 8-m REMA mosaic correction, while at the small local sample area the resulting

MAEs are similar. The MAEs are affected by the location of the laser points. We calculated a linear regression of MAE and slope combining all the validation data of the two experimental sites (Figure 14). The MAEs correspond to the corrected TDM DEM using 8-m REMA mosaic as ground reference given in Table 1 and Table 2. The median slopes were calculated from the corrected TDM DEM for the elevation range of each MAE value. A positive correlation between MAE and the terrain slope is observed, steep slopes being prone to large elevation errors in the DEM.

From perspective of the algorithm implementation, the elevation biases can be detected and corrected by the proposed algorithm as long as they can be identified from the elevation difference map with distinguishable boundaries. Theoretically, the influence of the spatial resolution between different datasets depends on the spatial size of the regions with elevation biases and whether these regions have complex topography or not. Therefore the difference in the resolution of the reference DEM datasets has slighter impact on the correction algorithm on flat ground than in areas with severe terrain undulations.


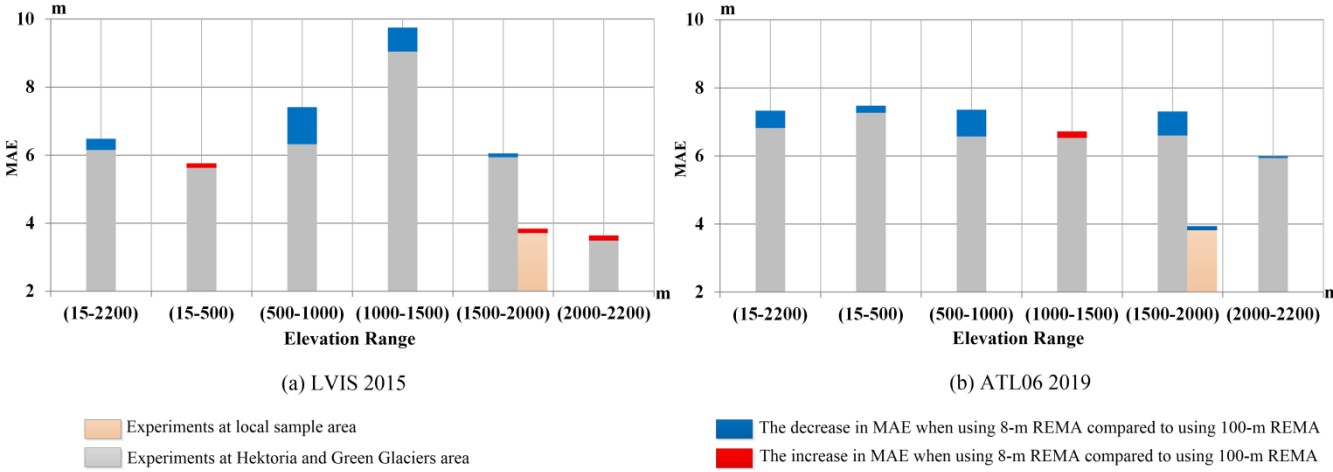

(a) LVIS 2015

(b) ATL06 2019

Experiments at local sample area

Experiments at Hektoria and Green Glaciers area

The decrease in MAE when using 8-m REMA compared to using 100-m REMA

The increase in MAE when using 8-m REMA compared to using 100-m REMA

**Figure 13.** Changes in MAE of the corrected TDM DEM when using 8-m and 100-m REMA mosaic as ground reference. Results were evaluated against (a) LVIS 2015 and (b) ATL06 2019 datasets.


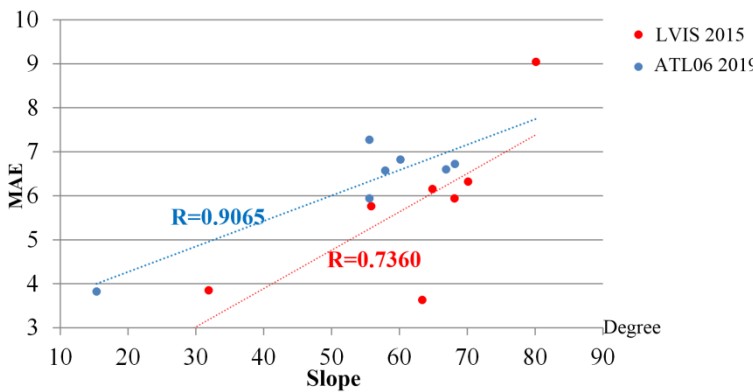

**Figure 14.** Linear regression between the MAE of the corrected TDM DEM using 8-m REMA mosaic as ground reference and the median slope calculated from the corrected TDM DEM at the elevation ranges as in Figure 13.

## 5.4 Potential applications of the corrected TDM DEM

The original TDM DEM and REMA mosaic have comparable absolute vertical accuracy according to our validation results in Table 3, but TDM DEM has better completeness (Figure S1), temporal consistency (Figure S2) and relative vertical accuracy based on the elevation error layers accompanying the DEM products (Figures S3 and S4). The residual systematic errors correction of the TDM DEM is minimally influenced by temporal or penetration differences between TDM DEM and REMA mosaic. The characteristics of an interferometric DEM are maintained and therefore the outcome is not a

hybrid DEM like it would be a gap-filled REMA mosaic. This may bring advantages of using the corrected TDM DEM for certain applications. It can be used in specific interferometric processing like topographic phase removal, PU corrections,

geocoding, single pass InSAR DEM generation and absolute phase calibration. Having a precise time stamp and a short acquisition time span, the TDM DEM 2013/2014 can be subtracted from other DEMs to derive surface elevation change and geodetic mass balance of AP glaciers over a time span of several years. The TanDEM-X change DEM (Lachaise et al., 2019) generated from data acquired between 2017 and 2019 could be one of the candidates. With 12 m pixel spacing the corrected and gapless TDM DEM has certain advantages compared to the former gapless reference DEMs covering AP like the 100-m edited ASTER GDEM (Cook et al., 2012), the 100-m REMA mosaic (Howat et al., 2019), the 90-m TanDEM-X PolarDEM (Wessel et al., 2021) for glaciological applications and morphological analysis requiring high spatial resolution. The corrected TDM DEM can also be used to fill the data voids in the 8-m REMA mosaic.

## 5.5 The comparison with DEM fusing method

To verify the effectiveness of the proposed algorithm in a comparative way, we fused the TDM DEM and REMA mosaic with weights determined by their random elevation errors as in Eq. (4). The difference maps between the fused DEMs and REMA mosaic over the experimental sites are shown in Figure 15. Compared with the difference map before and after the correction in Figure 11 and Figure 12, it is obvious that the weighted fusion cannot eliminate the elevation discrepancies in TDM DEM caused by PU errors because the HEM layer considered as random error used for weights determination is not capable to represent the PU errors. This contrast experiment well proves the effectiveness of the proposed algorithm in detecting and correcting the residual PU errors.

$$\text{H}_{fusion} = \text{TDM}\_H * \text{TDM}\_weight + \text{REMA}\_H * \text{REMA}\_weight \qquad (4)$$

Where $\text{TDM}\_weight = \dfrac{1/\text{TDM}\_error}{1/\text{TDM}\_error + 1/\text{REMA}\_error}$ , $\text{REMA}\_weight = 1 - \text{TDM}\_weight$ ; TDM_H and REMA_H represent the elevation values from TDM DEM and REMA mosaic.

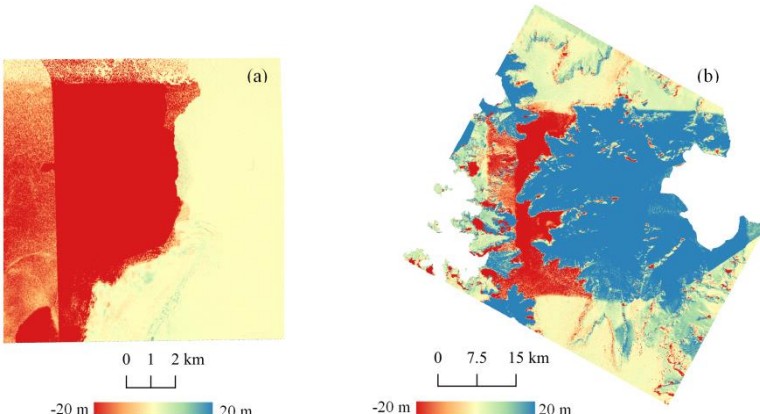

**Figure 15.** Elevation difference of the fused DEM minus REMA mosaic at (a) local sample and (b) HG area

**6 Conclusions**

In order to meet the high-resolution topography data demand of fine-scale glaciological research, we combined elevation
information provided by two up-to-date large scale high-resolution DEM products, the 12-m TanDEM-X DEM (TDM DEM)
and the 8-m REMA mosaic product, to generate a high-resolution precise, consistent and gapless DEM of the Antarctic
Peninsula (AP). Prior to the combination with REMA, the TDM DEM is characterized by good data consistency and few
data voids, but contains residual systematic elevation errors introduced by baseline calibration, geometric distortion and
phase unwrapping (PU). The REMA mosaic has in turn high absolute vertical accuracy (about 1 m) and absence of regional
outliers. Combining the advantages of TDM DEM and REMA mosaic, we identified the areas in the TDM DEM affected by
errors with a path propagation algorithm and developed the multi-scale method to automatically correct the elevation errors
in the TDM DEM. The effectiveness of the proposed method and the vertical accuracy of the resulting DEM were validated
by visual inspection and laser altimetry data. The main findings of our research are as follows:

1) The path propagation algorithm can effectively detect erroneous regions with similar elevation offsets caused by
remaining PU errors in TDM DEM and separate them from unaffected areas. By merging the adjacent homogeneous pixels
into one region and holding at background and heterogeneous pixels, the procedure allows a successful identification of
regions with different elevation offsets even with blurry boundaries.

2) The elevation offset compensation includes near the difference to REMA mosaic a fitted reference surface derived from
selected stable points in TDM DEM and thus preserves the reference elevation surface of the TDM DEM. The elevation
difference between TDM DEM and REMA mosaic caused by the penetration depth of the X-band radiation and temporal
surface change should be excluded from the elevation correction applied to TDM DEM. Buffer zones created around each
extracted erroneous region provide the above mentioned selected stable points, which in turn generate the compensation
elevation value.

3) The multi-scale method can comprehensively correct the TDM DEM by iteratively adjusting from large to small
elevation scale errors. The corrected TDM DEM is superior in data consistency and completeness to REMA mosaic.

In general, the DEM over AP resulting from the combination of TDM DEM and REMA mosaic maintains the
characteristics of the interferometric DEM and has an improved quality due to the correction of the residual elevation errors.
The absolute vertical accuracies of the corrected TDM DEM and the REMA mosaic validated against laser altimetry data
(Operation Ice Bridge LVIS and ICESat-2) are very similar. We therefore recommend to use the presented corrected DEM in
various glaciological applications requiring detailed gapless topography information. The precise time stamp (austral winter
2013/2014) is an advantage for direct comparisons with other DEMs and derivation of surface elevation changes. Besides,
DEM time series needed for the geodetic mass balance can be precisely vertically co-registered using our DEM as reference
surface. In interferometric SAR processing the presented DEM can support the modelling of the topographic phase when
separating this contribution from displacements and vertical deformation. Also drainage basin delineations of individual
glaciers rely on accurate DEMs. The proposed method can be extended to other areas of the Antarctic Ice Sheet where SAR

and optical DEMs are prone to errors like mountainous coastal regions or in the Transantarctic Mountains.

*Data availability*. The improved DEM dataset will be made available upon publication of the final version via the EOC Geoservice of the Earth Observation Center (EOC) of the German Aerospace Center (DLR) (https://geoservice.dlr.de/web/). In order to better support the use of the corrected TDM DEM, an additional bit mask layer is provided which marks the questionable areas in the REMA coverage where there is no REMA coverage and with high slopes (i.e. larger than 65 °).

*Acknowledgements.* The TanDEM-X data were made available by DLR through the projects DEM_GLAC1059 and XTI_GLAC7408. REMA data were downloaded from University of Minnesota, Polar Geospatial Center. The IceBridge LVIS and ICESat-2 laser altimeter data were obtained from the NASA Distributed Active Archive Center, US National Snow and Ice Data Center (NSIDC), Boulder, Colorado.

*Financial support.* This work was supported by the National Natural Science Foundation of China (Grant No. 41901413) and by the DLR project "Polar Monitor". Furthermore, the first author would like to thank the Sino-German (CSC-DAAD) Postdoc Scholarship Program for its scientific research funding.

*Competing interests*. The authors declare that they have no competing interests.

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
