# Peer review of "High-resolution topography of the Antarctic Peninsula combining TanDEM-X DEM and REMA mosaic"

_The Cryosphere, 2020_

## Referee Comment (RC1) · Anonymous Referee #1 · 20 Dec 2020

General comments: I have now gone through the manuscript more than twice. Generally speaking, this is a well-written manuscript with a thorough description of methods and analysis of results. Authors have used TDM DEM and REMA DEM of the AP region and improved the quality by combining them using propagation algorithm. Authors have demonstrated the improvement by comparing using laser altimetry data captured during two campaigns. Authors have demonstrated the improvement in terms of RMSE and clearly showed the improvement in iterative 3-steps of correction. My major criticisms are; (1) Authors have not explained the effect of using multi-temporal datasets captured during two different periods and later comparing them with laser altimetry campaign datasets captured in other periods. There is a significant temporal constraint

in merging these datasets- I suggest authors describing the effect of using such data and how much error it will introduce in their analysis. (2) Authors generally consider REMA as a ground reference DEM and improve the TDM DEM based on the values of REMA DEM. REMA is about 8m and then they used 100m coarse values where they have voids in the REMA DEM. Propagation algorithm works on two DEMs of slightly different spatial resolution; authors should explain the effect of different spatial resolutions of datasets on the algorithm. Put other words, could you resample your two DEMS on the same resolution and then run the algorithm to find out the performance? From result tables, I can see improvements varying in different steps of corrections and also for different elevation settings which are expected. However, the significance of final improvement has not been justified by authors. How authors can claim this improvement and not random noise? This is mainly because I can see instances in the result tables where improvement is around 2m. (3) My concern is why glaciologists would use the newly constructed improved TDM with accuracies still less than original REMA? REMA accuracies were reported less than 1m and TDM accuracies are reported around 10m. The only advantage I can see in merging is to fill data voids or gaps of REMA. From table 3, it is well demonstrated that there is no significant improvement (w.r.t REMA) in RMSE even after improving the TDM. The achievement of this study is to fill the data gaps in REMA using TDM. Put in other words, why reader can't call it as an improved REMS DEM or gapless REMA DEM as the basic foundation of the algorithm is the REMA and not the TDM? Authors must understand the data circularity created by the methodology and see that REMA was used as a reference to correct TDM values and then it is compared against the TDM and original REMA. In general, glaciologists will use this improved DEM if they find it more accurate than the REMA but this is not demonstrated. How if we simply patch up missing elevation values from REMA by TDM and smooth those gap areas? I suggest authors to suggesting future use of corrected TDM in glaciological applications. I encourage authors to describe this in the discussion section. (4) Authors have not demonstrated the viability of their methods w.r.t published methods of merging DEMs. This should be discussed in

the discussion section.

Section-wise comments are appended as follows:

Abstract: I have carefully read the abstract. It is generally well written, but it is somehow not attractive in the reader's perspective. Authors have failed to mention RMSE in absolute numbers rather they refer percentage. Between line 15-20, I encountered a very long statement which can be shortened. *To generate a consistent, gapless and high-resolution (12 m) topography product of the AP, we combine the TDM DEM and REMA mosaic by detecting and correcting the height errors in TDM DEM through a novel path propagation algorithm and multi-scale height error correction method based on the accurately calibrated REMA mosaic data. *. I would suggest authors to improve the abstract to make it more readable to readers and also boost it with quantitative results at the end.

Introduction: Simplify this: 2020). AP is a complex mountainous coastal glacier system and the mass balance of the outlet glaciers is affected by climate and oceanographic forcing and also by the subglacial and surrounding topography (Cook et al., 2012).

Good to see available DEMs of AP, mostly are Antarctic-wide. Table S1 provides a good overview but unfortunately, authors have missed a few regional attempts of making DEMs e.g. Fieber et al, 2018: https://doi.org/10.1016/j.rse.2017.10.042. Line 35-45, I would suggest authors revisit regional attempts of constructing DEMs of AP region.

Line 45: By analysing all these available DEMs, it can be noted that the DEMs of AP have always suffered from large elevation uncertainty, coarse resolution, wide data voids or incomplete data coverage, which are caused by the complex mountainous terrain and cloudy weather of AP. I think this a very generic statement which is applicable for most of the regions of the continent and restricted to only AP.

I see authors are using the term posting, are you referring to the spatial resolution?

Line 56: To obtain a consistent, gapless and precise DEM product at the high spatial

resolution of AP, we intend to create a high-resolution DEM of AP by combining the TDM DEM and REMA mosaic, the two up-to-date DEMs with similar posting. Authors should use comparable posting rather than a similar posting.

In general, the introduction section is not fully developed. It gives a feeling of missing information. For instance, authors should mention about the necessity of accurate and high-resolution DEM in the region and previous literature or applications of DEM used in the AP for various glaciological studies. This would provide a robust background on how accurate DEM can improve these existing studies. Authors mentioned about Cook et al. (2021) attempt of improving DEM but they ignore other efforts of combining multiple datasets to generate improved DEMs in Antarctica. To my knowledge, there are established attempts of developing DEMs in the Antarctic by combining two or more datasets- Authors should review those efforts in and then place their study at the end and explain how their effort is different than others.

Experimental area and data: Figure 1: Authors should mention elevation on the colour scale. And may consider naming a few landmark points in the figure to make it more readable. Somehow one yellow box is hidden behind the green coastline. You may consider changing the draping and make the yellow box above the green coastline layer so it is visible. Is the background RAMPv2 DEM or imagery? And you may also consider showing the high-resolution window showing sampling locations. Experimental data: This section is very well written, well done! Minor comment: use the term elevation and height consistently throughout the manuscript.

Methodology: Line 130-135: use the term ground reference and not the ground truth.

Figure 2: In the first section box, I cannot see x and y-axis numbers (Height difference against frequency graph). In section II, what are different shades of blue showing height error regions? Are you missing a colours scale here? I cannot see the text in blue in the Fitted reference surface model of section III. What is this blue line?. Authors should improve the caption of this figure describing the flow process briefly.

Figure 3: You may consider showing REMA DEM of the same region shown in (a)

Authors have mentioned of using empirical threshold but did not mention much about the process of defining the empirical threshold to execute propagation algorithm. I understood the method of correcting TDM DEM against REMA using propagation algorithm, but I am also concerned about pixel resolution difference between two datasets and then impact of this varying resolution on the algorithm. It is more evident when authors are using 100-m sampled data where REMA has data voids.

---

## Referee Comment (RC2) · Romain Hugonnet (Referee) · 10 Jan 2021

**1 General comments**

The paper is well-prepared and the authors make great effort to present their study rigorously. The text is generally well-written and the quality of Figures is good. The introduction accurately paints the context and limitations for Digital Elevation Models and related studies of the Antarctic Peninsula. The methodology presented for height correction is, to my knowledge, novel and its implementation is sound. The validation effort of the results with high-precision data is valuable. Finally, the resulting impacts

are significant and clearly highlighted in the conclusions.

However, I have several major concerns. In decreasing order of importance, those are:

1. the general logic when combining the TDM and REMA DEMs,

2. the relation to existing methodologies,

3. the limits of the validation with IceBridge and ICESat-2,

4. the statistical estimators used in the study.

Those are detailed below.

**2 Specific comments**

**2.1 Combining TDM and REMA DEMs**

With 2 DEMs covering approximately all the AP, one can ask himself: what is the best reference DEM to use? What I draw from the authors presentation is that they decided to correct TDM DEM with REMA DEM due to:
A) the short timespans of acquisitons of TanDEM-X (2013-2014) while REMA is based on WorldView acquisitions spanning a longer period of 2009-2017. This shorter timespan of TDM leads to less issues with glacier elevation change over time on the entire AP.
B) less data gaps in TanDEM-X compared to REMA.
These factors that motivated their choice are somewhat "implicit" along the flow of paper, except in the abstract. I think the authors should clearly state all their arguments at once at some point of the main text to make the entire reasoning behind their choice understandable (at the end of introduction? Or the start of methods?)

Additionally, I think that this choice is still subject to some discussion. And this for several reasons:

- firstly, I do not find the argument "less data gap in TDM DEM" completely valid. As the authors show, TDM DEM contains many height errors, or artefacts, that require correction. Those could be interpreted as data gaps as well. It is not clear how much surface area is affected, but the authors should have quantified this after application of their methodology. Many of REMA data gaps likely originate from photogrammetric blunders, very much like TDM height errors originate from interferometric ones. This would partly invalidate argument B) for selecting TDM.

- secondly, the REMA DEM is a mosaic based on WorldView 2 m and 8 m strips, freely available through the Polar Geospatial Center. I have not checked, but it is possible that most of the AP is covered by WorldView acquisition within a 2/3-year timespan of each other instead of suffering from the full 2009-2017 deviation. This would partly invalidate argument A) for selecting TDM.

- finally, the validation effort shows REMA DEM to consistenly have higher vertical precision than the TDM DEM (Table 1, 2, 3: value of 90

I understand that this choice is complex and that the methodology developed by the authors is already dependent on the type of blunders present in TDM DEM which might not be reapplicable to REMA with its own blunders and larger-scale data gaps. Therefore, although this choice is directionally important for the study, it certainly remains fully with the authors. A correction based on TDM DEM is undeniably valuable. In any case, the authors should provide:

- a Figure (maybe in the Supplementary) showing the surface area coverage over time of available REMA DEM strips for the AP. If many strips are closely available in time with good coverage of the AP (>80%), this really poses the question of correcting TDM blunders instead of correcting/gap-filling REMA.

- a Table with coverage statistics + mean vertical correction for identified blunders at each multi-scale step of the methodology somewhere in the main manuscript.

- a discussion on the influence of the 8% data gaps in REMA on the correction of TDM DEM should be provided. I acknowledge that this effect is unavoidable, yet adding a paragraph quantifying the possibly omitted TDM blunders in these 8% REMA data gaps would be useful (maybe based on the average surface area of blunders found on the rest of the AP?). Providing a map of the areas possibly affected (where there is no REMA coverage, and possibly high slopes/larger errors in the TDM Height Error Map?) would also be valuable for future users.

Optionally, the authors should consider using individual strips instead of the REMA mosaic product for height error correction. This additional effort could significantly reduce temporal biases related to glacier elevation change by selecting strips closest to 2013-2014 when available (e.g. biases shown on Figure 7f). The authors would however need to co-register those strips individually to TanDEM-X before using them for corrections.

**2.2 Relation to existing methodologies**

The paper surprisingly lacks references to previous DEM correction/fusion methodologies. This is true both for the introduction (supposed to explain the context of existing methods and why a new one is necessary), for the methods (supposed to reference/compare to existing methods, if applicable) and discussion (based on the results, what are the benefits of using this specific method compared to others? qualitatively at least. quantitatively would be even better e.g. by comparing with other methods locally).

Many studies have looked at merging DEMs, removing data gaps and improving general DEM quality, for example: Reuter et al. (2007), Papasaika et al. (2009)

(full thesis here: https://ethz.ch/content/dam/ethz/special-interest/baug/igp/igp-dam/documents/PhD_Theses/109.pdf), Yamazaki et al. (2017), etc.

The authors should:

- provide a scientific context referencing existing methods and justifying a new methodology,

- identify and cite possible similar existing methodologies, if applicable.

**2.3  Validation with ICESat-2 and IceBridge**

Seasonal and temporal biases of validation exist but are omitted in the validation methods and its discussion. Those should be quantified and discussed.

For temporal biases, the authors could use low-resolution, large-scale elevation change maps (Smith et al. (2020)) to partition their validation data over the AP. Binning the validation points by category of expected elevation change during the period (e.g., near stable; <0.2 m yr-1, small elevation change >0.2 and <0.5, strong elevation change >0.5) could provide improved statistics to evaluate the results through the validation effort. The impact of seasonal biases in elevation changes between the two validation datasets should also be discussed using known estimates of seasonal cycle in the AP.

**2.4  Statistical estimators**

Along the study, the authors provide Tables of the same format to quantify the improvement brought out by their methodology (Table 1, 2, 3 S2, S3). They use the mean of elevation difference to the validation data, the root mean square of differences and the 90% quantile of the distribution of elevation differences.

Currently, the mean does not bring much information and is hard to interpret due to temporal biases (preceding comments), but also because this statistical estimator is not very robust to outliers. I suggest using the median, as well as binning by category of expected elevation change during the period (preceding comments). Possibly, a Table showing median residuals normalized by the time difference between the validation dataset and the TanDEM-X date (1-2 years for LVIS 2015, 5-6 years for ATL06 2019) would allow for a better comparison between the biases identified in the two datasets.

The RMSE is generally a good metric, it is however overly sensitive to outliers which is exactly what the TDM DEM is here suffering from. Using this estimator might "oversell" the improvement in the results, especially to the reader unfamiliar with these effects. Consider:

- splitting your statistics by category of initial height differences, or showing the statistics independently for the corrected regions (before and after) and the untouched ones (once). I feel the second choice would be preferable.

- using the Mean Average Error (MAE), less sensitive to outliers.

For the 90% quantile: I see this is currently the raw quantile of the distribution (Table 1, line 3, the value is negative). I imagine that the authors want to show a measure of distribution spread (estimate of elevation precision). For this, consider either:

- taking the 90% quantile of the absolute value of the elevation difference.

- calculating the half-width between the 5% and 95% quantiles of the distribution.

**3 Technical corrections**

**General on the text:**
Unless the authors can justify a specific reason, I advocate for the two following changes along the manuscript:

1. Pick "elevation" or "height", don't use both interchangeably. Example in the caption of Table 2: Height differences calculated as DEM elevation minus laser height. This is confusing. I suggest using "elevation" everywhere, as it is the term most commonly used (Digital Elevation Model, Reference Elevation Model of Antarctica, etc).

2. The use of "height error" seems questionable to me. "Error" by itself is not precise enough as it can refer to random errors, (i.e. uncertainties) or refer to systematic errors (i.e. biases). Here, the artefacts in the TDM DEM and the related correction methods developed by the authors fit clearly in the box of systematic errors. Thus, it seems to me that it would be clearer to use "elevation bias" instead of "height error" along the text.

**Text line by line:**
75-96: Mention % value of data gaps in TDX
81: high-precision
88: stereophotogrammetry
89: such as the Advanced Spaceborne...
98-112: Please specify the REMA release used (is it r1 or r1.1?)
114-123: Mention exact acquisition date for the LVIS data (season)
136: "the" buffer zone: This is the first mention of such a zone. Change to "a buffer zone" and refer to the related section.
140-178: All this information is not really specific to the paper Methods: shorten +

optionally, move to "Data"?
204: To my knowledge, "path propagation algorithm" is not a nomenclature commonly used for this kind of method. This type of "flood-fill" method (https://en.wikipedia.org/wiki/Flood_fill) for region extraction is generally called "region extraction", "blob extraction" or most generally connected-component labeling (CCL) (https://en.wikipedia.org/wiki/Connected-component_labeling). Please adapt the nomenclature, and cite a reference for the algorithm used if applicable, and also possibly its relation to an existing computing package/parallel implementation.
252: increases
253: increases
254: high-precision
255: multi-scale (for consistency)
258: "Their spatial extent increased from... to..."
264: Figure 6b
271: Why use the "geographically closest point" in the DEM instead of a bilinear interpolation to the center of the LVIS/ATL06 point? With the TDM DEM at a posting of 12 m, the potential 6 m horizontal bias using a "nearest neighbour" approach from the center of the point can lead to a 3 m vertical error on a 25° slope (50% slope), and higher for larger slopes. This procedure might be deteriorating the quality of the validation effort, consider switching to bilinear interpolation of the raster data.
321: ATL06
333: elimination of the residual height errors
341: Refer to Figure 10 before mentioning profiles L3/L4
371: The vertical scales should be specified sooner, in the Methods section maybe?

**On the Figures and Tables:**
Fig. 2:
Axis labels of histogram are too small in I. Text of schematics is squeezed vertically in III.

Fig. 5:
Specify if this is a schematic was created for demonstration purposes, or from real AP data at a specific transect.
Fig. 6, 7:
Add glacier outlines.

**4   References for the review**

Yamazaki, D. et al. A high-accuracy map of global terrain elevations: Accurate Global Terrain Elevation map. Geophys. Res. Lett. 44, 5844–5853 (2017)

Papasaika, H., Poli, D. Baltsavias, E. Fusion of Digital Elevation Models from Various Data Sources. in 2009 International Conference on Advanced Geographic Information Systems Web Services 117–122 (2009).

Reuter, H. I., Nelson, A. Jarvis, A. An evaluation of void‐filling interpolation methods for SRTM data. Int. J. Geogr. Inf. Sci. 21, 983–1008 (2007)

Smith, B. et al. Pervasive ice sheet mass loss reflects competing ocean and atmosphere processes. Science 368, 1239–1242 (2020)

---

## Referee Comment (RC3) · Romain Hugonnet (Referee) · 10 Jan 2021

A LaTeX error has cut off one of my point in 2.1. Here it is in full:

- finally, the validation effort shows REMA DEM to consistenly have higher vertical precision than the TDM DEM (Table 1, 2, 3: value of 90% quantile for REMA vs TDM). As a potential user, I would highly prefer to have a "consistent" AP DEM with most of its coverage based on the dataset with the highest vertical precision, which here is REMA.

Best, Romain

---

## Author Comment (AC2) · 28 Feb 2021

**Interactive comment on "High-resolution topography of the Antarctic Peninsula combining TanDEM-X DEM and REMA mosaic"**

*Authors:* Yuting Dong, Ji Zhao, Dana Floricioiu, Lukas Krieger, Thomas Fritz, Michael Eineder

The Cryosphere Discuss., https://tc.copernicus.org/preprints/tc-2020-323/

Referee comments are shown in *black*, our response in blue. Line numbers refer to the manuscript version (pdf) of 4 December 2020.

**Authors' response to Dr. Romain Hugonnet**

*1 General comments*

*The paper is well-prepared and the authors make great effort to present their study rigorously. The text is generally well-written and the quality of Figures is good. The introduction accurately paints the context and limitations for Digital Elevation Models and related studies of the Antarctic Peninsula. The methodology presented for height correction is, to my knowledge, novel and its implementation is sound. The validation effort of the results with high-precision data is valuable. Finally, the resulting impacts are significant and clearly highlighted in the conclusions.*

*However, I have several major concerns. In decreasing order of importance, those are:*

*1. the general logic when combining the TDM and REMA DEMs,*

*2. the relation to existing methodologies,*

*3. the limits of the validation with IceBridge and ICESat-2,*

*4. the statistical estimators used in the study.*

*Those are detailed below.*

**Response:** Firstly, we want to thank Dr. Romain Hugonnet for the time and effort put in this detailed and thorough review. We carefully evaluated all comments and suggestions, which are extremely valuable in improving the paper. As for the four major concerns raised by his review, point-to-point responses are given in the following. For better clarification, we add figs. R1-R4 in this response letter, and all the figures as well as the corresponding clarification will be added into the revised manuscript or the revised supplementary material.

*2 Specific comments*

*2.1 Combining TDM and REMA DEMs*

*With 2 DEMs covering approximately all the AP, one can ask himself: what is the best reference DEM to use? What I draw from the authors presentation is that they decided to correct TDM DEM with REMA DEM due to:*

*A) the short timespans of acquisitons of TanDEM-X (2013-2014) while REMA is based on WorldView acquisitions spanning a longer period of 2009-2017. This shorter timespan of TDM leads to less issues with glacier elevation change over time on the entire AP.*

*B) less data gaps in TanDEM-X compared to REMA.*

*These factors that motivated their choice are somewhat "implicit" along the flow of paper, except in the abstract. I think the authors should clearly state all their arguments at once at*

*some point of the main text to make the entire reasoning behind their choice understandable (at the end of introduction? Or the start of methods?)*

**Response:** Thanks for the comments. We will make improvement to the manuscript from the two aspects: 1) add a comparison in terms of absolute vertical accuracy, data voids, temporal consistency and random errors (or relative vertical accuracy) between TDM DEM and REMA mosaic at AP in the discussion section to better support the logic to generate a corrected TDM DEM for glaciology applications; 2) add more clarification in the introduction about the logic based on the comparison between the two DEM products.

*Additionally, I think that this choice is still subject to some discussion. And this for several reasons:*
*• firstly, I do not find the argument "less data gap in TDM DEM" completely valid.*
*As the authors show, TDM DEM contains many height errors, or artefacts, that require correction. Those could be interpreted as data gaps as well. It is not clear how much surface area is affected, but the authors should have quantified this after application of their methodology. Many of REMA data gaps likely originate from photogrammetric blunders, very much like TDM height errors originate from interferometric ones. This would partly invalidate argument B) for selecting TDM.*

**Response:** The statistics of data voids in TDM DEM and REMA mosaic at AP between 63°S and 70°S are about 0.85% and 8%, respectively, within the ADD coastline based on our counts. The data voids are counted from the null value defined in the original data products, which are shown as white regions in Fig. R1. The elevation errors in TDM DEM to be corrected in our study are mainly caused by phase unwrapping errors which belong to systematic errors. Regions with phase unwrapping errors have abrupt elevation offsets to their neighboring areas due to the incorrect determination of height ambiguities of the wrapped phase. These regions cannot be viewed as blunders because the elevation information is effective as long as the offsets are compensated. Taking the REMA mosaic with high absolute vertical accuracy as ground reference, regions with elevation discrepancies are identified and corrected with the proposed algorithm. In the revised manuscript, regions being corrected or kept unchanged in TDM DEM will be quantified.

[Figure]

Fig. R1 REMA mosaic covering AP and the location of three sample areas. Right panel: detailed comparison of the REMA (left column) and TDM (right column) DEMs in the sample areas.

*• secondly, the REMA DEM is a mosaic based on WorldView 2 m and 8 m strips, freely available through the Polar Geospatial Center. I have not checked, but it is possible that most of the AP is covered by WorldView acquisition within a 2/3- year timespan of each other instead of suffering from the full 2009-2017 deviation. This would partly invalidate argument A) for selecting TDM.*

**Response:** In our study we are using the REMA mosaic tiles which are generated by the quality-controlled REMA strip DEMs. The specific acquisition time of REMA mosaic covering AP is shown in Fig. R2a and Fig R2b in year and month, respectively. So it is definitely longer than 2/3 years.

In the revised manuscript, we will mention more precisely the time span to generate REMA mosaic at AP as 2011-2017 instead of 2009-2017 and we will add Fig. R2 to the supplementary material.

[Figure]

Fig. R2 Acquisition time of REMA DEM mosaic covering AP.

• *finally, the validation effort shows REMA DEM to consistently have higher vertical precision than the TDM DEM (Table 1, 2, 3: value of 90% quantile for REMA vs TDM). As a potential user, I would highly prefer to have a "consistent "AP DEM with most of its coverage based on the dataset with the highest vertical precision, which here is REMA.*

**Response:** Based on the validation result in Table 1-3, we would like to say that the TDM DEM has comparable absolute vertical accuracy with the REMA mosaic. Because the REMA mosaic has more data voids than TDM DEM, the number of points used for validation of TDM DEM is much larger than of REMA mosaic. For example, there are 31,764,790 and 33,246,648 points from LVIS 2015 datasets used for validating the accuracy of REMA mosaic and TDM DEM, respectively. The discrepancies in the numbers of verification points partly account for the differences in the statistics. Therefore, we will extract the intersection of the points from REMA mosaic and TDM DEM for validation in the revised manuscript. We will improve the method and the statistical estimators in the revised manuscript. Discussion of validation results in the revised manuscript will be improved based on the new results.

In the revised manuscript, we will add a discussion section about comparison between REMA mosaic and TDM DEM in terms of absolute vertical accuracy, data voids, temporal consistency and random elevation errors. The former three points have been explained in the submitted manuscript and will be clarified and improved in the revised manuscript. The fourth point (about random elevation errors) will be added to the revised manuscript. Based on the elevation errors maps accompanying the DEM products, we found that TDM DEM has smaller random errors and better theoretical relative height accuracy than REMA mosaic. In the elevation error map of REMA DEM in Fig. R3a, the error value at each pixel is the

standard error from the residuals of the registration to altimetry data (Howat et al., 2019). Since each tile used for REMA mosaic generation has removed outliers and systematic errors with the preprocessing, the error value at each pixel provides an estimate of the DEM's random elevation errors. The Height Error Map (HEM) values of TDM DEM in Fig. R3b represent the corresponding height error in form of the standard deviation for each DEM pixel (Wessel, 2016). The TDM error estimates are exact and reproducible derived from rigorous mathematically correct steps (Wessel, 2016) verified in several papers (Rizzoli et al., 2012;Rizzoli et al., 2017). Fig. R4 shows the histograms of the random elevation errors of the REMA mosaic and TDM DEM covering AP. Comparing Fig. R3a and R4a to Fig. R3b and R4b, it can be seen that the TDM DEM covering AP has random elevation errors at lower level and thus better theoretical relative elevation accuracy than the REMA mosaic.

[Figure]

Fig. R3 Random elevation errors of (a) REMA mosaic and (b) TDM DEM covering AP.

[Figure]

Fig. R4 Histograms of random elevation errors of (a) REMA mosaic and (b) TDM DEM covering AP. Median value and 90% quantile of the errors (90%LE) are marked in red in the histograms.

*I understand that this choice is complex and that the methodology developed by the authors is already dependent on the type of blunders present in TDM DEM which might not be reapplicable to REMA with its own blunders and larger-scale data gaps. Therefore, although this choice is directionally important for the study, it certainly remains fully with the authors. A correction based on TDM DEM is undeniably valuable. In any case, the authors should provide:*
*• a Figure (maybe in the Supplementary) showing the surface area coverage over time of available REMA DEM strips for the AP. If many strips are closely available in time with good coverage of the AP (>80%), this really poses the question of correcting TDM blunders instead of correcting/gap-filling REMA.*

**Response:** Fig. R1 and R2 will be added into the revised manuscript or supplementary material.

*• a Table with coverage statistics + mean vertical correction for identified blunders at each multi-scale step of the methodology somewhere in the main manuscript.*

**Response:** A table with coverage statistics and mean vertical correction at each multi-scale step will be added in the revised manuscript.

*• a discussion on the influence of the 8% data gaps in REMA on the correction of TDM DEM should be provided. I acknowledge that this effect is unavoidable, yet adding a paragraph quantifying the possibly omitted TDM blunders in these 8% REMA data gaps would be useful (maybe based on the average surface area of blunders found on the rest of the AP?). Providing a map of the areas possibly affected (where there is no REMA coverage, and possibly high slopes/larger errors in the TDM Height Error Map?) would also be valuable for future users.*

**Response:** In addition to the 8 m tiles, the REMA mosaic provides reduced-resolution resampled version at 100 m resolution. The reduced-resolution dataset has an alternate filled version in which the data voids in REMA mosaic are filled with 100-m ASTER GDEM (Howat et al., 2019;Cook et al., 2012). In our study, we resampled the 100-m filled REMA mosaic to fill in the data voids of the 8-m REMA mosaic.

The proposed algorithm runs on the elevation difference map generated from TDM DEM minus REMA mosaic. In our experiments, the gapless 8-m REMA mosaic with data voids filled with 100-m REMA mosaic has negligible effect on the proposed elevation biases detection and correction algorithm through visual inspection. The examples shown in Fig. 6 and Fig. 7 of the submitted manuscript illustrate that there are data voids in REMA mosaic (marked in white) which do not affect the correction process. The reason is that REMA mosaic was not used to correct the TDM elevation point by point, but to provide a reference elevation to correct the TDM elevation biases region by region, which is determined by the characteristics of the phase unwrapping errors.

Ideally the reference DEM should have comparable spatial resolution with the DEM to be corrected like the 12-m TDM DEM and 8-m REMA mosaic. The influence of the spatial resolution differences between different datasets depends on the spatial size of the regions affected by elevation biases and whether these regions cover areas with complex topography. In a word, as long as the biases can be deduced from the elevation difference map with distinguishable boundaries, they can be detected and corrected by the proposed algorithms. In the revised manuscript, analysis about the effects of spatial resolution difference between DEM datasets will be added.

*Optionally, the authors should consider using individual strips instead of the REMA mosaic product for height error correction. This additional effort could significantly reduce temporal biases related to glacier elevation change by selecting strips closest to 2013-2014 when available (e.g. biases shown on Figure 7f). The authors would however need to co-register those strips individually to TanDEM-X before using them for corrections.*

**Response:** Thanks for the very constructive advice. Theoretically it is a good idea for correcting the residual elevation errors in TDM DEM with REMA tiles closest to 2013-2014, which will minimize the temporal changes between TDM DEM and REMA DEM. However, it will be much more work to do to select the right REMA tiles with high data quality and calibrate these tiles to TDM DEM or altimetry data. The REMA mosaic DEM tiles have already went through the quality-control process through visual inspection and manual

correction to remove erroneous regions, as well as accurately calibrated to the laser or radar altimetry data (Howat et al., 2019). Thus we believe that REMA mosaic has high absolute vertical accuracy and is suitable as ground reference. Furthermore, the proposed correction algorithm has taken the elevation differences between REMA mosaic and TDM DEM caused by temporal surface changes into consideration from the following two aspects. First, the phase unwrapping (PU) errors have distinguishable characteristics from the temporal elevation change. Specifically speaking, the elevation errors in TDM DEM caused by the PU errors are characterized by local elevation discrepancies with abrupt elevation jumps at the boundary, while the temporal changes in elevation or penetration depth are transitional changes with a certain trend. Hence, the proposed path propagation algorithm is based on the characteristic of the PU errors to automatically detect the elevation jumps at the boundaries of the erroneous regions. Secondly, to eliminate the influence of the possible temporal elevation changes between the TDM DEM and REMA mosaic, we do not simply correct the TDM DEM to the reference elevation surface of REMA mosaic directly. Instead, we create a buffer zone around each region which has to be corrected. Stable points whose elevation differences with REMA mosaic are less than a given threshold value are extracted from the buffer zone. The average surface elevation fitted from these selected stable points is used as a reference surface for the elevation offset correction as in Fig. 5 in the submitted manuscript.

*2.2 Relation to existing methodologies*
*The paper surprisingly lacks references to previous DEM correction/fusion methodologies. This is true both for the introduction (supposed to explain the context of existing methods and why a new one is necessary), for the methods (supposed to reference/compare to existing methods, if applicable) and discussion (based on the results, what are the benefits of using this specific method compared to others? Qualitatively at least. quantitatively would be even better e.g. by comparing with other methods locally).*
*Many studies have looked at merging DEMs, removing data gaps and improving general DEM quality, for example: Reuter et al. (2007), Papasaika et al. (2009)*
*(full thesis here:* [https://ethz.ch/content/dam/ethz/special-interest/baug/igp/igp-dam/](https://ethz.ch/content/dam/ethz/special-interest/baug/igp/igp-dam/) *documents/PhD_Theses/109.pdf), Yamazaki et al. (2017), etc.*
*The authors should:*
*• provide a scientific context referencing existing methods and justifying a new methodology,*
*• identify and cite possible similar existing methodologies, if applicable.*

**Response:** Thanks for the comments and suggestions. In our work, an automatic algorithm to detect and correct the residual elevation biases exiting in the non-edited TDM DEM was proposed. Different from the general DEM fusion methods to incorporate the elevation information from different DEMs equally or by weights (Papasaika et al., 2009), the proposed algorithm can effectively correct the residual systematic errors in TDM DEM. REMA mosaic is used not to correct the TDM elevation point by point, but to provide reference elevations to correct the TDM elevation biases region by region, which are determined by the characteristics of the phase unwrapping errors. Therefore this proposed method maintains the characteristics of an InSAR generated DEM and is minimally influenced by temporal or penetration differences between TDM DEM and REMA mosaic.

The references and comparisons to the existing relevant algorithms will be summarized in the introduction section and discussed specifically in the discussion section in the revised manuscript. The recommended literature will be cited.

*2.3 Validation with ICESat-2 and IceBridge*
*Seasonal and temporal biases of validation exist but are omitted in the validation methods and its discussion. Those should be quantified and discussed.*
*For temporal biases, the authors could use low-resolution, large-scale elevation change maps (Smith et al. (2020)) to partition their validation data over the AP. Binning the validation points by category of expected elevation change during the period (e.g., near stable; <0.2 m yr-1, small elevation change >0.2 and <0.5, strong elevation change >0.5) could provide improved statistics to evaluate the results through the validation effort. The impact of seasonal biases in elevation changes between the two validation datasets should also be discussed using known estimates of seasonal cycle in the AP.*

**Response:** Thanks for the very helpful suggestion to incorporating the low-resolution, large-scale elevation change maps (Smith et al., 2020) for temporal biases analysis. To show the vertical accuracy of different elevation intervals, we will partition the laser points based on the elevation ranges in the revised manuscript. The annual surface elevation change will be converted to elevation change by multiplying by the acquisition timespan between the DEMs and laser altimetry points. Then the temporal elevation change will be compensated from the elevation difference between the DEMs and laser points before calculating the statistics.
As for the seasonal biases, unfortunately we have not found available seasonal changes products to compensate the seasonal changes of surface elevation. Furthermore, for REMA DEM validation in (Howat et al., 2019), laser altimetry data collected within 18 months of the REMA strip acquisition date or mosaic date stamp were selected. For TDM DEM validation in (Rizzoli et al., 2017), ICESat points were selected. In a word, the seasonal elevation changes were not taken into consideration for neither REMA DEM nor TDM DEM vertical accuracy validation with altimetry data (Rizzoli et al., 2017;Howat et al., 2019). Therefore, we ignore the seasonal biases in our validation with laser altimetry data.

*2.4 Statistical estimators*
*Along the study, the authors provide Tables of the same format to quantify the improvement brought out by their methodology (Table 1, 2, 3 S2, S3). They use the mean of elevation difference to the validation data, the root mean square of differences and the 90% quantile of the distribution of elevation differences.*
*Currently, the mean does not bring much information and is hard to interpret due to temporal biases (preceding comments), but also because this statistical estimator is not very robust to outliers. I suggest using the median, as well as binning by category of expected elevation change during the period (preceding comments). Possibly, a Table showing median residuals normalized by the time difference between the validation dataset and the TanDEM-X date (1-2 years for LVIS 2015, 5-6 years for ATL06 2019) would allow for a better comparison between the biases identified in the two datasets.*

**Response:** Thanks for the advice. In replacement of mean value, the median of the elevation difference will be added in the revised manuscript.

*The RMSE is generally a good metric, it is however overly sensitive to outliers which is exactly what the TDM DEM is here suffering from. Using this estimator might "oversell" the improvement in the results, especially to the reader unfamiliar with these effects.*
*Consider:*
*• splitting your statistics by category of initial height differences, or showing the statistics independently for the corrected regions (before and after) and the untouched ones (once). I feel the second choice would be preferable.*

**Response:** In the revised manuscript, the statistics of the corrected regions (before and after) and uncorrected ones (once) will be calculated.

*• using the Mean Average Error (MAE), less sensitive to outliers.*

**Response:** Mean Average Error (MAE) will be adopted as one of the statistical estimators in the revised manuscript.

*For the 90% quantile: I see this is currently the raw quantile of the distribution (Table 1, line 3, the value is negative). I imagine that the authors want to show a measure of distribution spread (estimate of elevation precision). For this, consider either:*
*• taking the 90% quantile of the absolute value of the elevation difference.*
*• calculating the half-width between the 5% and 95% quantiles of the distribution.*

**Response:** We will calculate the 90% quantile of the absolute value of the elevation difference in the revised manuscript.

*3 Technical corrections*
*General on the text:*
*Unless the authors can justify a specific reason, I advocate for the two following changes along the manuscript:*
*1. Pick "elevation" or "height", don't use both interchangeably. Example in the caption of Table 2: Height differences calculated as DEM elevation minus laser height. This is confusing. I suggest using "elevation" everywhere, as it is the term most commonly used (Digital Elevation Model, Reference Elevation Model of Antarctica, etc).*

**Response:** In the revised manuscript, the "height" will be changed into "elevation" for unification.

*2. The use of "height error" seems questionable to me. "Error" by itself is not precise enough as it can refer to random errors, (i.e. uncertainties) or refer to systematic errors (i.e. biases). Here, the artefacts in the TDM DEM and the related correction methods developed by the authors fit clearly in the box of systematic errors.*

*Thus, it seems to me that it would be clearer to use "elevation bias" instead of "height error" along the text.*

**Response:** "Height error" will be specified as random errors, (i.e. uncertainties) or systematic errors (i.e. biases) in the revised manuscript.

***Text line by line:***
*75-96: Mention % value of data gaps in TDX*
**Response:** corrected.

*81: high-precision*
**Response:** corrected.

*88: stereophotogrammetry*
**Response:** corrected.

*89: such as the Advanced Spaceborne...*
**Response:** corrected.

*98-112: Please specify the REMA release used (is it r1 or r1.1?)*

**Response:** It is the REMA mosaic DEM r1.1 used which will be specified in the revised manuscript.

*114-123: Mention exact acquisition date for the LVIS data (season)*

**Response:** The acquisition data of the LVIS data were during September and October of 2015, which will be added to the revised manuscript.

*136: "the" buffer zone: This is the first mention of such a zone. Change to "a buffer zone" and refer to the related section.*
**Response:** corrected.

*140-178: All this information is not really specific to the paper Methods: shorten +optionally, move to "Data"?*

**Response:** I believe the reviewer mentions the TDM DEM errors analysis section 3.1. Section 3.1 explains that the remaining elevation errors in TDM DEM causing large inconsistencies are mainly introduced by the systematic elevation errors especially the phase unwrapping errors. The proposed method is adapted to the characteristics of the residual systematic errors in TDM DEM. Therefore, it is preferable for us to maintain section 3.1 in the revised manuscript.

*204: To my knowledge, "path propagation algorithm" is not a nomenclature commonly used*

*for this kind of method. This type of "flood-fill" method (https://en.wikipedia.org/wiki/Flood_fill) for region extraction is generally called "region extraction", "blob extraction" or most generally connected-component labeling (CCL) (https://en.wikipedia.org/wiki/Connected-component_labeling). Please adapt the nomenclature, and cite a reference for the algorithm used if applicable, and also possibly its relation to an existing computing package/parallel implementation.*

**Response:** Thanks for your valuable comment. Indeed, as you said, the nomenclature "path propagation" used in the paper is very similar to the generally connected-component labeling, which is used to detect erroneous areas based on the elevation difference. As we all know, connected-component labeling is generally used to detect connected regions in binary digital images (Shapiro, 1996) (https://en.wikipedia.org/wiki/Connected-component_labeling). However, our input data for the detection algorithm in the paper is elevation difference image, whose value is from negative several thousand to positive several thousand instead of 0 and 1. For the "flood-fill" method, it is also called seed fill to determine and alter the area connected to a given node (called seed) in a multi-dimensional array with some matching attribute (https://en.wikipedia.org/wiki/Flood_fill). However, the algorithm used in the paper does not need seed points to detect erroneous areas. Although these commonly used region extraction algorithms are very similar in general, they often have some subtle differences and have different application scenarios. To avoid confusion, we use the nomenclature "path propagation" in the manuscript based on the characteristics of the algorithm. In fact, the connected-component labeling algorithm and the flood-fill algorithm implemented on the famous computer vision library (such as scikit-image library and OpenCV library) cannot be directly used for our task. In order to effectively process high-resolution DEM data in the AP area (about 19G), we implemented the detection algorithm based on C/C++ language. The erroneous areas detection algorithm takes elevation difference as input, and draws on the idea of the merge strategy in the split and merge segmentation algorithm. It merges spatially adjacent target pixels with similar local elevation offsets into common regions, and each merged regions will be labeled for subsequent corrections. For each correction area, each adjacent target point with similar local elevation offsets will be gradually merged along the searched path starting from any one of the target points. This process is like propagating a certain label along the path to form a correction area, so we use the nomenclature "path propagation". In the revised paper, the difference between the used detection algorithm and the connected component labeling algorithm implemented in existing computing package will be explained and the corresponding reference cited. The nomenclature "path propagation" will be still used in the revised paper, considering the subtle difference of these region extraction methods. If you think it would be better to use another nomenclature (such as connected component labeling), we can revise the nomenclature in the revised paper.

*252: increases*
**Response:** corrected.

*253: increases*
**Response:** corrected.

*254: high-precision*
**Response:** corrected.

*255: multi-scale (for consistency)*
**Response:** corrected.

*258: "Their spatial extent increased from: : : to: : :"*
**Response:** corrected.

*264: Figure 6b*
**Response:** corrected.

*271: Why use the "geographically closest point" in the DEM instead of a bilinear interpolation to the center of the LVIS/ATL06 point? With the TDM DEM at a posting of 12 m, the potential 6 m horizontal bias using a "nearest neighbour" approach from the center of the point can lead to a 3 m vertical error on a 25◦ slope (50% slope), and higher for larger slopes. This procedure might be deteriorating the quality of the validation effort, consider switching to bilinear interpolation of the raster data.*

**Response:** In the revised manuscript, the bilinear interpolation to the center of the laser altimetry point will be adopted as replacement of the searching for the "geographically closest point".

*321: ATL06*
**Response:** corrected.

*333: elimination of the residual height errors*
**Response:** corrected.

*341: Refer to Figure 10 before mentioning profiles L3/L4*
**Response:** corrected.

*371: The vertical scales should be specified sooner, in the Methods section maybe?*

**Response:** The vertical scales will be specified sooner in the methods section in the revised manuscript.

***On the Figures and Tables:***
*Fig. 2: Axis labels of histogram are too small in I. Text of schematics is squeezed vertically in III.*
**Response:** Fig. 2 will be improved in the revised manuscript and all the mentioned issues will be improved.

*Fig. 5: Specify if this is a schematic was created for demonstration purposes, or from real AP data at a specific transect.*

**Response:** Fig. 5 is a schematic created for demonstration purposes, which will be specified in the revised manuscript.

*Fig. 6, 7: Add glacier outlines.*
**Response:** corrected.

*4 References for the review*

*Yamazaki, D. et al. A high-accuracy map of global terrain elevations: Accurate Global Terrain Elevation map. Geophys. Res. Lett. 44, 5844–5853 (2017)*

*Papasaika, H., Poli, D. Baltsavias, E. Fusion of Digital Elevation Models from Various Data Sources. in 2009 International Conference on Advanced Geographic Information Systems Web Services 117–122 (2009).*

*Reuter, H. I., Nelson, A. Jarvis, A. An evaluation of void-filling interpolation methods for SRTM data. Int. J. Geogr. Inf. Sci. 21, 983–1008 (2007)*

*Smith, B. et al. Pervasive ice sheet mass loss reflects competing ocean and atmosphere processes. Science 368, 1239–1242 (2020)*

References (for this response):

Cook, A. J., Murray, T., Luckman, A., Vaughan, D. G., and Barrand, N. E.: A new 100-m Digital Elevation Model of the Antarctic Peninsula derived from ASTER Global DEM: methods and accuracy assessment, Earth system science data., 4, 129-142, 2012.

Howat, I. M., Porter, C., Smith, B. E., Noh, M.-J., and Morin, P.: The reference elevation model of Antarctica, The Cryosphere, 13, 665-674, 2019.

Papasaika, H., Poli, D., and Baltsavias, E.: Fusion of Digital Elevation Models from Various Data Sources, 2009 International Conference on Advanced Geographic Information Systems & Web Services, 2009, 117-122,

Rizzoli, P., Bräutigam, B., Kraus, T., Martone, M., and Krieger, G.: Relative height error analysis of TanDEM-X elevation data, ISPRS J. Photogramm. Remote Sens., 73, 30-38, http://dx.doi.org/10.1016/j.isprsjprs.2012.06.004, 2012.

Rizzoli, P., Martone, M., Gonzalez, C., Wecklich, C., Tridon, D. B., Bräutigam, B., Bachmann, M., Schulze, D., Fritz, T., and Huber, M.: Generation and performance assessment of the global TanDEM-X digital elevation model, ISPRS J. Photogramm. Remote Sens., 132, 119-139, 2017.

Shapiro, L. G.: Connected Component Labeling and Adjacency Graph Construction, in: Machine Intelligence and Pattern Recognition, edited by: Kong, T. Y., and Rosenfeld, A., North-Holland, 1-30, 1996.

Smith, B., Fricker, H. A., Gardner, A. S., Medley, B., Nilsson, J., Paolo, F. S., Holschuh, N., Adusumilli, S., Brunt, K., Csatho, B., Harbeck, K., Markus, T., Neumann, T., Siegfried, M. R., and Zwally, H. J.: Pervasive ice sheet mass loss reflects competing ocean and atmosphere processes, Science, 368, 1239, 10.1126/science.aaz5845, 2020.

Wessel, B.: TanDEM-X ground segment–DEM products specification document, 2016.

---

## Author Comment (AC3) · 28 Feb 2021

Thanks for your additional information. The response letter is submitted as AC2.
* * *

---

## Author Response (AR1)

**Author's response on "High-resolution topography of the Antarctic Peninsula combining TanDEM-X DEM and REMA mosaic"**

*Authors:* Yuting Dong, Ji Zhao, Dana Floricioiu, Lukas Krieger, Thomas Fritz, Michael Eineder

The Cryosphere Discuss., https://tc.copernicus.org/preprints/tc-2020-323/

Referee comments are shown in *black*, our response in blue, changes in red. Line numbers refer to the newly submitted revised manuscript (pdf).

**Authors' response to Anonymous Referee #1**

**General comments:**

*I have now gone through the manuscript more than twice. Generally speaking, this is a well-written manuscript with a thorough description of methods and analysis of results. Authors have used TDM DEM and REMA DEM of the AP region and improved the quality by combining them using propagation algorithm. Authors have demonstrated the improvement by comparing using laser altimetry data captured during two campaigns. Authors have demonstrated the improvement in terms of RMSE and clearly showed the improvement in iterative 3-steps of correction.*

**Response**: We thank the anonymous reviewer for the very constructive and helpful comments. We carefully evaluated all comments and suggestions and point-to-point responses are given in the following. Corresponding improvement was added in the revised manuscript and marked in red.

*My major criticisms are; (1) Authors have not explained the effect of using multi-temporal datasets captured during two different periods and later comparing them with laser altimetry campaign datasets captured in other periods. There is a significant temporal constraint in merging these datasets- I suggest authors describing the effect of using such data and how much error it will introduce in their analysis.*

**Response**: In our work, we want to detect and correct the residual systematic elevation errors in TDM DEM which are mainly introduced by the phase unwrapping (PU) errors. REMA mosaic is used as the reference DEM for the proposed algorithm. The temporal difference between the acquisition time of the REMA mosaic (acquired between 2011 and 2017) and the TDM DEM (acquired between 2013 and 2014) covering AP has negligible impact on the proposed algorithm to detect and correct the residual PU errors in TDM DEM. The reasons can be explained from two aspects. First, the PU errors have distinguishable characteristics from the temporal elevation change. Specifically speaking, the elevation errors in TDM DEM caused by the PU errors are characterized by local elevation discrepancies with abrupt elevation jumps at the boundaries while the temporal changes in elevation are transitional changes with a certain trend. Hence, the proposed path propagation algorithm is based on the

characteristic of the PU errors to automatically detect the elevation jumps at the boundaries of the erroneous regions.

Secondly, to eliminate the influence of the possible temporal elevation changes between the TDM DEM and REMA mosaic, we do not simply correct the TDM DEM to the reference elevation surface of REMA mosaic directly. Instead, we create a buffer zone around each region which has to be corrected. Stable points whose elevation differences with REMA mosaic are less than a given threshold value are extracted from the buffer zone. The average surface elevation fitted from these selected stable points is used as a reference surface for the elevation offset correction as in Fig. 5 in the submitted manuscript.

For validation with the laser altimetry points, the acquisition time difference of the DEM datasets and laser altimetry points were considered in the revised manuscript. Thanks for inspiring us to consider the impacts of temporal changes between different datasets. The second reviewer also points out this issue and he suggest us to incorporate the surface elevation change rate (SECR) product from Smith et al. (2020) which was calculated from ICESat/ICESat-2 surface elevation change (SEC) between 2003 and 2019. The timespan of this SEC product covers the acquisition time of TDM DEM, REMA mosaic and laser altimetry points used in this manuscript. Therefore, we interpolate the SECR based on the acquisition time difference between the DEMs and the laser altimetry data to compensate for the temporal difference before calculating the statistical evaluation results.

**Changes**: The temporal difference has been compensated using the large-scale elevation change maps (Smith et al., 2020), and has been explained in Section 4. The quantitatively evaluation results are shown in the Table 1–3 in the revised paper. Relative discussion and analysis about evaluation results in Table 1–3 have been added to the experiment section.

*(2) Authors generally consider REMA as a ground reference DEM and improve the TDM DEM based on the values of REMA DEM. REMA is about 8m and then they used 100m coarse values where they have voids in the REMA DEM. Propagation algorithm works on two DEMs of slightly different spatial resolution; authors should explain the effect of different spatial resolutions of datasets on the algorithm. Put other words, could you resample your two DEMs on the same resolution and then run the algorithm to find out the performance?*

**Response**: When filling the data voids of the 8-m REMA mosaic, the 100-m REMA mosaic was resampled into the same grid size of 8-m. The proposed path propagation algorithm works on the elevation difference map between the TDM DEM and REMA mosaic. To generate the elevation difference map, the voids-filled REMA mosaic has been resampled into the same spatial resolution with the TDM DEM. The clarification about spatial resolution adjustment is added in the revised manuscript.

**Changes**: How to deal with the resolution differences and voids of REMA DEM has been explained in the Section 4 (P. 13) of the revised paper. In addition, the influence of reference DEMs' spatial resolution and data voids on the results of the proposed algorithm is also discussed in Section 5.3.

*From result tables, I can see improvements varying in different steps of corrections and also for different elevation settings which are expected. However, the significance of final improvement has not been justified by authors. How authors can claim this improvement and not random noise? This is mainly because I can see instances in the result tables where improvement is around 2m.*

**Response**: The TDM DEM elevation bias correction results can be evaluated both qualitatively and quantitatively. For the residual PU errors in TDM DEM, there exist abrupt elevation jumps at the boundaries of the erroneous regions, which have been eliminated after the correction process and validated by visual inspection.

In terms of quantitative validation with laser altimetry points, the statistical results are influenced by whether the laser points are located at the regions with elevation biases or not. Therefore, in the revised manuscript, in order to better validate the proposed correction algorithm, we calculate the statistics of elevation differences between the DEMs and laser altimetry data at the corrected and unchanged regions separately. The elevation errors of the corrected region of the TDM DEM decrease evidently after the PU error correction. For the experimental results at local area in Table 1, the RMSE has decreased from larger than 90 m to less than 5 m and the MAE has decreased from larger than 110 m to less than 5 m at the corrected regions of TDM DEM for both LVIS 2015 and ATL06 datasets. For the experimental results on Hektoria and Green Glaciers in Table 2, the MAE and RMSE of the original TDM DEM larger than 50 m for the LVIS 2015 and larger than 40 m for the ATL06 2019 datasets have been reduced to about 10 m and 8 m for both validation datasets. For the experimental results at the Antarctic Peninsula in Table 3, the RMSEs have jumped from about 100 m to 20 m and the MAEs have decreased from about 60 m to 10 m for the LVIS 2015 and ATL06 2019 datasets. These experimental results show the effectiveness of the proposed algorithm. More discussions about the validation results have been added in the Section 4.

**Changes**: The validation of experimental results have been revised in the Section 4 by calculating the statistics of elevation differences between the DEMs and laser altimetry data at the corrected and unchanged regions separately. The quantitatively evaluation results are shown in the Table 1–3 in the revised paper.

*(3) My concern is why glaciologists would use the newly constructed improved TDM with accuracies still less than original REMA? REMA accuracies were reported less than 1m and TDM accuracies are reported around 10m. The only advantage I can see in merging is to fill data voids or gaps of REMA. From table 3, it is well demonstrated that there is no significant improvement (w.r.t (with respect to) REMA) in RMSE even after improving the TDM. The achievement of this study is to fill the data gaps in REMA using TDM. Put in other words, why reader can't call it as an improved REMA DEM or gapless REMA DEM as the basic foundation of the algorithm is the REMA and not the TDM?*
*Authors must understand the data circularity created by the methodology and see that REMA was used as a reference to correct TDM values and then it is compared against the TDM and original REMA.*
*In general, glaciologists will use this improved DEM if they find it more accurate than the*

*REMA but this is not demonstrated. How if we simply patch up missing elevation values from REMA by TDM and smooth those gap areas? I suggest authors to suggesting future use of corrected TDM in glaciological applications. I encourage authors to describe this in the discussion section.*

**Response**: Here we want to compare TDM DEM and REMA mosaic from the perspectives of absolute vertical accuracy, temporal consistency, data voids and random elevation errors (or relative vertical accuracy).

1.  Although absolute accuracies of REMA mosaic and TDM DEM were reported as less than 1 m and around 10 m, respectively, the method to estimate the statistical accuracy is different and the statistics are estimated at a global level for TDM DEM and circum-Antarctic level for REMA mosaic (Rizzoli et al., 2017;Howat et al., 2019). Therefore, it is not meaningful to compare the two reported accuracies directly over a certain region. As mentioned by Howat et al. (2019), the AP area is a long coastal area with mountainous topography and is challenging for REMA DEM generation. According to our validation results in the submitted manuscript, the corrected TDM DEM has achieved comparable absolute vertical accuracy with the REMA mosaic at AP area. For a better absolute accuracy comparison, we calculate statistics for the corrected and unchanged regions in TDM DEM and compare them to those of REMA mosaic separately in the revised manuscript.

2.  The TDM DEM covering AP was acquired during austral winter of 2013 and 2014, while REMA mosaic covering AP was acquired between 2011 and 2017. The specific acquisition time of REMA mosaic covering AP is shown in Fig. R1a and Fig R1b in year and month, respectively. The short acquisition time of TDM DEM benefits from the high data acquisition efficiency of the TanDEM-X mission and minimizes the influence of temporal surface change which guarantees a good temporal consistency of the TDM DEM.

[Figure]

Figure R1 Acquisition time of REMA mosaic covering AP.

3.    The TDM DEM has fewer data gaps than the REMA mosaic covering AP as shown in Fig. R2. The data voids in REMA mosaic in Fig. R2 are counted as about 8%, while about 0.85% for TDM DEM. For the 0.85% data voids existing in TDM DEM, we reprocess some of the TanDEM-X bistatic data of austral winter of 2013 and 2014 to fill in these data voids in the revised manuscript.

[Figure]

Figure R2 REMA mosaic covering AP and the location of three sample areas. Right panel: detailed comparison of the REMA (left column) and TDM (right column) DEMs in the sample areas.

4.    Based on the elevation errors maps accompanying the DEM products, we can find that TDM DEM has smaller random errors and thus better theoretical relative vertical accuracy than REMA mosaic. In the elevation error map of REMA mosaic in Fig. R3a, the error value at each pixel is the standard error from the residuals of the registration to altimetry data (Howat et al., 2019). Since each tile used for REMA mosaic generation has removed outliers and systematic errors with the preprocessing, the error value at each pixel provides an estimate of the DEM's random elevation errors. The Height Error Map (HEM) values of TDM DEM in Fig. R3b represent for each DEM pixel the corresponding elevation error in form of the standard deviation (Wessel, 2016). The TDM error estimates are exact and reproducible derived from rigorous mathematically correct steps (Wessel, 2016) and are verified in several papers (Rizzoli et al., 2012;Rizzoli et al., 2017). Fig. R4 show the histograms of the random elevation errors of the REMA mosaic and TDM DEM covering AP. Comparing Figs. R3a and R4a to Figs. R3b and R4b, it can be seen that the TDM DEM covering AP has random elevation errors at lower level and thus better theoretical relative vertical accuracy than the REMA mosaic.

[Figure]

Figure R3 Random elevation errors of (a) REMA mosaic and (b) TDM DEM covering AP.

[Figure]

Figure R4 Histograms of random elevation errors of (a) REMA mosaic and (b) TDM DEM covering AP.
Median value and 90% quantile of the errors (90%LE) are marked in red in the histograms.

Based on the above analysis, it can be found that TDM DEM has comparable absolute vertical accuracy, and better completeness, temporal consistency and relative vertical accuracy compared with the REMA mosaic. In this manuscript, we developed algorithm to automatically correct the residual systematic errors in TDM DEM, which is minimally influenced by temporal or penetration differences between TDM DEM and REMA mosaic. The characteristics of an InSAR generated DEM are maintained. Therefore, we can conclude that the corrected TDM DEM is more consistent than a gap-filled REMA mosaic in perspectives of data acquisition time and vertical accuracy.

Several examples are given in the following and more potential applications can be left for the readers to explore.

- Assisting in the generation of TanDEM-X raw DEMs (Rott et al., 2018;Abdel Jaber et al., 2019) and TanDEM-X change DEM (Lachaise et al., 2019) by removing the reference topographic phase, correcting the phase unwrapping errors and calibrating the absolute phase. The corrected TDM DEM is the best choice considering the same TanDEM-X bistatic interferometric data source to generate the DEM products. From a long-term perspective, the TDM DEM acquired between 2013 and 2014 can be combined with other DEM products with a specific time stamp (such as the TDM DEM change DEM generated from data acquired between 2017 and 2019) for surface elevation change analysis over a large spatial coverage. For a voids-filled REMA mosaic with acquisition times between 2011 and 2017 this particular application is not readily possible.
- Since the REMA mosaic is obtained from optical data, the photogrammetric data acquisition is much influenced by the sunlight illumination and therefore data covering different parts of AP were acquired by different years and seasons as in Fig. R1. For applications with an interest in the seasonal elevation changes at AP, TDX DEM acquired in the austral winters of 2013 and 2014 is better suited.
- Before the release of the corrected TDM DEM covering AP with this manuscript, the gapless reference DEMs covering AP are the edited ASTER GDEM with spatial resolution of 100 m (Cook et al., 2012) and the 100-m REMA mosaic whose voids are filled with 100-m ASTER GDEM (Howat et al., 2019). Hence the corrected 12-m TDM DEM can be used for glaciological application at AP with much higher spatial resolution, like calculating the glaciological characteristics for glacier morphological analyses or filling data voids in 8-m REMA mosaic.

**Changes**: In the revised manuscript, we improve the introduction to better clarify our reasons to choose the corrected TDM DEM in potential glaciological application. The comparison between TDM DEM and REMA mosaic in terms of absolute vertical accuracy, temporal consistency, data voids and relative vertical accuracy has also been discussed in the Section 5.4 in the revised paper. In addition, potential applications of the corrected TDM DEM have been added in the Section 5.4. Figs. R1-R4 have been added to the supplementary material.

*(4) Authors have not demonstrated the viability of their methods w.r.t published methods of merging DEMs. This should be discussed in the discussion section.*

**Response**: Thanks for your comments. In our work, an automatic algorithm to detect and correct the residual elevation biases existing in the non-edited TDM DEM was proposed.

Different from the general DEM fusion methods to incorporate the elevation information from different DEMs equally or by weights (Papasaika et al., 2009;Jiang et al., 2014;Gruber et al., 2016;Dong et al., 2018), the proposed algorithm can effectively correct the residual systematic errors in TDM DEM. REMA mosaic is not used to correct the TDM elevation point by point, but to provide reference elevations to correct the TDM elevation biases region by region, which are determined by the characteristics of the phase unwrapping errors. Therefore, this proposed method maintains the characteristics of an InSAR generated DEM and is minimally influenced by temporal or penetration differences between TDM DEM and REMA mosaic.

**Changes**: The references and comparisons to the existing relevant algorithms have been discussed in the Section 5.5 in the revised manuscript. In the introduction, existing methods are also summarized and relevant literature has also been cited.

*Section-wise comments are appended as follows:*
*Abstract: I have carefully read the abstract. It is generally well written, but it is somehow not attractive in the reader's perspective. Authors have failed to mention RMSE in absolute numbers rather they refer percentage. Between line 15-20, I encountered a very long statement which can be shortened. *To generate a consistent, gapless and high-resolution (12 m) topography product of the AP, we combine the TDM DEM and REMA mosaic by detecting and correcting the height errors in TDM DEM through a novel path propagation algorithm and multi-scale height error correction method based on the accurately calibrated REMA mosaic data. *. I would suggest authors to improve the abstract to make it more readable to readers and also boost it with quantitative results at the end.*

**Response**: Thank you for the suggestions. The quantitative indicator RMSE has been added to illustrate the effectiveness of the algorithm. In addition, the sentence between lines 15-20 has been shortened. The abstract has been improved in the revised manuscript.

**Changes**: as above.

*Introduction: Simplify this: 2020). AP is a complex mountainous coastal glacier system and the mass balance of the outlet glaciers is affected by climate and oceanographic forcing and also by the subglacial and surrounding topography (Cook et al., 2012). Good to see available DEMs of AP, mostly are Antarctic-wide. Table S1 provides a good overview but unfortunately, authors have missed a few regional attempts of making DEMs e.g. Fieber et al, 2018: https://doi.org/10.1016/j.rse.2017.10.042. Line 35-45, I would suggest authors revisit regional attempts of constructing DEMs of AP region.*

**Response**: Thank you for the additional reference. The regional attempts of constructing DEMs of AP is referred in the revised manuscript and also added to Table S1 in the supplementary.

**Changes**: as above.

*Line 45: By analysing all these available DEMs, it can be noted that the DEMs of AP have*

*always suffered from large elevation uncertainty, coarse resolution, wide data voids or incomplete data coverage, which are caused by the complex mountainous terrain and cloudy weather of AP. I think this a very generic statement which is applicable for most of the regions of the continent and restricted to only AP.*

**Response**: Thanks. The phrasing of this sentence is improved in the introduction at Line 50.

**Changes**: as above.

*I see authors are using the term posting, are you referring to the spatial resolution? Line 56: To obtain a consistent, gapless and precise DEM product at the high spatial resolution of AP, we intend to create a high-resolution DEM of AP by combining the TDM DEM and REMA mosaic, the two up-to-date DEMs with similar posting. Authors should use comparable posting rather than a similar posting.*

**Response:** Yes, the term posting is referred to spatial resolution in the submitted manuscript. In the revised version, all the "posting" is replaced with "spatial resolution" for consistency. "Similar posting" has been changed into "comparable spatial resolution".

**Changes**: as above.

*In general, the introduction section is not fully developed. It gives a feeling of missing information. For instance, authors should mention about the necessity of accurate and high-resolution DEM in the region and previous literature or applications of DEM used in the AP for various glaciological studies. This would provide a robust background on how accurate DEM can improve these existing studies.*

**Response:** Thanks for your advice. At the beginning of the introduction, we have introduced some typical glaciological applications for the DEM data and relevant literatures are cited. In the revised manuscript, we add emphasis on the characteristics of the AP glaciers which are large in numbers and many of them are with small basins. Therefore, the high-resolution reference DEM data can improve the glaciers' features and dynamics analysis at fine scale.

**Changes**: More clarification has been added to the revised manuscript at Line 56.

*Authors mentioned about Cook et al. (2012) attempt of improving DEM but they ignore other efforts of combining multiple datasets to generate improved DEMs in Antarctica. To my knowledge, there are established attempts of developing DEMs in the Antarctic by combining two or more datasets- Authors should review those efforts in and then place their study at the end and explain how their effort is different than others.*

**Response:** Thanks for your advice. In the revised paper, we add the citation of the newest released TanDEM-X PolarDEM in Line 55 of the introduction. Existing methods are also summarized and cited between Lines 77-79 of the introduction. Then the difference between the proposed algorithm and existing relevant DEM merging algorithms have been analysed in

the introduction. In Section 5.5 of the revised manuscript, contrast experiments to combine DEMs between the proposed algorithm and existing DEM merging algorithms have been performed and discussed.

**Changes**: as above.

*Experimental area and data: Fig. 1: Authors should mention elevation on the colour scale. And may consider naming a few landmark points in the figure to make it more readable. Somehow one yellow box is hidden behind the green coastline. You may consider changing the draping and make the yellow box above the green coastline layer so it is visible. Is the background RAMPv2 DEM or imagery? And you may also consider showing the high-resolution window showing sampling locations. Experimental data: This section is very well written, well done! Minor comment: use the term elevation and height consistently throughout the manuscript.*

**Response:** Thanks for your comments. Fig. 1 is improved with a few landmark points added. To increase the contrast between the DEM and footprints of the laser altimetry points, the elevation values are shown in grey scale. Moreover, in Fig. 8, the DEM elevation values are shown in the colour scale. For the yellow box in the Fig. 1, actually the yellow box is on the top of all the layers. We select a small sample area to present the details of the experimental results marked by the small yellow box. In the revised manuscript, we add zoom-in windows of the sample areas. In addition, the term "height" has been changed into "elevation" in the revised manuscript.

**Changes**: as above.

*Methodology: Line 130-135: use the term ground reference and not the ground truth.*

**Response:** The term "ground truth" has been changed into "ground reference" in the revised manuscript.

**Changes**: as above.

*Fig. 2: In the first section box, I cannot see x and y-axis numbers (Height difference against frequency graph). In section II, what are different shades of blue showing height error regions? Are you missing a colours scale here? I cannot see the text in blue in the Fitted reference surface model of section III. What is this blue line?. Authors should improve the caption of this figure describing the flow process briefly.*

**Response:** Thanks for your comments. In the revised manuscript, x and y-axis numbers is enlarged to be readable. The different shades of blue represent the detected erroneous regions with elevation biases. Each region corresponds to a similar elevation bias value. A color scale is added in section II of Fig. 2. The small figure of section is improved to make every line and text readable. The blue line represents the corrected TDM DEM elevation surface and more details can also be found in Fig. 5 of section 3.1. Fig. 2 is a framework of the proposed algorithm in four different modules and every module corresponds to a sub-section in

methodology 3.1-3.4. For some key process like the elevation bias correction procedure, detailed and enlarged figure is illustrated in each sub-section. Fig. 2 is improved in the revised manuscript with all the details clarified and the caption is also extended.

**Changes**: as above.

*Fig. 3: You may consider showing REMA DEM of the same region shown in (a)*

**Response:** Fig. 3 is improved in the revised manuscript with REMA DEM of the same region added.

**Changes**: as above.

*Authors have mentioned of using empirical threshold but did not mention much about the process of defining the empirical threshold to execute propagation algorithm.*

**Response:** The process to define the empirical threshold has been added in Section 3.4 and Section 5.2 in red.

**Changes**: as above.

*I understood the method of correcting TDM DEM against REMA using propagation algorithm, but I am also concerned about pixel resolution difference between two datasets and then impact of this varying resolution on the algorithm. It is more evident when authors are using 100-m sampled data where REMA has data voids.*

**Response:** Thanks for your comments. The proposed algorithm operates on the elevation difference map generated from TDM DEM minus REMA mosaic. Before the generation of the elevation map, the 8-m REMA mosaic has been resampled to the same spatial resolution of the TDM DEM of 12 m. The data voids of 8-m REMA mosaic are filled by the 100-m REMA mosaic whose voids have been filled by the 100-m edited ASTER GDEM (Howat et al., 2019). The clarification about spatial resolution adjustment of DEM datasets are added in the revised manuscript.

In our experiments, the gapless 8-m REMA mosaic (with data voids filled with 100-m REMA mosaic) has negligible effect on the proposed elevation biases detection and correction algorithm. The examples shown in Fig. 6 and Fig. 7 of the submitted manuscript illustrate that there are data voids in REMA DEM (marked in white) which do not affect the correction process. The reason is that REMA mosaic was not used to correct the TDM elevation point by point, but to provide a reference elevation to correct the TDM elevation biases region by region, which is determined by the characteristics of the phase unwrapping errors.

Ideally the reference DEM should have comparable spatial resolution with the DEM to be corrected like the 12-m TDM DEM and 8-m REMA mosaic. The influence of the spatial resolution differences between different datasets depends on the spatial size of the regions affected by elevation biases and whether these regions cover areas with complex topography. In a word, as long as the biases can be deduced from the elevation difference map with distinguishable boundaries, they can be detected and corrected by the proposed algorithms. In

the revised manuscript, analysis about the effects of spatial resolution difference between DEM datasets is added.

**Changes**: Section 5.4 is added to analyse the influence of reference DEMs' spatial resolution and data voids in the revised manuscript.

**References (for this response):**

Abdel Jaber, W., Rott, H., Floricioiu, D., Wuite, J., and Miranda, N.: Heterogeneous spatial and temporal pattern of surface elevation change and mass balance of the Patagonian ice fields between 2000 and 2016, The Cryosphere, 13, 2511-2535, 2019.

Cook, A. J., Murray, T., Luckman, A., Vaughan, D. G., and Barrand, N. E.: A new 100-m Digital Elevation Model of the Antarctic Peninsula derived from ASTER Global DEM: methods and accuracy assessment, Earth system science data., 4, 129-142, 2012.

Dong, Y., Liu, B., Zhang, L., Liao, M., and Zhao, J.: Fusion of Multi-Baseline and Multi-Orbit InSAR DEMs with Terrain Feature-Guided Filter, Remote Sens., 10, 1511, 2018.

Gruber, A., Wessel, B., Martone, M., and Roth, A.: The TanDEM-X DEM Mosaicking: Fusion of Multiple Acquisitions Using InSAR Quality Parameters, IEEE J. Sel. Topics Appl. Earth Observ. Remote Sens., 9, 1047-1057, 10.1109/jstars.2015.2421879, 2016.

Howat, I. M., Porter, C., Smith, B. E., Noh, M.-J., and Morin, P.: The reference elevation model of Antarctica, The Cryosphere, 13, 665-674, 2019.

Jiang, H., Zhang, L., Wang, Y., and Liao, M.: Fusion of high-resolution DEMs derived from COSMO-SkyMed and TerraSAR-X InSAR datasets, J. Geod., 88, 587-599, 2014.

Lachaise, M., Bachmann, M., Fritz, T., Huber, M., Schweißhelm, B., and Wessel, B.: Generation Of the Tandem-X Change Dem From the New Global Acquisitions (2017-2019), IGARSS 2019-2019 IEEE International Geoscience and Remote Sensing Symposium, 2019, 4480-4483,

Papasaika, H., Poli, D., and Baltsavias, E.: Fusion of Digital Elevation Models from Various Data Sources, 2009 International Conference on Advanced Geographic Information Systems & Web Services, 2009, 117-122,

Rizzoli, P., Bräutigam, B., Kraus, T., Martone, M., and Krieger, G.: Relative height error analysis of TanDEM-X elevation data, ISPRS J. Photogramm. Remote Sens., 73, 30-38, http://dx.doi.org/10.1016/j.isprsjprs.2012.06.004, 2012.

Rizzoli, P., Martone, M., Gonzalez, C., Wecklich, C., Tridon, D. B., Bräutigam, B., Bachmann, M., Schulze, D., Fritz, T., and Huber, M.: Generation and performance assessment of the global TanDEM-X digital elevation model, ISPRS J. Photogramm. Remote Sens., 132, 119-139, 2017.

Rott, H., Abdel Jaber, W., Wuite, J., Scheiblauer, S., Floricioiu, D., Van Wessem, J. M., Nagler, T., Miranda, N., and Van Den Broeke, M. R.: Changing pattern of ice flow and mass balance for glaciers discharging into the Larsen A and B embayments, Antarctic Peninsula, 2011 to 2016, Cryosphere, 12, 1273-1291, 2018.

Smith, B., Fricker, H. A., Gardner, A. S., Medley, B., Nilsson, J., Paolo, F. S., Holschuh, N., Adusumilli, S., Brunt, K., and Csatho, B.: Pervasive ice sheet mass loss reflects competing ocean and atmosphere processes, Science, 2020.

Wessel, B.: TanDEM-X ground segment–DEM products specification document, 2016.

**Author's response on "High-resolution topography of the Antarctic Peninsula combining TanDEM-X DEM and REMA mosaic"**

*Authors:* Yuting Dong, Ji Zhao, Dana Floricioiu, Lukas Krieger, Thomas Fritz, Michael Eineder

The Cryosphere Discuss., https://tc.copernicus.org/preprints/tc-2020-323/

Referee comments are shown in *black*, our response in blue, changes in red. Line numbers refer to newly submitted revised manuscript (pdf).

**Authors' response to Dr. Romain Hugonnet**

*1 General comments*

*The paper is well-prepared and the authors make great effort to present their study rigorously. The text is generally well-written and the quality of Figures is good. The introduction accurately paints the context and limitations for Digital Elevation Models and related studies of the Antarctic Peninsula. The methodology presented for height correction is, to my knowledge, novel and its implementation is sound. The validation effort of the results with high-precision data is valuable. Finally, the resulting impacts are significant and clearly highlighted in the conclusions.*

*However, I have several major concerns. In decreasing order of importance, those are:*

*1. the general logic when combining the TDM and REMA DEMs,*

*2. the relation to existing methodologies,*

*3. the limits of the validation with IceBridge and ICESat-2,*

*4. the statistical estimators used in the study.*

*Those are detailed below.*

**Response:** Firstly, we want to thank Dr. Romain Hugonnet for the time and effort put in this detailed and thorough review. We carefully evaluated all comments and suggestions, which are extremely valuable in improving the paper. As for the four major concerns raised by this review, point-to-point responses and corresponding changes are given in the following. For better clarification, we add Figs. R1-R4 in this response letter and all the figures and the corresponding clarification are added into the revised manuscript or the revised supplementary.

*2 Specific comments*

*2.1 Combining TDM and REMA DEMs*

*With 2 DEMs covering approximately all the AP, one can ask himself: what is the best reference DEM to use? What I draw from the authors presentation is that they decided to correct TDM DEM with REMA DEM due to:*

*A) the short timespans of acquisitons of TanDEM-X (2013-2014) while REMA is based on WorldView acquisitions spanning a longer period of 2009-2017. This shorter timespan of TDM leads to less issues with glacier elevation change over time on the entire AP.*

*B) less data gaps in TanDEM-X compared to REMA.*

*These factors that motivated their choice are somewhat "implicit" along the flow of paper, except in the abstract. I think the authors should clearly state all their arguments at once at some point of the main text to make the entire reasoning behind their choice understandable (at the end of introduction? Or the start of methods?)*

Response: Thanks for the comments. We make improvement to the manuscript from the two aspects: 1) add a comparison in terms of absolute vertical accuracy, data voids, temporal consistency and random errors (or relative vertical accuracy) between TDM DEM and REMA DEM at AP in the discussion section to better support the logic to generate a corrected TDM DEM for glaciology applications; 2) add more clarification in the introduction about the logic based on the comparison between the two DEM products.

Change: Section 5.4 has been added about the comparison between TDM DEM and REMA mosaic and more clarification about the logic of our research idea is given in Section 5.4 and last paragraph of the introduction.

*Additionally, I think that this choice is still subject to some discussion. And this for several reasons:*
*• firstly, I do not find the argument "less data gap in TDM DEM" completely valid.*
*As the authors show, TDM DEM contains many height errors, or artefacts, that require correction. Those could be interpreted as data gaps as well. It is not clear how much surface area is affected, but the authors should have quantified this after application of their methodology. Many of REMA data gaps likely originate from photogrammetric blunders, very much like TDM height errors originate from interferometric ones. This would partly invalidate argument B) for selecting TDM.*

Response: The statistics of data voids in TDM DEM and REMA mosaic DEM at AP between 63 °S and 70 °S are about 0.85% and 8% , respectively, within the ADD coastline based on our counts. The data voids are counted from the null value defined in the original data products, which are shown as white regions in Fig. R1. The elevation errors in TDM DEM to be corrected in our study are mainly caused by phase unwrapping errors which belong to systematic errors. Regions with phase unwrapping errors have abrupt elevation offsets to their neighboring areas due to the incorrect determination of height ambiguities of the wrapped phase. These regions cannot be viewed as blunders because the elevation information is effective as long as the offsets are compensated. Taking the REMA mosaic DEM with high absolute vertical accuracy as ground reference, regions with elevation discrepancies are identified and corrected with the proposed algorithm.

Changes: The comparison between TDM DEM and REMA mosaic has been added discussed in Section 5.4 in the revised paper. Fig. R1 comparing data voids of TDM DEM and REMA mosaic has been added to the supplementary.

[Figure]

Fig. R1 REMA DEM mosaic covering AP and the location of three sample areas. Right panel: detailed comparison of the REMA (left column) and TDM (right column) DEMs in the sample areas.

• *secondly, the REMA DEM is a mosaic based on WorldView 2 m and 8 m strips, freely available through the Polar Geospatial Center. I have not checked, but it is possible that most of the AP is covered by WorldView acquisition within a 2/3- year timespan of each other instead of suffering from the full 2009-2017 deviation. This would partly invalidate argument A) for selecting TDM.*

Response: In our study we are using the REMA mosaic tiles which are generated by the quality-controlled REMA strip DEMs. The specific acquisition time of REMA DEM covering AP is shown in Fig. R2a and Fig R2b in year and month, respectively. So it is definitely longer than 2/3 years.

Changes: In the revised manuscript, we mention more precisely the time span to generate REMA DEM at AP as 2011-2017 instead of 2009-2017 and we added Fig. R2 to the supplementary.

[Figure]

Fig. R2 Acquisition time of REMA DEM mosaic covering AP.

• *finally, the validation effort shows REMA DEM to consistently have higher vertical precision than the TDM DEM (Table 1, 2, 3: value of 90% quantile for REMA vs TDM). As a potential user, I would highly prefer to have a "consistent "AP DEM with most of its coverage based on the dataset with the highest vertical precision, which here is REMA.*

Response: Based on the validation result in Table 1-3, we would like to say that the TDM DEM has comparable absolute vertical accuracy with the REMA DEM. Because the REMA DEM has more data voids than TDM DEM, the number of points used for validation of TDM DEM is much larger than of REMA DEM. For example, there are 31,764,790 and 33,246,648 points from LVIS 2015 datasets used for validating the accuracy of REMA DEM and TDM DEM, respectively. The discrepancies in the numbers of verification points partly account for the differences in the statistics. Therefore, we extract the intersection of the points from REMA DEM and TDM DEM for validation in the revised manuscript. We improve the validation method and the statistical estimators in the revised manuscript. Discussions of validation results in the revised manuscript are improved based on the new results.

In the revised manuscript, we add a discussion section about comparison between REMA DEM and TDM DEM in terms of absolute vertical accuracy, data voids, temporal consistency and random elevation errors. The former three points have been explained in the submitted manuscript and will be clarified and improved in the revised manuscript. The fourth point (about random elevation errors) is added to the revised manuscript. Based on the elevation errors maps accompanying the DEM products, we found that TDM DEM has smaller random errors and better theoretical relative height accuracy than REMA DEM. In the elevation error map of REMA DEM in Fig. R3a, the error value at each pixel is the standard error from the

residuals of the registration to altimetry data (Howat et al., 2019). Since each tile used for REMA mosaic DEM generation has removed outliers and systematic errors with the preprocessing, the error value at each pixel provides an estimate of the DEM's random elevation errors. The Height Error Map (HEM) values of TDM DEM in Fig. R3b represent for each DEM pixel the corresponding height error in form of the standard deviation (Wessel, 2016). The TDM error estimates are exact and reproducible derived from rigorous mathematically correct steps (Wessel, 2016) verified in several papers (Rizzoli et al., 2012;Rizzoli et al., 2017). Fig. R4 shows the histograms of the random elevation errors of the REMA DEM and TDM DEM covering AP. Comparing Fig. R3a and R4a to Fig. R3b and R4b, it can be seen that the TDM DEM covering AP has random elevation errors at lower level and thus better theoretical relative elevation accuracy than the REMA mosaic DEM.

Changes: The validation process with laser altimetry data is all improved in the revised manuscript as in Tables 1-3. The comparison between TDM DEM and REMA mosaic is added in Section 5.4. Figs. R1-R4 have been added to the supplementary.

[Figure]

Figure R3 Random elevation errors of (a) REMA DEM and (b) TDM DEM covering AP.

[Figure]

Fig. R4 Histograms of random elevation errors of (a) REMA DEM and (b) TDM DEM covering AP. Median value and 90% quantile of the errors (90%LE) are marked in red in the histograms.

*I understand that this choice is complex and that the methodology developed by the authors is already dependent on the type of blunders present in TDM DEM which might not be reapplicable to REMA with its own blunders and larger-scale data gaps. Therefore, although this choice is directionally important for the study, it certainly remains fully with the authors. A correction based on TDM DEM is undeniably valuable. In any case, the authors should provide:*
*• a Figure (maybe in the Supplementary) showing the surface area coverage over time of available REMA DEM strips for the AP. If many strips are closely available in time with good coverage of the AP (>80%), this really poses the question of correcting TDM blunders instead of correcting/gap-filling REMA.*

Response: The comparison of data voids of TDM DEM and REMA mosaic shown in Fig. R1 and the specific acquisition time of REMA DEM covering AP shown in Fig. R2 have been added into the supplementary.

Changes: As above.

*• a Table with coverage statistics + mean vertical correction for identified blunders at each multi-scale step of the methodology somewhere in the main manuscript.*

Response: Instead of "blunders", this manuscript targets at detecting and correcting the residual systematic errors in non-edited TDM DEM which are mainly introduced by the phase unwrapping errors. A table with coverage statistics and mean vertical correction at each multi-scale was added in the revised manuscript as Table 4, which shows that the mean absolute correction values are reducing evidently after each iteration of correction and the corrected regions are also decreasing in pixel numbers.

Changes: As above.

• *a discussion on the influence of the 8% data gaps in REMA on the correction of TDM DEM should be provided. I acknowledge that this effect is unavoidable, yet adding a paragraph quantifying the possibly omitted TDM blunders in these 8% REMA data gaps would be useful (maybe based on the average surface area of blunders found on the rest of the AP?). Providing a map of the areas possibly affected (where there is no REMA coverage, and possibly high slopes/larger errors in the TDM Height Error Map?) would also be valuable for future users.*

Response: In addition to the 8 m tiles, the REMA mosaic DEM provides reduced-resolution resampled version at 100 m resolution. The reduced-resolution dataset has an alternate filled version in which the data voids in REMA DEM are filled with 100-m ASTER GDEM (Howat et al., 2019;Cook et al., 2012). In our study, we resampled the 100-m filled REMA DEM to fill in the data voids of the 8-m REMA DEM.

The proposed algorithm runs on the elevation difference map generated from TDM DEM minus REMA DEM. In our experiments, the gapless 8-m REMA DEM with data voids filled with 100-m REMA DEM has negligible effect on the proposed elevation biases detection and correction algorithm through visual inspection. The examples shown in Fig. 6 and Fig. 7 of the submitted manuscript illustrate that there are data voids in REMA DEM (marked in white) which do not affect the correction process. The reason is that REMA DEM was not used to correct the TDM elevation point by point, but to provide a reference elevation to correct the TDM elevation biases region by region, which is determined by the characteristics of the phase unwrapping errors.

Ideally the reference DEM should have comparable spatial resolution with the DEM to be corrected like the 12-m TDM DEM and 8-m REMA DEM. The influence of the spatial resolution differences between different datasets depends on the spatial size of the regions affected by elevation biases and whether these regions cover areas with complex topography. In a word, as long as the biases can be deduced from the elevation difference map with distinguishable boundaries, they can be detected and corrected by the proposed algorithms.

A flag map of the areas possibly affected areas (where there is no REMA coverage, and high slopes) is provided which is mentioned in the last paragraph of Section 5.3 and Table S4 of the supplementary material.

Changes: In the revised manuscript, the influence of reference DEMs' spatial resolution and data voids on the results of the proposed algorithm is discussed in Section 5.3.

*Optionally, the authors should consider using individual strips instead of the REMA mosaic product for height error correction. This additional effort could significantly reduce temporal biases related to glacier elevation change by selecting strips closest to 2013-2014 when available (e.g. biases shown on Figure 7f). The authors would however need to co-register those strips individually to TanDEM-X before using them for corrections.*

Response: Theoretically it is a good idea for correcting the residual elevation errors in TDM DEM with REMA tiles closest to 2013-2014, which will minimize the temporal changes between TDM DEM and REMA DEM. However, it will be much more work to do to select the right REMA tiles with high data quality and calibrate these tiles to TDM DEM or altimetry data. The REMA mosaic DEM tiles have already went through the quality-control process through visual inspection and manual correction to remove erroneous regions, as well as accurately calibrated to the laser or radar altimetry data (Howat et al., 2019). Thus we believe that REMA mosaic DEM has high absolute vertical accuracy and is suitable as ground reference. Furthermore, the proposed correction algorithm has taken the elevation differences between REMA DEM and TDM DEM caused by temporal surface changes or penetration depth into consideration from the following two aspects. First, the phase unwrapping (PU) errors have distinguishable characteristics from the temporal elevation change. Specifically speaking, the elevation errors in TDM DEM caused by the PU errors are characterized by local elevation discrepancies with abrupt elevation jumps at the boundary where they occur while the temporal change in elevation or penetration depth are transitional changes with a certain trend. Hence, the proposed path propagation algorithm is based on the characteristic of the PU errors to automatically detect the elevation jumps at the boundaries of the erroneous regions. Secondly, to eliminate the influence of the possible temporal elevation changes between the TDM DEM and REMA mosaic or the differences due to the SAR signal penetration depth into snow and firn, we do not simply correct the TDM DEM to the reference elevation surface of REMA mosaic directly. Instead, we create a buffer zone around each region which has to be corrected. Stable points whose elevation differences with REMA mosaic are less than a given threshold value are extracted from the buffer zone. The average surface elevation fitted from these selected stable points is used as a reference surface for the elevation offset correction as in Fig. 5 in the submitted manuscript.

*2.2 Relation to existing methodologies*
*The paper surprisingly lacks references to previous DEM correction/fusion methodologies. This is true both for the introduction (supposed to explain the context of existing methods and why a new one is necessary), for the methods (supposed to reference/compare to existing methods, if applicable) and discussion (based on the results, what are the benefits of using this specific method compared to others? Qualitatively at least. quantitatively would be even better e.g. by comparing with other methods locally).*
*Many studies have looked at merging DEMs, removing data gaps and improving general DEM quality, for example: Reuter et al. (2007), Papasaika et al. (2009)*
*(full thesis here: https://ethz.ch/content/dam/ethz/special-interest/baug/igp/igp-dam/ documents/PhD_Theses/109.pdf), Yamazaki et al. (2017), etc.*
*The authors should:*

*• provide a scientific context referencing existing methods and justifying a new methodology,*

*• identify and cite possible similar existing methodologies, if applicable.*

Response: Thanks for the comments and suggestions. In our work, an automatic algorithm to detect and correct the residual elevation biases exiting in the non-edited TDM DEM was proposed. Different from the general DEM fusion methods to incorporate the elevation information from different DEMs equally or by weights (Papasaika et al., 2009), the proposed algorithm can effectively correct the residual systematic errors in TDM DEM. REMA DEM is used not to correct the TDM elevation point by point, but to provide reference elevations to correct the TDM elevation biases region by region, which are determined by the characteristics of the phase unwrapping errors. Therefore this proposed method maintains the characteristics of an InSAR generated DEM and is minimally influenced by temporal or penetration differences between TDM DEM and REMA DEM.

Changes: The references and comparisons to the existing relevant algorithms are summarized in the last paragraph of the introduction section and discussed specifically in the discussion section 5.5 in the revised manuscript. The recommended **reference**s are cited.

*2.3 Validation with ICESat-2 and IceBridge*

*Seasonal and temporal biases of validation exist but are omitted in the validation methods and its discussion. Those should be quantified and discussed.*

*For temporal biases, the authors could use low-resolution, large-scale elevation change maps (Smith et al. (2020)) to partition their validation data over the AP. Binning the validation points by category of expected elevation change during the period (e.g., near stable; <0.2 m yr-1, small elevation change >0.2 and <0.5, strong elevation change >0.5) could provide improved statistics to evaluate the results through the validation effort. The impact of seasonal biases in elevation changes between the two validation datasets should also be discussed using known estimates of seasonal cycle in the AP.*

Response: Thanks for the very helpful suggestion to incorporating the large-scale elevation change maps (Smith et al., 2020) for temporal biases analysis. To show the vertical accuracy of different elevation intervals, we partition the laser points based on the elevation ranges in the revised manuscript. The annual surface elevation change is converted to elevation change by multiplying by the acquisition timespan between the DEMs and laser altimetry points. Then the temporal elevation change is compensated from the elevation difference between the DEMs and laser points before calculating the statistics in Tables 1-3.

As for the seasonal biases, we have not found available seasonal changes products to compensate the seasonal changes of surface elevation. Furthermore, for REMA DEM validation in (Howat et al., 2019), laser altimetry data collected within 18 months of the REMA strip acquisition date or mosaic date stamp were selected. For TDM DEM validation in (Rizzoli et al., 2017), ICESat points were selected. In a word, the seasonal elevation changes were not taken into consideration for neither REMA DEM nor TDM DEM vertical accuracy validation with altimetry data (Rizzoli et al., 2017;Howat et al., 2019). Therefore, we ignore the seasonal biases in our validation with laser altimetry data.

Changes: The temporal difference has been compensated using the large-scale elevation change maps (Smith et al., 2020), and has been explained in Section 4. The quantitatively evaluation results are shown in the Tables 1–3 in the revised paper.

*2.4 Statistical estimators*
*Along the study, the authors provide Tables of the same format to quantify the improvement brought out by their methodology (Table 1, 2, 3 S2, S3). They use the mean of elevation difference to the validation data, the root mean square of differences and the 90% quantile of the distribution of elevation differences.*
*Currently, the mean does not bring much information and is hard to interpret due to temporal biases (preceding comments), but also because this statistical estimator is not very robust to outliers. I suggest using the median, as well as binning by category of expected elevation change during the period (preceding comments). Possibly, a Table showing median residuals normalized by the time difference between the validation dataset and the TanDEM-X date (1-2 years for LVIS 2015, 5-6 years for ATL06 2019) would allow for a better comparison between the biases identified in the two datasets.*

Response: Thanks for the advice. In replacement of mean value, the median of the elevation difference is added in the revised manuscript as in Tables 1-3.

Changes: as above.

*The RMSE is generally a good metric, it is however overly sensitive to outliers which is exactly what the TDM DEM is here suffering from. Using this estimator might "oversell" the improvement in the results, especially to the reader unfamiliar with these effects.*
*Consider:*
*• splitting your statistics by category of initial height differences, or showing the statistics independently for the corrected regions (before and after) and the untouched ones (once). I feel the second choice would be preferable.*

Response: Thanks for the suggestions. In the revised manuscript, the statistics of the corrected regions (before and after) and uncorrected ones (once) are calculated as in Tables 1-3.

Changes: as above.

*• using the Mean Average Error (MAE), less sensitive to outliers.*

Response: Mean Average Error (MAE) is adopted as one of the statistical estimators in the revised manuscript as in Tables 1-3.

Changes: as above.

*For the 90% quantile: I see this is currently the raw quantile of the distribution (Table 1, line*

*3, the value is negative). I imagine that the authors want to show a measure of distribution spread (estimate of elevation precision). For this, consider either:*
*• taking the 90% quantile of the absolute value of the elevation difference.*
*• calculating the half-width between the 5% and 95% quantiles of the distribution.*

Response: We calculate the 90% quantile of the absolute value of the elevation difference in the revised manuscript as in Tables 1-3.

Changes: as above.

*3 Technical corrections*
*General on the text:*
*Unless the authors can justify a specific reason, I advocate for the two following changes along the manuscript:*
*1. Pick "elevation" or "height", don't use both interchangeably. Example in the caption of Table 2: Height differences calculated as DEM elevation minus laser height. This is confusing. I suggest using "elevation" everywhere, as it is the term most commonly used (Digital Elevation Model, Reference Elevation Model of Antarctica, etc).*

Response: In the revised manuscript, the "height" is changed into "elevation" for unification.

Changes: as above.

*2. The use of "height error" seems questionable to me. "Error" by itself is not precise enough as it can refer to random errors, (i.e. uncertainties) or refer to systematic errors (i.e. biases). Here, the artefacts in the TDM DEM and the related correction methods developed by the authors fit clearly in the box of systematic errors.*
*Thus, it seems to me that it would be clearer to use "elevation bias" instead of*
*"height error" along the text.*

Response: "Height error" is specified as random errors, (i.e. uncertainties) or systematic errors (i.e. biases) in the revised manuscript.

Changes: as above.

*Text line by line:*
*75-96: Mention % value of data gaps in TDX*

Response: Corrected.

Changes: as above.

*81: high-precision*

Response: Corrected.

Changes: as above.

*88: stereophotogrammetry*

Response: Corrected.

Changes: as above.

*89: such as the Advanced Spaceborne...*

Response: Corrected.

Changes: as above.

*98-112: Please specify the REMA release used (is it r1 or r1.1?)*

Response: It is the REMA mosaic DEM r1.1 used which specified in the revised manuscript in Section 2.2.2 at Line 130.

Changes: as above.

*114-123: Mention exact acquisition date for the LVIS data (season)*

Response: The acquisition data of the LVIS data were during September and October of 2015, which is added to the revised manuscript in section 2.2.3 at Line 135.

Changes: as above.

*136: "the" buffer zone: This is the first mention of such a zone. Change to "a buffer zone" and refer to the related section.*

Response: Corrected.

Changes: as above.

*140-178: All this information is not really specific to the paper Methods: shorten +optionally, move to "Data"?*

Response: I believe the reviewer mentions the TDM DEM errors analysis in Section 3.1. Section 3.1 explains that the remaining elevation errors in TDM DEM causing large inconsistencies are mainly introduced by the systematic elevation errors especially the phase unwrapping errors. The proposed method is adapted to the characteristics of the residual

systematic errors in TDM DEM. Therefore, Section 3.1 will be maintained in the revised manuscript.

*204: To my knowledge, "path propagation algorithm" is not a nomenclature commonly used for this kind of method. This type of "flood-fill" method (https://en.wikipedia.org/wiki/Flood_fill) for region extraction is generally called "region extraction", "blob extraction" or most generally connected-component labeling (CCL) (https://en.wikipedia.org/wiki/Connected-component_labeling). Please adapt the nomenclature, and cite a reference for the algorithm used if applicable, and also possibly its relation to an existing computing package/parallel implementation.*

Response: Thanks for your valuable comment. Indeed, as you said, the nomenclature "path propagation" used in the paper is very similar to the generally connected-component labeling, which is used to detect erroneous areas based on the elevation difference. As we all know, connected-component labeling is generally used to detect connected regions in binary digital images (Shapiro, 1996) (https://en.wikipedia.org/wiki/Connected-component_labeling). However, our input data for the detection algorithm in the paper is elevation difference image, whose value is from negative several thousand to positive several thousand instead of 0 and 1. For the "flood-fill" method, it is also called seed fill to determine and alter the area connected to a given node (called seed) in a multi-dimensional array with some matching attribute (https://en.wikipedia.org/wiki/Flood_fill). However, the algorithm used in the paper does not need seed points to detect erroneous areas. Although these commonly used region extraction algorithms are very similar in general, they often have some subtle differences and have different application scenarios. To avoid confusion, we use the nomenclature "path propagation" in the manuscript based on the characteristics of the algorithm. In fact, the connected-component labeling algorithm and the flood-fill algorithm implemented on the famous computer vision library (such as scikit-image library and OpenCV library) cannot be directly used for our task. In order to effectively process high-resolution DEM data in the AP area (about 19G), we implemented the detection algorithm based on C/C++ language. The erroneous areas detection algorithm takes elevation difference as input, and draws on the idea of the merge strategy in the split and merge segmentation algorithm. It merges spatially adjacent target pixels with similar local elevation offsets into common regions, and each merged regions will be labeled for subsequent corrections. For each correction area, each adjacent target point with similar local elevation offsets will be gradually merged along the searched path starting from any one of the target points. This process is like propagating a certain label along the path to form a correction area, so we use the nomenclature "path propagation". In the revised paper, the difference between the used detection algorithm and the connected component labeling algorithm implemented in existing computing package is explained. The nomenclature "path propagation" will be still used in the revised paper, considering the subtle difference of these region extraction methods. If you think it would be better to use another nomenclature (such as connected component labeling), we can revise the nomenclature in the revised paper.

Changes: In the revised paper, the difference between the used detection algorithm and the

connected component labeling algorithm implemented in existing computing package has been explained in Section 3.2.

*252: increases*

Response: Corrected.

Changes: as above.

*253: increases*

Response: Corrected.

Changes: as above.

*254: high-precision*

Response: Corrected.

Changes: as above.

*255: multi-scale (for consistency)*

Response: Corrected.

Changes: as above.

*258: "Their spatial extent increased from: : : to: : :"*

Response: Corrected.

Changes: as above.

*264: Figure 6b*

Response: Corrected.

Changes: as above.

*271: Why use the "geographically closest point" in the DEM instead of a bilinear interpolation to the center of the LVIS/ATL06 point? With the TDM DEM at a posting of 12 m, the potential 6 m horizontal bias using a "nearest neighbour" approach from the center of the point can lead to a 3 m vertical error on a 25◦ slope (50% slope), and higher for larger slopes. This procedure might be deteriorating the quality of the validation effort, consider switching*

*to bilinear interpolation of the raster data.*

Response: Thanks for the suggestions. In the revised manuscript, the bilinear interpolation to the center of the laser altimetry point is adopted as replacement of the searching for the "geographically closest point".

Changes: as above.

*321: ATL06*

Response: Corrected.

Changes: as above.

*333: elimination of the residual height errors*

Response: Corrected.

Changes: as above.

*341: Refer to Figure 10 before mentioning profiles L3/L4*

Response: Corrected.

Changes: as above.

*371: The vertical scales should be specified sooner, in the Methods section maybe?*

Response: The vertical scales are specified sooner in the methods section in the revised manuscript at Line 276.

Changes: as above.

***On the Figures and Tables:***
*Fig. 2: Axis labels of histogram are too small in I. Text of schematics is squeezed vertically in III.*

Response: Fig. 2 is improved in the revised manuscript and all the mentioned issues are improved.

Changes: as above.

*Fig. 5: Specify if this is a schematic was created for demonstration purposes, or from real AP data at a specific transect.*

Response: Fig. 5 is a schematic created for demonstration purposes, which is specified in the revised manuscript.

Changes: as above.

*Fig. 6, 7: Add glacier outlines.*

Response: Corrected.

Changes: as above.

***4 References for the review***

*Yamazaki, D. et al. A high-accuracy map of global terrain elevations: Accurate Global Terrain Elevation map. Geophys. Res. Lett. 44, 5844–5853 (2017)*

*Papasaika, H., Poli, D. Baltsavias, E. Fusion of Digital Elevation Models from Various Data Sources. in 2009 International Conference on Advanced Geographic Information Systems Web Services 117–122 (2009).*

*Reuter, H. I., Nelson, A. Jarvis, A. An evaluation of void-filling interpolation methods for SRTM data. Int. J. Geogr. Inf. Sci. 21, 983–1008 (2007)*

*Smith, B. et al. Pervasive ice sheet mass loss reflects competing ocean and atmosphere processes. Science 368, 1239–1242 (2020)*

**References (for this response):**

Cook, A. J., Murray, T., Luckman, A., Vaughan, D. G., and Barrand, N. E.: A new 100-m Digital Elevation Model of the Antarctic Peninsula derived from ASTER Global DEM: methods and accuracy assessment, Earth system science data., 4, 129-142, 2012.

Howat, I. M., Porter, C., Smith, B. E., Noh, M.-J., and Morin, P.: The reference elevation model of Antarctica, The Cryosphere, 13, 665-674, 2019.

Papasaika, H., Poli, D., and Baltsavias, E.: Fusion of Digital Elevation Models from Various Data Sources, 2009 International Conference on Advanced Geographic Information Systems & Web Services, 2009, 117-122,

Rizzoli, P., Bräutigam, B., Kraus, T., Martone, M., and Krieger, G.: Relative height error analysis of TanDEM-X elevation data, ISPRS J. Photogramm. Remote Sens., 73, 30-38, http://dx.doi.org/10.1016/j.isprsjprs.2012.06.004, 2012.

Rizzoli, P., Martone, M., Gonzalez, C., Wecklich, C., Tridon, D. B., Bräutigam, B., Bachmann, M., Schulze, D., Fritz, T., and Huber, M.: Generation and performance assessment of the global TanDEM-X digital elevation model, ISPRS J. Photogramm. Remote Sens., 132, 119-139, 2017.

Shapiro, L. G.: Connected Component Labeling and Adjacency Graph Construction, in: Machine Intelligence and Pattern Recognition, edited by: Kong, T. Y., and Rosenfeld, A., North-Holland, 1-30, 1996.

Smith, B., Fricker, H. A., Gardner, A. S., Medley, B., Nilsson, J., Paolo, F. S., Holschuh, N., Adusumilli, S., Brunt, K., Csatho, B., Harbeck, K., Markus, T., Neumann, T., Siegfried,

M. R., and Zwally, H. J.: Pervasive ice sheet mass loss reflects competing ocean and atmosphere processes, Science, 368, 1239, 10.1126/science.aaz5845, 2020.

Wessel, B.: TanDEM-X ground segment–DEM products specification document, 2016.

---

## Author Response (AR2)

**Author's response on "High-resolution topography of the Antarctic Peninsula combining TanDEM-X DEM and REMA mosaic"**

*Authors:* Yuting Dong, Ji Zhao, Dana Floricioiu, Lukas Krieger, Thomas Fritz, Michael Eineder

The Cryosphere Discuss., https://tc.copernicus.org/preprints/tc-2020-323/

Referee comments are shown in *black*, our response in blue, changes in red. Line numbers refer to the newly submitted revised manuscript (pdf).

**Authors' response to Dr. Kenny Matsuoka**

**General comments:**

*I am also happy to see substantial improvements in the manuscript. One reviewer pointed out that Section 5.4 can be shortened, which is a valid point in my opinion. It is not necessary to merge it with Section 3, but please consider a more concise presentation of Section 5.4.*

**Response**: We thank the editor Dr. Kenny Matsuoka very much for the positive feedback and the very helpful comments. We carefully evaluated all comments and suggestions and point-to-point responses are given in the following. Corresponding improvements were added in the revised manuscript and are marked in red. In the revised manuscript, section 5.4 has been shortened and the redundant description with the introduction or section 3 has been removed.

**Changes**: Section 5.4 has been shortened and its subtitle has been changed into "Potential applications of the corrected TDM DEM".

*This paper describes the methods to generate a new DEM, so the release of the associated DEM is critically important. Because the review process is very close to complete, please upload the dataset to obtain DOI, and include DOI to this article. A part of the meta data can be described in data availability section (see my comment below for Table S4).*

**Response & Changes**: Thanks for the advice. The brief description of the meta data has been moved to the data availability and Table S4 has been deleted. Besides, there will be a text file accompanying the data product to describe the meanings of values in the flag file.
As for the DOI of the data, we give a formal website in the data availability as (https://geoservice.dlr.de/web/). Later the data will be released there. However we have to say this will take some more time because of the DLR's legal data policy. Anyhow we are working very hard toward it now and the DEM data will be available soon.

*Below, please find other minor comments:*

*Figure 1: add distance scale to the two zoom-up insets. Cite a reference for RAMP and*

*Matsuoka et al. (2021) for Quantarctica (https://doi.org/10.1016/j.envsoft.2021.105015); the former is a requirement to use Quantarctica, and the latter is the latest recommendation from the Quantarctica project.*

**Response & Changes**: Figure 1 has been improved with distance scales to the two zoom-up insets added. The relevant references have been cited.

*P8L197ff: this paragraph presents the minimum elevation inconsistency, minimum elevation discrepancy and minimum elevation difference. These terms need to be better defined and readability of this paragraph needs to be improved.*

**Response**: All the three terms have been unified as "minimum elevation difference". Clarification about this term has been added. The wording of this paragraph has been improved.

**Changes**: Changes can be found in Point (4) of Section 3.1 in red.

*P13L262: Is the citation of Figure 1 correct?*

**Response & Changes**: Sorry for the mistake. Citation of Figure 1 has been changed into Figure 2 as in P12 Line 265.

*P14L294: briefly describe the magnitude of the temporal corrections based on Smith et al. (2020). For example, add mean and quantile values.*

**Response**: Thanks for your comments. We have added the statistics of the temporal elevation corrections including mean, 10% and 90% quantiles.

**Changes**: Changes can be found in second paragraph of Section 4 in red.

*Tables 2 and 3: do you need to put elevation ranges into parenthesis?*

**Response & Changes**: The parentheses in the tables have been removed.

*Figure 14 and associated text (P30L470-477): the caption needs to be expanded. Why are there so few data points in this figure? The description/interpretation if this figure is also very limited.*

**Response**: Thanks for the very helpful comments. There are respective 7 points in Figure 14 for both LVIS 2015 and ATL06 2019 altimetry data corresponding to different elevation ranges in Figure 13. The slope values are the mean slopes of each elevation range. The corresponding MAEs can be obtained from Tables 1 and 2 same as in Figure 13.

**Changes**: Caption of Figure 14 has been improved with more clarification. More description of Figure 14 has been added (Line 485 in red).

*P32L519: HEM -> DEM, typo?*

**Response & Changes**: Sorry for the incorrect expression. Since Section 5.4 has been improved, this word has been deleted.

*Table S4: this is a description of the dataset associated with this article. So, please move this information to the data availability section (or delete).*

**Response & Changes**: Table S4 has been deleted.

**Author's response on "High-resolution topography of the Antarctic Peninsula combining TanDEM-X DEM and REMA mosaic"**

*Authors:* Yuting Dong, Ji Zhao, Dana Floricioiu, Lukas Krieger, Thomas Fritz, Michael Eineder

The Cryosphere Discuss., https://tc.copernicus.org/preprints/tc-2020-323/

Referee comments are shown in *black*, our response in blue, changes in red. Line numbers refer to the newly submitted revised manuscript (pdf).

**Authors' response to Dr. Romain Hugonnet**

**General comments:**

*The author have done an excellent job accounting for the referee's comments, and also relating to the closely related paper by Wessel et al. (2021) which, since the last iteration, entered discussion in TC.*
*The additional discussion points, updated calculations and tables, and material presented in the Supplementary have satisfactorily addressed all raised points.*

*I only have a comment on the revised text: I found the added section 5.4 of the Discussion a bit long, and redundant with several aspects discussed either the end of the Introduction or in section 3.*
*I suggest shortening the related section, or possibly embedding it totally in section 3.*

*Other than that, I see no additional change necessary for the publication of this manuscript.*

**Response**: We thank Dr. Romain Hugonnet very much for the positive response. We carefully considered your advice to re-organize the redundant section 5.4 which is also the comments from the editor. In the revised manuscript, section 5.4 has been shortened and the redundant description with the introduction or section 3 has been removed. The improved content is in red.

**Changes**: Section 5.4 has been shortened and its subtitle has been changed into "Potential applications of the corrected TDM DEM".

---

## Author Response (AR3)

**Author's response on "High-resolution topography of the Antarctic Peninsula combining TanDEM-X DEM and REMA mosaic"**

*Authors:* Yuting Dong, Ji Zhao, Dana Floricioiu, Lukas Krieger, Thomas Fritz, Michael Eineder

The Cryosphere Discuss., https://tc.copernicus.org/preprints/tc-2020-323/

Referee comments are shown in *black*, our response in blue, changes in red. Line numbers refer to the newly submitted revised manuscript (pdf).

**Authors' response to Dr. Kenny Matsuoka**

**General comments:**

*Thanks for your careful revisions. Your responses are satisfactory, and I am happy to accept this article. Please correct one typo in Fig. 1 caption (Quantarctica, not QAntarctica), when you submit the final file for the production.*

**Response& Changes**: We sincerely thank the editor in chief for taking the time to read our manuscript, provide us with constructive comments, and finally agree to accept our article.

We carefully corrected the "QAntarctica" to "Quantarctica" in Fig. 1 caption and marked in red in the revised manuscript.